**REPORT**

# Protein Kinase C promotes peroxisome biogenesis and peroxisome–endoplasmic reticulum interaction

Anya Borisyuk[1], Charlotte Howman[2], Sundararaghavan Pattabiraman[3], Daniel Kaganovich[2,4], and Triana Amen[1,2]

**Peroxisomes carry out a diverse set of metabolic functions, including oxidation of very long-chain fatty acids, degradation of D-amino acids and hydrogen peroxide, and bile acid production. Many of these functions are upregulated on demand; therefore, cells control peroxisome abundance, and by extension peroxisome function, in response to environmental and developmental cues. The mechanisms upregulating peroxisomes in mammalian cells have remained unclear. Here, we identify a signaling regulatory network that coordinates cellular demand for peroxisomes and peroxisome abundance by regulating peroxisome proliferation and interaction with ER. We show that PKC promotes peroxisome PEX11b-dependent formation. PKC activation leads to an increase in peroxisome–ER contact site formation through inactivation of GSK3β. We show that removal of VAPA and VAPB impairs peroxisome biogenesis and PKC regulation. During neuronal differentiation, active PKC leads to a significant increase in peroxisome formation. We propose that peroxisomal regulation by transient PKC activation enables fine-tuned responses to the need for peroxisomal activity.**

## Introduction

The peroxisome is a small organelle with a large functional repertoire (De Duve and Baudhuin, 1966; Wanders, 2004). Prominent examples of peroxisomal function are alpha- and beta-fatty acid oxidation, bile acid and ether-lipid synthesis, degradation of D-amino acids and leukotrienes, the glyoxylate cycle and methanol metabolism in yeast, bioluminescence in fireflies, plant hormone synthesis, and degradation of hydrogen peroxide (Adams and Miller, 2020; Kunze et al., 2006; Martin et al., 2012; Paudyal et al., 2017; Spiess and Zolman, 2013; van den Bosch et al., 1992; van der Klei et al., 2006; Zaar, 1996). In addition to their numerous metabolic functions, peroxisomes are increasingly seen as signaling regulators (Alexander et al., 2008; Benjamin and Hall, 2013). Several kinases, (e.g., the NDR2 kinase regulator of ciliogenesis) contain peroxisome localization signals (Abe et al., 2017; Kataya et al., 2022). The peroxisomal membrane has been shown to recruit multiple signaling regulators, including the tuberous sclerosis complex (TSC1/2) regulator of the target of rapamycin (mTOR) kinase pathway, which binds to peroxisomal proteins PEX19 and PEX5 (Benjamin and Hall, 2013; Gough, 2013; Zhang et al., 2013).

Due to the diversity of peroxisomal functions, the cellular demand for peroxisomes depends on specific conditions, such that peroxisomes must be rapidly mass-produced during specific instances of stress adaptation and development; for example, an increase in peroxisome number in the liver upon ethanol stress protects cells from alcohol-associated liver injury (Chang et al., 1999; Mitra et al., 2024), necessitating precise regulation. Disruption of peroxisomal biogenesis adversely affects downstream cellular adaptive functions, including autophagy, cholesterol metabolism, immune response, and ciliogenesis (Cai et al., 2018; Charles et al., 2020; Di Cara et al., 2017; Magalhaes et al., 2016; Nath et al., 2022). Not surprisingly, therefore, multiple human disorders result from defects in peroxisome biogenesis (Aubourg and Wanders, 2013; Braverman et al., 2013; Wanders, 2004). Due to the "on demand" need for peroxisomal metabolism, the key to understanding the role of peroxisomes in cell biology and disease is to investigate the mechanisms regulating peroxisomal biogenesis.

Peroxisomes form through fission of preexisting peroxisomes (Fujiki, 2000; Ghaedi et al., 2000; Schrader et al., 2012; Schrader et al., 2016) and de novo from intracellular membranes (Hettema et al., 2014; Hoepfner et al., 2005; Lazarow and Fujiki, 1985; Tabak et al., 2013). The peroxisome biogenesis machinery consists of peroxins (PEX), which are conserved in fungi, plants, and animals (Itoyama et al., 2013; Matsuzaki and Fujiki, 2008; Schrader and Pellegrini, 2017). Deletion of PEX3, PEX19, and PEX16 results in conditionally viable cells lacking peroxisomes (Agrawal and Subramani, 2016; Gotte et al., 1998; Hoepfner et al., 2005; Kim and Mullen, 2013; Sacksteder et al., 2000; Titorenko and Rachubinski, 1998). Peroxisome division is initiated by the

[1]Global Health Institute, Faculty of Life Sciences, Ecole Polytechnique Fédérale de Lausanne, Lausanne, Switzerland; [2]School of Biological Sciences, University of Southampton, Southampton, UK; [3]2Prime Ltd, Southampton, UK; [4]Conquest, Palo Alto, CA, USA.

Correspondence to Triana Amen: t.amen@soton.ac.uk; Daniel Kaganovich: d.kaganovich@soton.ac.uk.

PEX11 protein family, which orchestrates a multistep process that involves peroxisome membrane elongation and prime assembly of additional peroxisome division components, including MFF, FIS1, and DNM1L (Koch et al., 2010; Li and Gould, 2002; Orth et al., 2007; Schrader et al., 1998; Schrader et al., 2022). Both peroxisome biogenesis mechanisms are regulated by several non-mutually exclusive pathways. Transcriptional peroxisome proliferator receptor alpha (PPARα) increases the level of peroxisomal proteins, including acyl coenzyme A oxidase 1 and division factor PEX11a (Rakhshandehroo et al., 2009). Upregulation of PEX11b, and to a much lesser extent PEX11a, is sufficient to promote peroxisome division (Schrader et al., 1998). Peroxisomal substrates, such as very long-chain fatty acids, can stimulate PPAR response and promote peroxisome formation (Guo et al., 2024). Peroxisome tethering to the ER through a membrane contact site that is formed between acyl-coenzyme A–binding domain protein 5 (ACBD5) and the ER protein vesicle-associated membrane protein-associated protein (VAPB) can source peroxisomal membrane for dividing organelles (Costello et al., 2017; Hua et al., 2017). Peroxisome-ER membrane contacts are negatively regulated by glycogen synthase kinase beta (GSK3β) that phosphorylates ACBD5, preventing interaction with VAPB and reducing peroxisome–ER contacts (Costello et al., 2017; Kors et al., 2022b; Kors et al., 2022c). Notably, abolishing the peroxisome–ER contacts through knockout of VAPB and its homolog VAPA leads to a significant defect in peroxisome biogenesis (Korotkova et al., 2024).

Here, using a tool compound kinase inhibitor screen, we identify positive and negative regulators of peroxisome abundance, including PKC as a positive driver. We show that active PKC inhibits GSK3β, promoting peroxisome–ER tethering and peroxisome biogenesis. We further show that PKC regulation is dependent on PEX11b and VAPs. Inhibition of PKC reduces peroxisome–ER contacts, peroxisome abundance, and function.

## Results and discussion

### Tool compound kinase inhibitor screen reveals signaling regulators of peroxisome abundance

We postulated that a sensitive and tractable assay of peroxisome abundance was the key to discovering regulators of peroxisome formation and function. We therefore developed a new tool for quantifying peroxisomes in live cells. We used CRISPR/Cas9 gene editing to genomically fuse GFP to the C terminus of the transmembrane peroxisomal protein PMP70 in HEK293T cells (Fig. 1 A; and Fig. S1, A and B). PMP70-GFP localized to peroxisomes and peroxisomal protein import was not impaired (Fig. S1, C–E). Considering the prominent role of kinase signaling in metabolic regulation, we set out to identify kinase modulators of peroxisome abundance. We opted for a small molecule "tool compound" library of 152 kinase inhibitors (Fig. 1 B, Fig. S1 F, Fig. S2, Fig. S3, and Table S1). Peroxisome cellular density (abundance) was calculated as the number of peroxisomes per square micron of the cytoplasmic space in confocal microscopy images, using the overexpression of PEX19 peroxisome biogenesis factor

as a positive control (Gotte et al., 1998) and overexpression of PEX3 as a negative control (Fig. 1 C) (Yamashita et al., 2014). PEX3 promotes peroxisome degradation when overexpressed, reducing peroxisome number (Yamashita et al., 2014).We found that 16 kinase inhibitors reduced, and 5 upregulated, the number of peroxisomes in HEK293T cells (Fig. S1, F and G). We performed a secondary screen of the hits in untagged human fibroblasts (Fig. 1 D), validating 10 compounds that reduced and 2 that upregulated the number of peroxisomes (Fig. 1 E). We then associated the molecules with the kinases they inhibit and mapped the kinases on the human kinome (Fig. 1 F). Positive regulators (whereby inhibition of the kinase reduces peroxisome number) included PKC, TGFβR, MEK1/2, ERK2, PDHK, IGF1R, and ALK4/7. Negative regulators (whereby inhibition of the kinase increases peroxisome numbers) were CK2 and IKKβ (Fig. 1 F). Two different tool compounds targeted the TGFβ and PKC pathways. Interestingly, TGFβ is a known positive regulator of peroxisome proliferation (Azadi et al., 2020), underlining the broad coverage of our tool compound screen. We further focus on the previously uncharacterized role of PKC in regulating peroxisome abundance.

### PKC delta positively regulates peroxisome abundance but is not essential for peroxisome biogenesis

PKC is a serine/threonine kinase that controls GSK3 activity, eisosome assembly, response to oxidative stress, and lysosome biogenesis, among many other cellular functions (Alvi et al., 2007; Amen and Kaganovich, 2020c; Anastasia et al., 2012; Clarke et al., 2017; Larsson, 2006; Li et al., 2016; Regala et al., 2005; Thai et al., 2017; Vilella et al., 2005). The PKC superfamily consists of four distinct subgroups (Azzi et al., 1992) that share a kinase domain and have a variety of regulatory domains that regulate spatiotemporal specificity (Oancea and Meyer, 1998) (Fig. 2 A). First, we analyzed which PKC isoforms are inhibited by the small molecules that we identified. Go6983, a PKC inhibitor, prevents kinase activity as well as the steric PKC interaction with the substrates (Ma et al., 2018). Chelerythrine and D-erythrosphingosine C18 are potent inhibitors of PKC (Herbert et al., 1990; Khan et al., 1990). The human PKC superfamily consists of classical (cPKC): α, βI, βII, and γ; novel (nPKC): δ, ε, θ, and η; atypical: ζ and ι/λ; and PKN isoforms (Azzi et al., 1992; Oancea and Meyer, 1998). Identified molecules are not specific to one isoform and inhibit cPKC (α, β. and γ), nPKC (nPKC δ), and at least one atypical isoform (ζ), decreasing peroxisome number (Fig. 1 E). We, therefore, decided to test whether PKC activation is sufficient to trigger peroxisome proliferation. First, we overexpressed a representative isoform of each class—PKC(α), PKC(δ), and PKC(ζ)—fused to mCherry and mCherry as a control and measured peroxisome number (Fig. 2, A and B). Only PKCδ overexpression was able to induce peroxisome proliferation (Fig. 2 A and Fig. S1 H). Both cPKC and nPKC bind to and are activated by phorbol esters (PMA) (Boni and Rando, 1985; Sakai et al., 1997). We incubated human fibroblasts with PMA, which resulted in a significant increase in peroxisome number (Fig. 2 C). Together, these data argue that PKCδ is a positive regulator of peroxisome abundance in human cells. To assay whether PKCδ is the only PKC isoform that regulates peroxisome formation, we

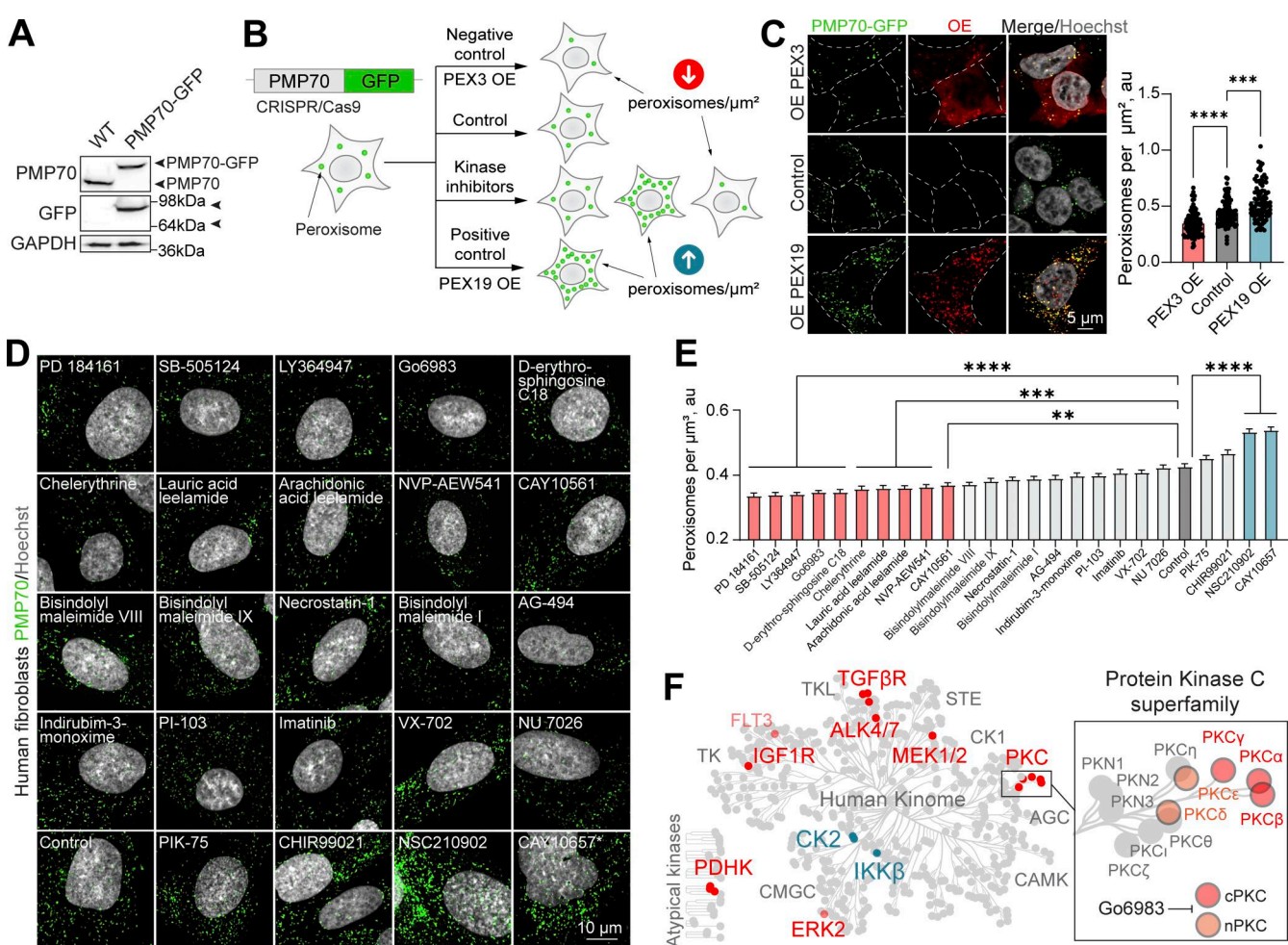

Figure 1. **Kinase inhibitor screen reveals signaling regulators of peroxisome abundance. (A)** Western blot of WT and CRISPR/Cas9 PMP70-GFP HEK293T cells. Arrowheads indicate the size shift of the tagged PMP70-GFP. **(B)** Schematic of the peroxisome biogenesis regulators screen. CRISPR/Cas9 PMP70-GFP HEK293T cells were incubated in control or 1 μM of small molecules for 2 days. PEX3 overexpression was used as a negative control, and PEX19 overexpression was used as a positive control. Refer to Fig. S1, F and G, and Figs. S2 and S3 for the screen details. **(C)** Confocal microscopy of CRISPR/Cas9 PMP70-GFP HEK293T cells overexpressing PEX3-myc, flag-PEX19, or an empty vector. Images are included in Fig. S2 as controls. Quantification shows the number of peroxisomes per square micron of the cytoplasm in the 2D confocal image, mean ± SEM, N = 100 cells pooled from three biological repeats, ***P < 0.001, ****P < 0.0001, Kruskal–Wallis test. **(D–E)** Confocal microscopy of peroxisomes in human primary fibroblasts AFF11 treated with indicated kinase inhibitors for 2 days (1 μM). Peroxisomes were visualized using PMP70 antibody, and nuclei were stained with Hoechst (10 μg/ml). Representative images are shown; scale bar: 10 μm, *—abnormal nuclear morphology. **(E)** Quantification shows the number of peroxisomes per square micron of the cytoplasm in the flattened 3D image (indicated as cubic micron), mean ± SEM, N = 100 cells pooled from three biological repeats, **P < 0.01, ***P < 0.001, ****P < 0.0001, Kruskal–Wallis test. **(F)** Identified kinase inhibitors plotted on the human kinome network (Manning et al., 2002; Metz et al., 2018). Positive regulators (inhibition decreases the number of peroxisomes) are indicated in red, and negative regulators (inhibition increases the number of peroxisomes) are indicated in blue. PKC superfamily is shown in the inlet; G06983 inhibits indicated PKC isoforms. Source data are available for this figure: SourceData F1. OE, overexpressing.

constructed a CRISPR/Cas9 KO of PKCδ. However, the number of peroxisomes was not significantly altered in PKCδ KO compared with WT (Fig. S1 I), implying that several PKC isoforms can coordinate the response. To independently test that inhibition of PKC affects peroxisome biogenesis, we used a radioactive assay of peroxisomal fatty acid oxidation that we previously established (Fig. 2 D) (Korotkova et al., 2024). Inhibition of PKC resulted in a significant reduction of peroxisome function but did not inhibit it completely (Fig. 2 D), consistent with a reduction of peroxisome numbers. PKC, therefore, regulates peroxisome biogenesis—formation and function of peroxisomes. Next, we investigated the mechanism of this new role of PKC in peroxisome biology.

**PKC induces PEX11b-dependent peroxisome formation**

Peroxisome abundance is maintained by peroxisome biogenesis and peroxisome degradation (Agrawal and Subramani, 2016; Fujiki et al., 2014; Kim et al., 2006; Schrader and Pellegrini, 2017; Schrader et al., 1998; Zientara-Rytter and Subramani, 2016). To understand how PKC activation results in an increased number of peroxisomes, we first reproduced the PEX19 functional complementation assay (Matsuzono et al., 1999) in HEK293T cells (Fig. 3 A). This assay monitors de novo peroxisome formation in a PEX19 KO cell line that lacks peroxisomes by complementing it with PEX19 and measuring peroxisome formation. We constructed CRISPR/Cas9 PEX19 KO cells that lack peroxisomes (Korotkova et al., 2024). In line with published observations, it

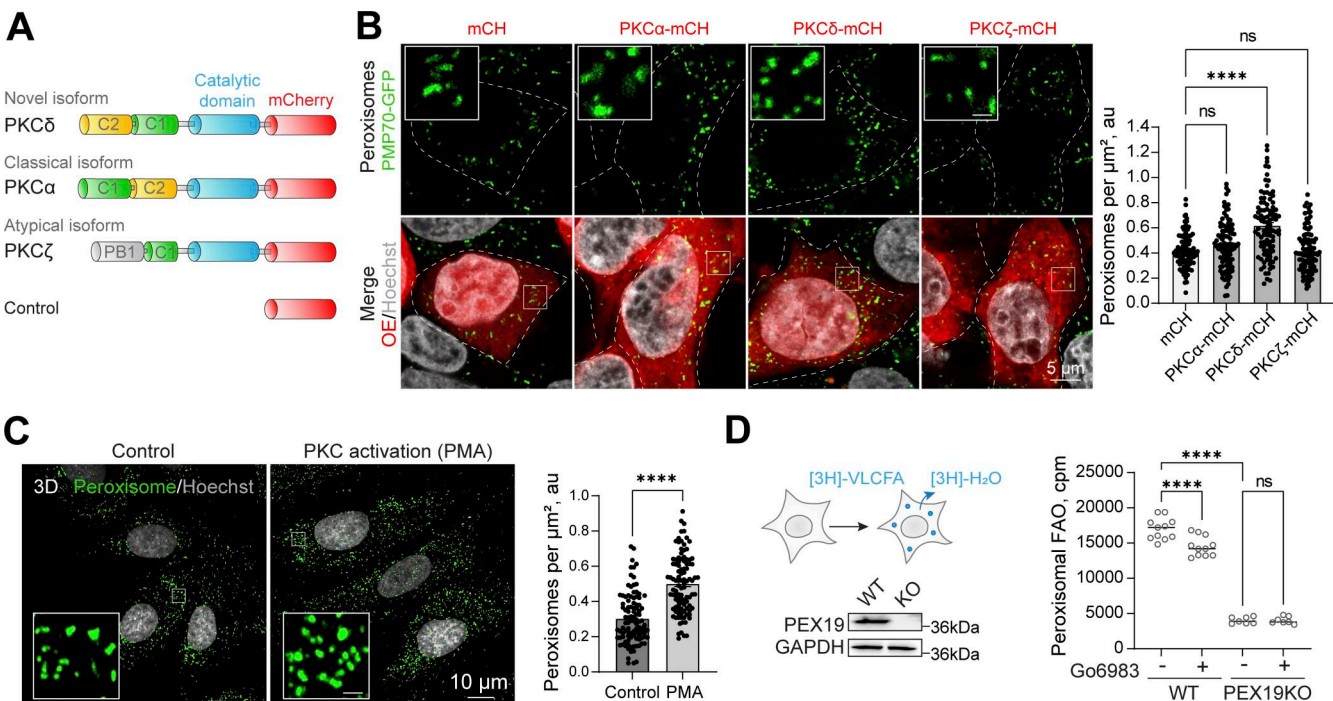

Figure 2. **PKC Delta positively regulates peroxisome abundance. (A)** Schematic of different PKC isoforms used in B. **(B)** Confocal microscopy of HEK293T CRISPR/Cas9 PMP70-GFP cells overexpressing (OE) PKCα-mCherry, PKCδ-mCherry, or PKCζ-mCherry. See zoomed out images in Fig. S1 H. Quantification shows the number of peroxisomes per square micron of the cytoplasm in the 2D confocal image, mean ± SEM, N = 100 cells pooled from three biological repeats, ****P < 0.0001, Kruskal–Wallis test. Scale bar: 5 µm. **(C)** Confocal microscopy of peroxisomes in human primary fibroblasts AFF11 treated with PMA (0.5 µM) for 1 day. Peroxisomes were visualized using PMP70 antibody, and nuclei were stained with Hoechst (10 µg/ml). Representative images are shown; scale bar: 10 µm. Quantification shows the number of peroxisomes per square micron, mean ± SEM, N = 103 cells pooled from three biological repeats, ****P < 0.0001, Mann–Whitney test. **(D)** Radioactive peroxisomal FAO measurement using 3H-docosanoic acid in HEK293T WT or PEX19KO in control or Go6983 (5 µM for 48 h) conditions. Quantification shows the number of counts per minute, mean ± SEM, N = 6–12 pooled from three biological repeats, ***P < 0.001, ****P < 0.0001, One-way ANOVA. Source data are available for this figure: SourceData F2. ns, nonsignificant.

takes more than 24 h to restore peroxisomes de novo (Sugiura et al., 2017) (Fig. 3 B), leading to a 70% restoration on day 4 (Fig. 3, B–D). PKC inhibition or activation did not change the de novo peroxisome formation. Peroxisome division is thought to be a more frequent event than peroxisome de novo formation (Jean Beltran et al., 2018). Peroxisome proliferation was upregulated with a short 2-h exposure to PMA that activates PKC, pointing toward the proliferation pathway as a point of PKC regulation of peroxisome abundance (Fig. 3 E). Peroxisome proliferation is controlled by PEX11b—a key PEX responsible for early membrane remodeling, elongation, and association with the rest of the division machinery (Li and Gould, 2002; Schrader et al., 2016; Schrader et al., 1998). To understand whether PKC regulates PEX11b-induced peroxisome division, we constructed a CRISPR/Cas9 PEX11b KO in PMP70-GFP background in HEK293T cells (Fig. 3 F). PEX11b KO has a reduction in peroxisome abundance, as was previously shown, and it did not respond to PKC inhibition by further reduction of peroxisome numbers (Fig. 3, G and H) (Erdmann and Blobel, 1995; Li and Gould, 2002). Additionally, overexpression of PKCδ in PEX11b KO did not increase the number of peroxisomes (Fig. S4 A).

Peroxisome abundance is a balance between several pathways that produce and degrade peroxisomes, as well as PPARα-regulated transcriptional response (Honsho et al., 2016; Sakai et al., 2006; Schrader et al., 2016). Classical isoforms of PKC

regulate PPARα receptor (Blanquart et al., 2004). One possibility is that PKC inhibition leads to reduced transcription of PEX genes. To rule out PKC-driven regulation of the PPARα peroxisomal transcriptional regulator (Shimizu et al., 2004), we measured PPARα levels following PKC inhibition and activation. PKC inhibition had no effect on PPARα, whereas PKC activation with PMA showed a modest increase in PPARα levels, though not PPARα activity (Fig. S4, B–E) (Amen and Kaganovich, 2020b). Another possibility is that PKC inhibition increases pexophagy (Deosaran et al., 2013). To test this, we silenced the NBR1 receptor, which did not affect peroxisome abunbdance, nor did silencing of NBR1/NIX (Barone et al., 2023) in control and PMA conditions (Fig. S4, F and G), confirming that PKC does not upregulate peroxisomes by inhibiting their degradation. The regulation of peroxisome abundance by pexophagy may be more relevant during autophagy response upregulation, such as starvation response (Honsho et al., 2016; Iwata et al., 2006; Zientara-Rytter and Subramani, 2016). The timing of PKC-induced peroxisome induction, as well as its dependence on PEX11b, suggested peroxisome division as the mechanism of PKC regulation. We tested whether PEX11b changes its association with other division factors. However, neither the interaction of PEX11b with other factors of peroxisome division machinery nor PEX19 (Fig. S5, A and B) were significantly affected. To confirm PKC effect on the early steps of peroxisome division—elongation

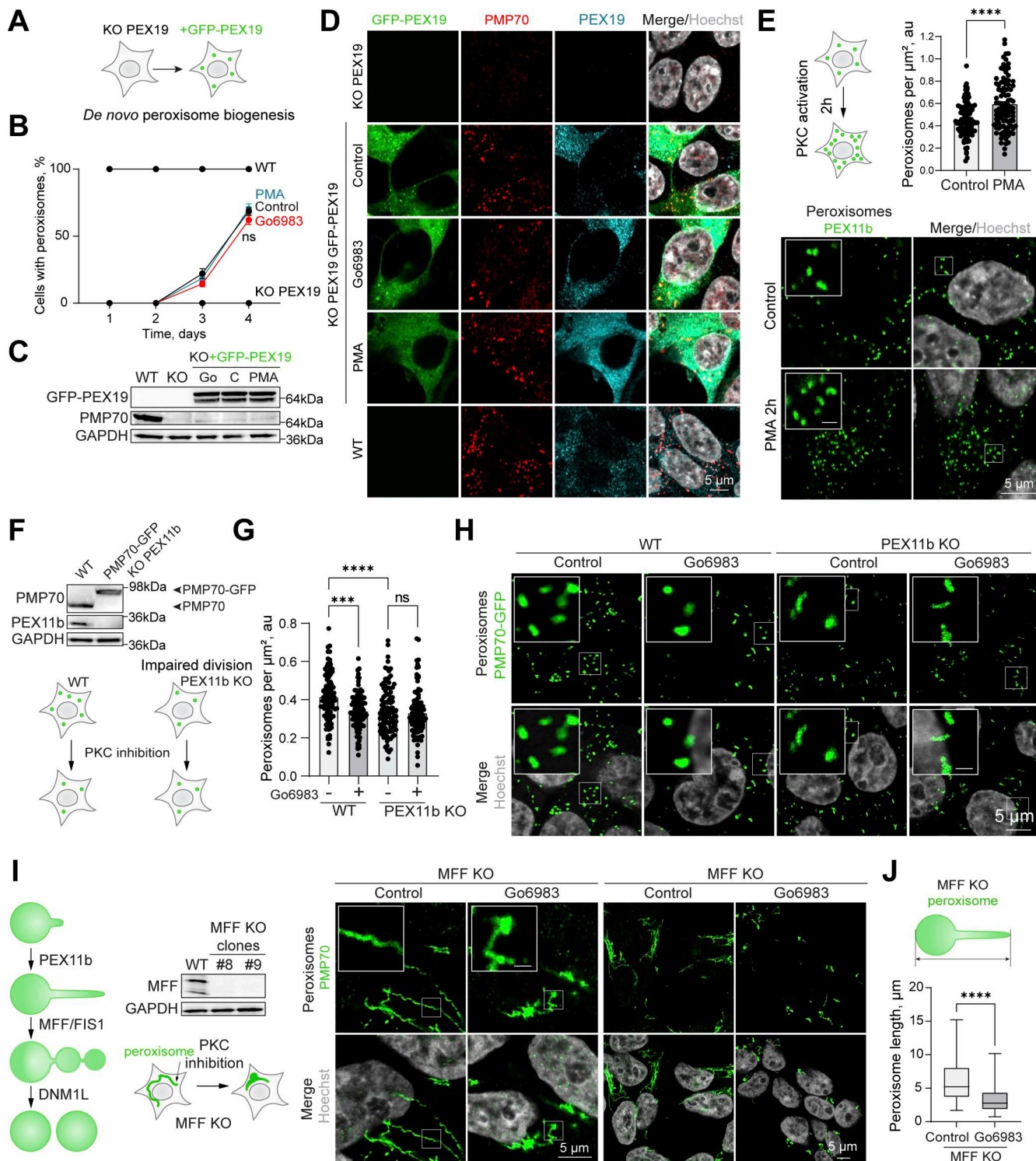

Figure 3. **PKC regulation depends on peroxisome division. (A)** Schematic of PEX19 complementation assay. **(B–D)** Confocal microscopy of the peroxisomes in HEK293T WT, PEX19 KO, and PEX19 KO cells overexpressing flag-GFP-PEX19 in control, Go6983 1 μM, and PMA 0.5 μM conditions. Cells were fixed and visualized on identified days. **(B)** Quantification shows the ratio of cells that restored peroxisomes among the transfected cells. Scale bar: 5 μm. **(C)** Western blot of PMP70 expression on day 4 of the complementation assay. **(E)** Confocal microscopy of peroxisomes in HEK293T CRISPR/Cas9 flag-GFP-PEX11b cells treated with 0.5 μM PMA for 2 h. Nuclei were stained with Hoechst (10 μg/ml). Representative images are shown; scale bar: 5 μm; inlet: 1 μm. Quantification shows the number of peroxisomes per square micron of the cytoplasm in the 2D confocal image, mean ± SEM, $N = 100$, ****$P < 0.0001$, Mann–Whitney test. **(F)** Western blot of PEX11b and PMP70 in the WT and CRISPR/Cas9 PMP70-GFP PEX11b KO HEK293T cells. Arrowheads indicate the size shift of the tagged PMP70-GFP. **(G and H)** Confocal microscopy of peroxisomes in HEK293T CRISPR/Cas9 PMP70-GFP WT and PEX11b KO cells grown in control or Go6983 5 μM conditions. Nuclei were stained with Hoechst (10 μg/ml). Representative images are shown; scale bar: 5 μm; inlet: 1 μm. Quantification shows the number of peroxisomes per square micron of the cytoplasm in the 2D confocal image, mean ± SEM, $N = 100$ cells pooled from 3 biological repeats, ***$P < 0.001$, ****$P <$

0.0001, Kruskal–Wallis test. **(I)** Confocal microscopy of peroxisomes in HEK293T CRISPR/Cas9 MFF KO cells grown in control or Go6983 5 µM conditions. Western blot shows MFF in WT and MFF KO cells. Nuclei were stained with Hoechst (10 µg/ml). Representative images are shown; scale bar: 5 µm; inlet: 1 µm. **(J)** Quantification of the length of the extended peroxisomes in HEK293T CRISPR/Cas9 MFF KO cells grown in control or Go6983 5 µM conditions, mean ± SEM, ****P < 0.0001, Mann–Whitney test. Source data are available for this figure: SourceData F3. ns, nonsignificant.

of peroxisomes—we used MFF KO. MFF KO has elongated peroxisomes (Fig. 3, I and J). Treating MFF KO with PKC inhibitors leads to a reduction in the length of peroxisomes, indicating that PKC regulates earlier steps preceding division (Fig. 3, I and J).

## PKC regulates peroxisome–ER contact sites through GSK3β inhibition

Based on the discoveries of the VAPB-ACDB5 peroxisome–ER contacts (Hua et al., 2017; Kors et al., 2022a) and a significant decrease in peroxisome abundance in the absence of VAPs (Korotkova et al., 2024), we hypothesized that PKC can modulate the contact. In addition to acting as a structural anchor, the VAPB-ACBD5 contact site regulates lipid transfer between the ER and peroxisomes. Since peroxisomes do not produce their own phospholipids, this activity may be essential for division-associated membrane growth (Costello et al., 2017). To independently confirm that peroxisome-ER contact sites are required for peroxisome formation, we used VAPB/VAPA knockout cells (Dong et al., 2016). The significant decrease in peroxisome number in the VAP KO cells was restored upon complementation with VAPB (Fig. 4 A and Fig. S4 H). Peroxisome abundance in VAP KO cells was not affected by PKC inhibition (Fig. 4 B) (Korotkova et al., 2024).

We then assessed peroxisome–ER interaction (Costello et al., 2017; Hua et al., 2017). Inhibition of PKC resulted in a decrease in the VAPB–ACBD5 interaction, as evidenced by the co-immunoprecipitation of VAPB (Fig. 4 C), and resulted in a significant increase in free-roaming peroxisomes, as evidenced by the lack of peroxisome–ER proximity (Fig. 4 D), and a significant increase in mobility (Fig. 4, E and F; and Videos 1, 2, 3, 4, 5, and 6). Overexpression of PKCδ resulted in increased interaction between VAPB and ACBD5 (Fig. 4 G). One obvious candidate for the mechanism of this regulation was GSK3β, due to its established role in the VAPB–ACBD5 negative regulation (Costello et al., 2017; Kors et al., 2022a). PKC negatively regulates GSK3β (Li et al., 2016; Moore et al., 2013), which, in turn, negatively regulates peroxisome contact sites with the ER (Kors et al., 2022a). We therefore hypothesized that PKC modulates contact site formation through GSK3β. We confirmed that PKC activation inhibits GSK3β, whereas inhibition of PKC leads to GSK3β activation, as assayed by measuring GSK3β S9 inhibitory phosphorylation (Fig. 4 H). Next, to confirm that PKC regulates peroxisome biogenesis through GSK3β inactivation, we constructed a GSK3β S9A constitutively active mutant (Stambolic and Woodgett, 1994), which suppressed peroxisome proliferation in response to PMA treatment (PKC activation) (Fig. 4 I). Finally, analysis of our hits from the small molecule screen revealed that the inhibitor of GSK3β, CHIR99021, increased peroxisome number in HEK293T cells (Fig. S1 G). Taken together, these data show that PKC can positively regulate peroxisome contact sites through inhibition of GSK3β, promoting

peroxisome–ER interaction through known mechanisms (Fig. 4 J) (Costello et al., 2017; Kors et al., 2022a). Whether increasing peroxisome contacts with the ER results in an increase in peroxisome number remains to be determined. It is possible as overexpression of VAPB increased the number of peroxisomes (Fig. 4 A). Although the loss of VAPA/B results in a significant decrease in peroxisome numbers, it is not clear whether it is due to the loss of peroxisome contacts with the ER or a combination of dysregulation in other ER membrane contact sites regulated by VAPA/B (Dong et al., 2016; Taskinen et al., 2023).

## PKC regulates peroxisome abundance during neuronal differentiation

Finally, we investigated the biological relevance of PKC-regulated peroxisome division. PKC activity varies between tissues, with the highest levels detected in the brain, including PKC-positive neuronal populations that control feeding, alcohol use, and fear response (Cai et al., 2014; Domi et al., 2021; Haubensak et al., 2010; Irani et al., 2010; Kikkawa et al., 1982). Further, peroxisome regulation is critical for brain development, as evidenced by the neurodegenerative conditions stemming from impaired peroxisome biogenesis (Berger et al., 2016). We, therefore, tracked peroxisome abundance in differentiating neuronal cells. We used human neuroblastoma SH-SY5Y cells that can be differentiated into neuron-like cells in 18 days (Fig. 5 A; and Fig. S6, A and B). PKC activity was significantly increased in differentiating cells (Fig. 5 B), as was peroxisome number. Several novel PKC isoforms were upregulated during neuronal differentiation, including PKCδ, θ and η (Fig. S6 C). We then differentiated SH-SY5Y neuronal cells in the presence of a PKC inhibitor that was added from day 10 to 18 of neuronal differentiation. Apart from PAX6, differentiation markers, including MAP2 and β-3 tubulin, were upregulated to a similar extent in control and PKC-inhibited samples (Fig. 5 C and Fig. S6 B), but peroxisome number was decreased significantly (Fig. 5, D–F and Fig. S6 D). Similarly, we detected an increase in PKC activity and peroxisome number while differentiating induced pluripotent stem cells to neuronal progenitor cells (Fig. 5, G–J and Fig. S6, E–G).

PKC regulation allows cells to rapidly induce peroxisomes, providing an on demand pool of functional peroxisomes (Fig. 5 K). The PKC-regulated pathway has broad implications for peroxisome physiology. For example, PKC and peroxisomes both regulate redox signaling and control levels of D-serine in the brain (Fransen and Lismont, 2019; Konishi et al., 1997; Mitchell et al., 2010; Morgan et al., 2017; Paul and de Belleroche, 2012; Sasabe et al., 2012; Steinberg, 2015). It would be of interest to explore the peroxisomal contribution to diseases associated with abnormal PKC signaling, such as cancer, cardiovascular diseases, diabetes, psoriasis, and neurodegenerative disorders (Alkon et al., 2007; Bowling et al., 1999; Garrido et al., 2002; Ishii

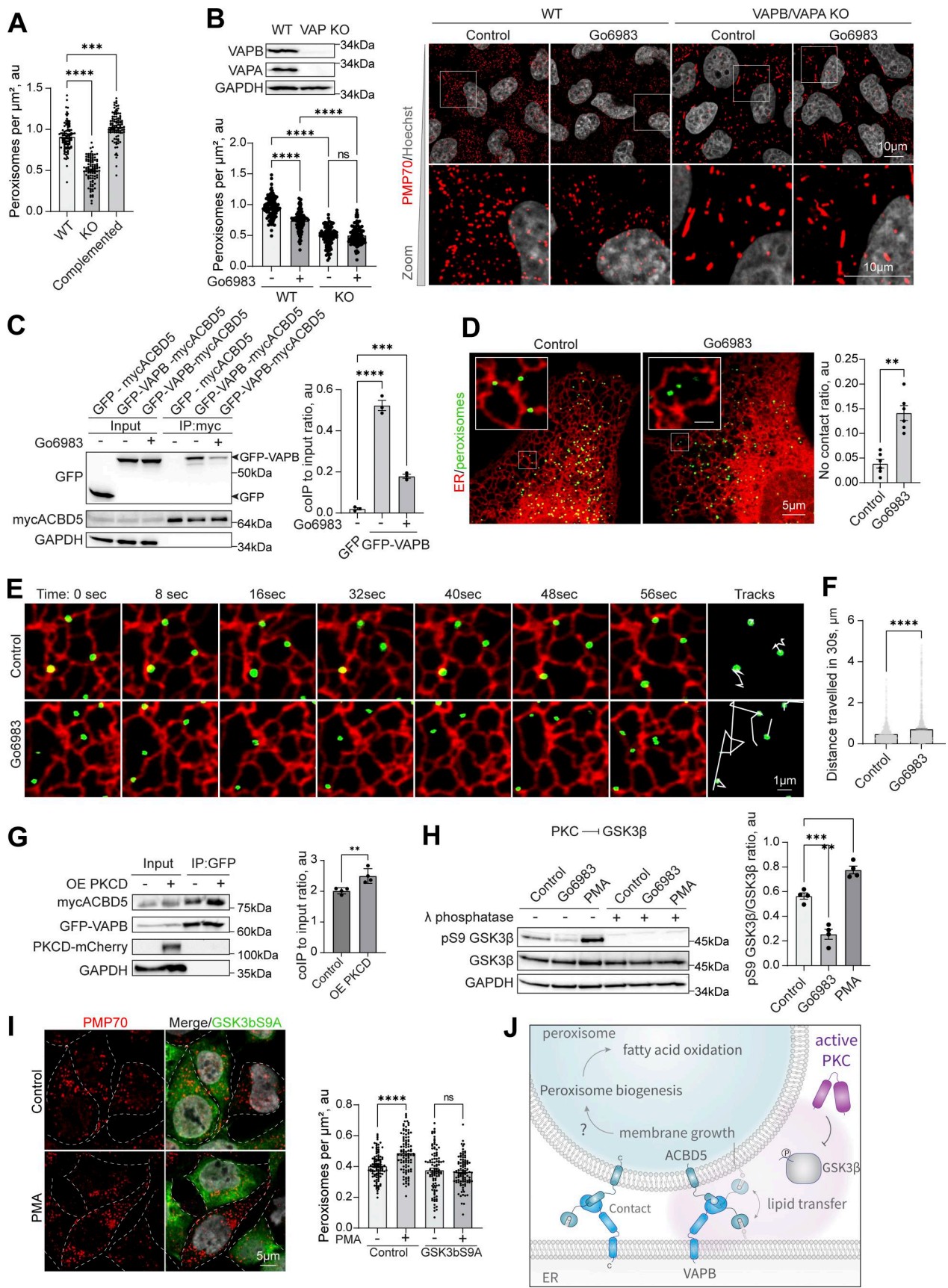

Figure 4. **PKC regulates peroxisome–ER VAP-dependent interaction through GSK3b inhibition. (A)** Quantification of confocal microscopy of peroxisomes stained with anti-PMP70 antibody in WT and VAPB/VAPA KO HeLa cells in control and GFP-VAPB overexpression conditions. Quantification shows the

number of peroxisomes per micron square, mean ± SEM, $N = 100$ cells pooled from three biological repeats, ***$P < 0.001$, ****$P < 0.0001$, Kruskal–Wallis test. **(B)** Confocal microscopy of peroxisomes stained with anti-PMP70 antibody in WT and VAPB/VAPA KO HeLa cells in control Go6983 (5 μM for 48 h). Quantification shows the number of peroxisomes per micron square, mean ± SEM, $N = 100$, ***$P < 0.001$, ****$P < 0.0001$, one-way ANOVA. Western blot confirming the KO is shown. **(C)** Co-immunoprecipitation of mycACBD5 in control and PKC inhibition conditions (Go6983 5 μM, 24 h); mycACBD5 and GFP or GFP-VAPB were overexpressed in HEK293T cells. Quantification shows the IP to input ratio, mean ± SEM, $N = 3$, ***$P < 0.001$, ****$P < 0.0001$, one-way ANOVA. **(D–F)** Live cell confocal microscopy of ER and peroxisomes in U2OS cells in control and Go6983 conditions. Quantification shows the ratio of peroxisomes that are not proximal or overlapping with the ER, mean ± SEM, $N = 6$ pooled from 1,000 peroxisomes in at least 30 cells per condition, **$P < 0.01$, and a distance traveled by a single peroxisome in 30 s, mean ± SEM, $N = 1,000$, ****$P < 0.0001$, unpaired $t$ test with Welch's correction. **(G)** Co-immunoprecipitation of GFP-VAPB in control and PKCD-mCherry overexpression conditions in HEK293T cells overexpressing mycACBD5 and GFP-VAPB. Quantification shows the IP to input ratio, mean ± SEM, $N = 4$, ***$P < 0.001$, ****$P < 0.0001$, unpaired $t$ test. **(H)** Western blot of GSK3β inhibitory S9 phosphorylation in PMA(0.5 μM for 2 h) and Go6983 (5 μM for 4 h) conditions. Quantification shows the ratio of phosphorylated pS9 GSK3β to non-phosphorylated GSK3β, mean ± SEM, $N = 4$, ***$P < 0.001$, **$P < 0.01$, one-way ANOVA. **(I)** Confocal microscopy of peroxisomes stained with anti-PMP70 antibody in HEK293T control or overexpression of GSK3β S9A mutant in control or PMA (0.5 μM for 4 h) conditions. **(I)** Quantification shows the number of peroxisomes per micron square, mean ± SEM, $N = 100$ pooled from three biological repeats, ****$P < 0.0001$, Welch ANOVA. **(J)** Model of PKC regulation of peroxisome– ER interaction. Source data are available for this figure: SourceData F4. OE, overexpressing; ns, nonsignificant; coIP, co-immunoprecipitation.

## Materials and methods

### Cell culture
Human fibroblasts (AFF11, a gift from van der Goot Lab, EPFL), U2OS (obtained from https://www.atcc.org/products/htb-96), HeLa (obtained from https://www.atcc.org/products/ccl-2), CHO (a gift from Kaganovich Lab), and HEK293T (obtained from https://www.atcc.org/products/crl-3216) cells were maintained in high-glucose DMEM supplemented with 10% FBS and 1% penicillin/streptomycin at 37°C/5% CO₂; SH-SY5Y (Kaganovich Lab, obtained from https://www.atcc.org/products/crl-2266) cells were maintained in high-glucose DMEM/F12 1:1 media supplemented with 10% FBS and 1% penicillin/streptomycin at 37°C/5% CO₂. Cells modified via CRISPR/Cas9 were maintained as above with the addition of puromycin (1 μg/ml, Sigma-Aldrich) or blasticidin (5 μg/ml, Sigma-Aldrich) during selection of the clonal populations. Neural progenitor cells (Kaganovich Lab, [Amen and Kaganovich, 2021], obtained from Cedar Sinai) were maintained in STEMdiff neural progenitor medium (STEMCELL Technologies). Neuronal differentiation of SH-SY5Y cells was done according to Shipley et al. (2016) with modifications. SH-SY5Y cells (initial seeding density 350,000 cells on 60-mm dish) were gradually FBS starved for 10 days before plating them on a Matrigel (Corning) extracellular matrix (seeding density (100,000 per a well of a 6-well plate); neuronal media was replaced every third day, and cells were collected on day 18. Neuronal differentiation of induced pluripotent stem cells derived neuronal progenitor cells (Amen and Kaganovich, 2021) was done using the STEMdiff neural differentiation kit (STEMCELL Technologies). The concentration of cells for plating was determined using cell counter (Countess II FL, Life Technologies) with the cell-counting chambers (Invitrogen). All the cell lines tested negatively for mycoplasma contamination.

### Antibodies
We used the following reagents to detect proteins: anti-GAPDH (sc-47724, mouse; Santa Cruz Biotechnology), anti-mCherry (34974, rabbit; Invitrogen), anti-PMP70 (SAB4200181, mouse; Sigma-Aldrich), anti-PEX19 (14713-1-AP, rabbit, Proteintech), anti-GFP (SAB4301138, rabbit; Sigma-Aldrich), anti-NBR1 (16004-1-AP, rabbit; Proteintech), anti-PEX11b (PA5-37011, rabbit; Thermo Fisher Scientific), anti-PKC (P5704, mouse; Sigma-Aldrich), anti-Flag (F1804, mouse; Sigma-Aldrich), anti-myc (9E10, mouse; Sigma-Aldrich), anti-MFF (17090-1-aP, rabbit; Proteintech), β-3 tubulin (D71G9, rabbit; Cell Signaling Technology), and anti-VAPB (14477-1-AP, rabbit; Proteintech). Primary antibodies were used at 1:1,000 dilution for western blots and 1:250 for immunofluorescence.

Secondary antibodies for immunofluorescence: anti-Rabbit IgG Cy3 conjugated (C2306; Sigma-Aldrich), anti-mouse IgG Cy3 conjugated (C2181; Sigma-Aldrich), anti-rabbit IgG Cy5 conjugated (A10523; Invitrogen), anti-mouse IgG H&L (Alexa Fluor 488) (Abcam). Secondary antibodies were used at 1:10,000 dilution for western blots and 1:1,000 for immunofluorescence.

### SDS-PAGE and immunoblotting
Cell lysates were obtained using RIPA buffer (NaCl 0.15 M, Np-40 1%, sodium deoxycholate 0.5%, SDS 0.1%, Tris-HCl, pH7.4, 50 mM, and protease inhibitor cocktail [53875100; Pierce]) on ice. Protein concentration was estimated using bicinchoninic acid method (23228; Pierce). 10 micrograms was loaded on the home-cast Tris-glycine 10 or 12% gels. After SDS-PAGE, proteins were transferred onto nitrocellulose membrane (10600015; Amersham Protran) using wet transfer chamber at 110 V for 1 h (Bio-Rad). Western blots were visualized using BioRad Gel Doc Imager with exposure time of 2–5 min, and images were processed using Image Lab 6.1 software (BioRad). Quantification of band intensity was done using FiJi software, and band intensities were quantified from three biological repeats for each experiment and normalized to GAPDH band intensity obtained with the same samples.

### Chemicals
Hoechst (Sigma-Aldrich), fatty acid–free BSA (Pan-Biotech), PMSF (Sigma-Aldrich), Go6983 (Cayman Chemicals), behenic acid (VLCFA, Sigma-Aldrich), D-serine (Sigma-Aldrich), palmitic acid (LCFA, Sigma-Aldrich), phorbol 12-myristate 13-acetate (Cayman Chemicals), oleic acid (Sigma-Aldrich), brain-derived neurotrophic factor (STEMCELL Technologies), dibutyryl-cAMP (Santa Cruz Biotechnology), Neuropan 27 Supplement 50× (Pan Biotech), FBS (S100082425; Pan-Biotech),

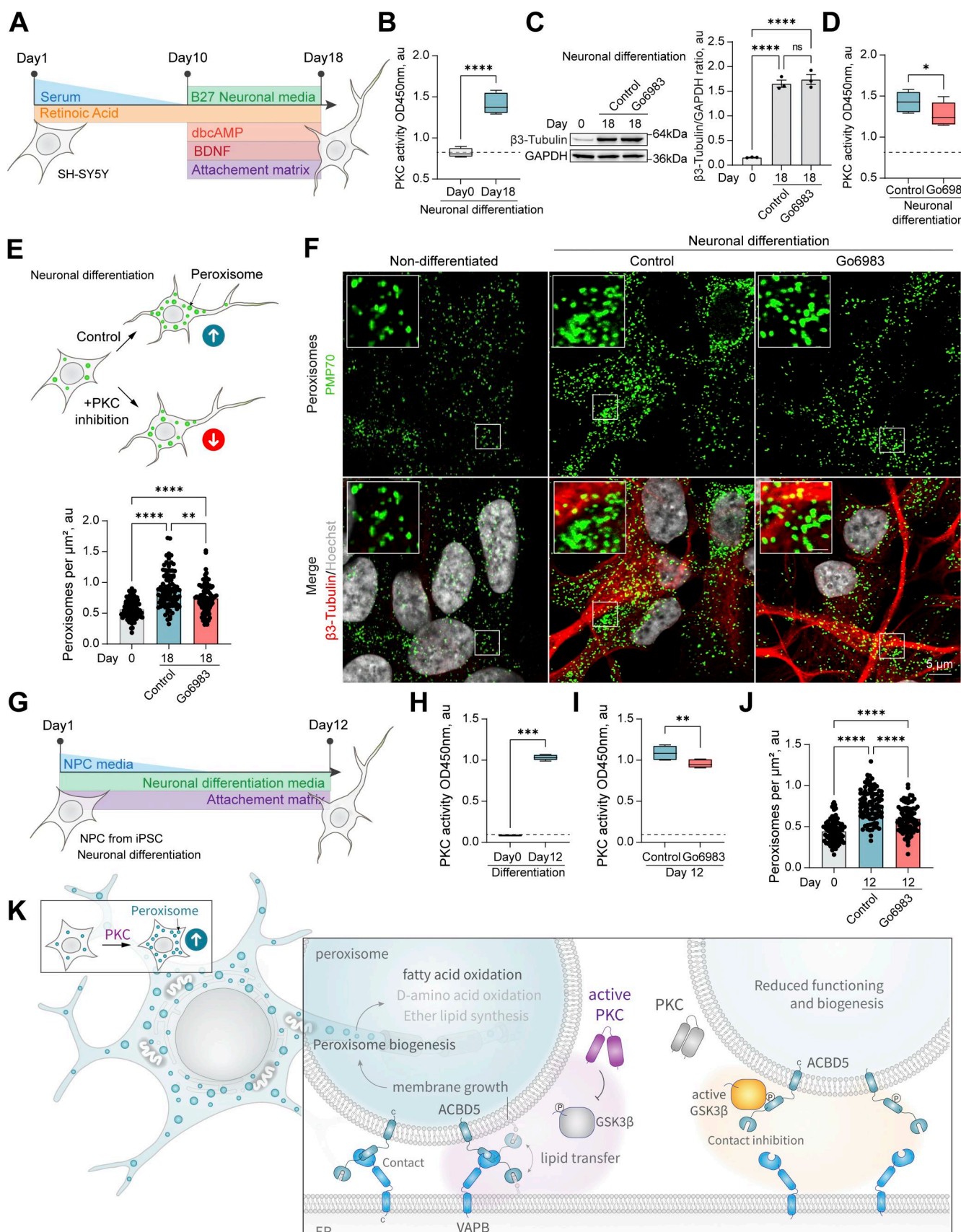

Figure 5. **PKC regulates peroxisome abundance in neurons. (A)** Schematic of the SH-SY5Y neuronal differentiation. **(B)** PKC activity in the non-differentiated and 18-day differentiated SH-SY5Y cells. Quantification shows PKC activity, mean ± SEM, *N* = 10, ****P < 0.0001, Mann–Whitney test.

**(C)** Western blot of β-3 tubulin in the non-differentiated and 18-day differentiated SH-SY5Y cells in control and Go6983 1 μM conditions. Go6983 was added on days 10–18 of the differentiation protocol. Quantification shows the ratio of and β-3 tubulin to GAPDH, mean ± SEM, N = 3, ****P < 0.0001, one-way ANOVA. **(D)** PKC activity in the 18-day differentiated SH-SY5Y in control and Go6983 1 μM conditions. Go6983 was added on days 10–18 of the differentiation protocol. Quantification shows PKC activity, mean ± SEM, N = 10, *P < 0.05, Mann–Whitney test. **(E and F)** Confocal microscopy of peroxisomes in the non-differentiated and 18-day differentiated SH-SY5Y cells in control and Go6983 1 μM conditions. Go6983 was added on days 10–18 of the differentiation protocol. Peroxisomes were visualized with the PMP70 antibody, nuclei were stained with Hoechst (10 μg/ml), and neuronal differentiation was visualized with β-3 tubulin antibody. Representative images are shown. Scale bar: 5 μm; inlet: 1 μm. See zoomed out images in Fig. S6 D. **(F)** Quantification shows the number of peroxisomes per square micron of the cytoplasm in the 2D confocal image, mean ± SEM, N = 102 cells pooled from three biological repeats, ****P < 0.0001, Kruskal–Wallis test. **(G)** Schematic of the neuronal progenitor cells (NPCs) neuronal differentiation. **(H)** PKC activity in the non-differentiated and 12-day differentiated NPCs. Quantification shows PKC activity, mean ± SEM, N = 8, ****P < 0.0001, Mann–Whitney test. **(I)** PKC activity in the 12-day differentiated NPCs in control and Go6983 1 μM conditions. Go6983 was added on days 1–12 of the differentiation protocol. Quantification shows PKC activity, mean ± SEM, N = 8, **P < 0.01, Mann–Whitney test. **(J)** Quantification of the number of peroxisomes per square micron of the cytoplasm in the 2D confocal image in the non-differentiated and 12-day differentiated NPCs in control and Go6983 1 μM conditions. Go6983 was added on days 1–12 of the differentiation protocol, mean ± SEM, N = 90 cells pooled from three biological repeats, ****P < 0.0001, one-way ANOVA. **(K)** Model of PKC regulation of peroxisome abundance. Source data are available for this figure: SourceData F5. ns, nonsignificant.

## CRISPR/Cas9

Knockout and endogenously tagged cell lines were constructed using CRISPR/Cas9 protocol and plasmids described in Ran et al. (2013). Knockout cell lines were verified by western blotting and immunofluorescence. Genomic DNA was sequenced to verify the disrupted region in knockout or the fidelity of endogenous tagging. A functional assay for the knockout verification was performed where applicable (reduced viability of PEX19 on VLCFA). Endogenous tagging was performed by fusing the tagging construct (linker-GFP-polyA-Blasticidin for C-terminal tagging of PMP70 and flag-GFP- for N-terminal tagging of PEX11b) to the region upstream of the stop codon (~500 bp) and downstream of the stop codon (~500 bp) for C-terminal tagging and to the region upstream of the start codon and downstream of the start codon for N-terminal tagging (Amen and Kaganovich, 2020a). PCR product containing homologous regions flanking the tagging construct was co-transfected with px330-gRNA corresponding construct for C-terminal tagging and px459-gRNA construct for N-terminal tagging. Endogenous tagging was verified by western blotting, immunofluorescence staining, and genomic DNA sequencing. CRISPR specificity was profiled using the Digenome-Seq web tool (https://www.rgenome.net/cas-offinder/) (Bae et al., 2014). Off-targets were not found. The following target sequences are used to modify genomic DNA: endogenous tagging of PMP70 on the C terminus—5′-GTT GAGTTTGGCTCTTAGAGAAATC-3′, endogenous tagging of PEX11b on the N terminus—5′-CGCGGAGCCTGGGCTGCGGC TGTCA-3′, knockout of PEX19—5′-GGAGGTAGCAAGATGGCC GCCGCTG-3′, knockout of PEX11b—5′-CGCGGAGCCTGGGCT GCGGCTGTCA-3′, knockout of MFF—5′-TACTGAAGGCAT-TAGTCAGCGAATG-3′, and knockout of PKCD—5′-GCACAGC CCCACTGCAGGCCCCACCA-3′.

## Plasmid construction

All plasmids were constructed using *Escherichia coli* strain DH5α. Plasmids used in this study are summarized in the Table 1. We used px459 and px330 plasmids to clone CRISPR/Cas9 constructs for gene knockout and endogenous tagging. pSpCas9(BB)-

2A-Puro (PX459) V2.0 was a gift from Feng Zhang (plasmid #62988; Addgene; https://n2t.net/addgene:62988; RRID:Add-gene_62988) (Ran et al., 2013). pX330-U6-Chimeric_BB-CBh-hSpCas9 was a gift from Feng Zhang (The Broad Institute,

Table 1.   **Plasmids used in this study**

| Plasmid name | Source |
|---|---|
| pcDNA3.1 mCherry*4skl | (Amen and Kaganovich, 2021) |
| pcDNA3.1 GSK3beta | Addgene 14753 |
| pcDNA3.1 PPARA | Addgene 169019 |
| pcDNA3.1 PEX3hismyc | This study, human PEX3 under CMV promoter tagged with HIS6MYC tags |
| pcDNA3.1 flagPEX19 | This study, human PEX19 under CMV promoter tagged with a flag tag |
| Px459-PEX19 KO | (Korotkova et al., 2024) |
| Px330-PMP70-C | This study, for CRISPR/Cas9 endogenous tagging of PMP70 on its C terminus |
| Px459-PEX11b-N | This study for CRISPR/Cas9 KO and endogenous tagging of PEX11b on its N terminus |
| pCtag-PMP70-GFP-Blasti | This study, cassette for CRISPR/Cas9 tagging of PMP70 with GFP, with blasticidin selection after polyA tail |
| pNtag-PEX11b-flag-GFP | This study, cassette to endogenous tagging of PEX11b with GFP on its N terminus |
| pcDNA3.1- PKCα-mCherry | This study, PKCα is from Addgene 21232 under CMV promoter |
| pcDNA3.1 PKCδ -mCherry | This study, PKCδ is from Addgene 20603 under CMV promoter |
| pcDNA3.1- PKCζ -mCherry | This study, PKCζ is from Addgene 24609, under CMV promoter |
| pcDNA3.1 hismyc-PEX11b | This study, human PEX11b tagged with HIS6MYC tags under CMV promoter |
| pEGFP-VAPB | (Anwar et al., 2022) |
| pcDNA3.1 hismyc-ACBD5 | This study, human ACBD5 under CMV promoter tagged with HIS6MYC |
| pcDNA3.1 GFP-SKL | This study, GFP fused to SKL on the C terminus under CMV promoter |
| pcDNA3.1 ER-mCherry | (Amen and Kaganovich, 2021) |

Cambridge, Massachusetts, MA, USA; plasmid #42230; Addgene; http://n2t.net/addgene:42230; RRID:Addgene_42230) (Cong et al., 2013). pcDNA4-PKCZeta WT His tag was a gift from Jeff Wrana (Mount Sinai Hospital, Toronto, Canada; plasmid #24609; Addgene; https://n2t.net/addgene:24609; RRID:Addgene_24609) (Wang et al., 2003). PKC alpha WT was a gift from Bernard Weinstein (Columbia University, New York, NY, USA; plasmid #21232; Addgene; https://n2t.net/addgene:21232; RRID:Addgene_21232) (Soh and Weinstein, 2003). pWZL Neo Myr Flag PRKCD was a gift from William Hahn & Jean Zhao (Dana-Farber Cancer Institute, Boston, MA, USA; plasmid #20603; Addgene; https://n2t.net/addgene:20603; RRID:Addgene_20603) (Boehm et al., 2007). HA GSK3 beta WT pcDNA3 was a gift from Jim Woodgett (National Institutes of Health, Bethesda, MD, USA; plasmid #14753; Addgene; https://n2t.net/addgene:14753; RRID:Addgene_14753) (He et al., 1995). pCDNA3.1-PPARA was a gift from Claes Wadelius (Uppsala University, Uppsala, Sweden; plasmid #169019; Addgene; https://n2t.net/addgene:169019; RRID:Addgene_169019) (Pan et al., 2021). Human PEX3, PEX19, DAO, and ACBD5 were amplified from the lentiviral plasmid collection. Site-directed mutagenesis was verified by sequencing. Site-directed mutagenesis was performed to obtain GSK3bS9A, using the following primers: SDMGSK3bS9AF—5′-GTCAGGGCGGCCCAGAACCACCGCCTTTGCGGAGAGCTGCAAGCCG-3′ and SDMGSK3bS9AR—5′-CGGCTTGCAGCTCTCCGCAAAGGCGGTGGTTCTGGGCCGCCCTGAC-3′. Overexpression of proteins was done using pcDNA3.1 plasmid with a constitutive CMV promoter.

## PKC activity

Protein kinase activity was determined by the PKC Kinase Activity Assay Kit (ab139437; Abcam) according to the manufacturer's instructions. Protein concentration in lysates was quantified using a bicinchoninic acid assay kit (Interchim).

## Peroxisome abundance measurement

We used peroxisome density as a readout for peroxisome abundance in this study. To calculate the number of peroxisomes, cells were visualized by confocal microscopy. Images were captured using the same parameters between conditions within the experiment. Images were analyzed by Fiji (ImageJ) software, peroxisome numbers were quantified as maxima of intensity in the cytoplasmic space, and the number was divided by the size of the areas (square micron), which results in a peroxisome per square micron cytoplasmic ratio. Peroxisomes were defined using the Find Maxima function in FiJi after assigning the threshold based on the control sample (the same threshold was then used for all images). The cytoplasmic space was defined using a threshold mask that excludes the nuclei (stained with Hoechst) and background using the ImageJ Threshold function. The resultant area was divided into N parts equal to the number of cells (nuclei number). At least 100 areas were quantified per condition per experiment. 2D areas with a nucleus in focus were chosen for all of the experiments because 3D projections often create overlaps between peroxisomal compartments.

## Knockdown using siRNA

siRNA was obtained from Qiagen: siNBR1 (S100082425; Qiagen FlexiTube), siNIX (S104023152; Qiagen FlexiTube), siControl (S104023152; Qiagen FlexiTube). Three microliters of siRNA was mixed with 9 µl of RNAiMax Lipofectamine (13778030; Thermo Fisher Scientific) in transfection FBS-free media and incubated for 15 min. 60% confluent cells were transfected with the mix.

## RNA preparation and real-time PCR

Total mRNA was extracted from cells using the RNeasy Mini Kit (Qiagen). cDNA synthesis was performed using the iScript cDNA synthesis kit (BioRad). Real-time PCR was performed using QuantStudio6 (Thermo Fisher Scientific). mRNA levels were quantified using QuantStudio6 software. Experiments were repeated three times with two technical repeats, and fold difference in expression was calculated by the ΔΔCt method using GAPDH as a housekeeping gene (Livak and Schmittgen, 2001; Table 2).

## Co-immunoprecipitation

Harvested cells were lysed with CHAPS (1%) in 50 mM Tris-HCl, pH7.4 and 150 mM NaCl with a complete protease inhibitor cocktail (Pierce) for 20 min at 4°C with rotation (1%CHAPS in PBS for ACBD5 immunoprecipitation). After 1 min 10,000 $g$ centrifugation, the supernatant was incubated with protein G Sepharose (Cytiva 17-0618-01; GE Healthcare) for 30 min at 4°C with rotation. After 1 min 10,000 g centrifugation, 10% of the supernatant was collected as an input, and the resulting supernatant was incubated with 25 µl of Myc-Trap Agarose beads (Chromotek) overnight at 4°C with rotation. After the incubation, the beads were washed four times for 5 min with the lysis buffer at 4°C with rotation. Before loading on the gel, the beads and input were boiled with a sample buffer. Equal amounts of input and eluted bead sample were loaded on the SDS-PAGE.

## Immunofluorescence

Cells were grown on glass-bottom plates or glass slides, fixed using 4% paraformaldehyde in PBS for 10 min, washed with PBS, permeabilized with 0.5% Triton X-100, and then blocked overnight in 5% BSA in PBS prior to antibody staining.

## Radioactive measurement of beta fatty acid oxidation using 13,14-3H docosanoic acid (C22:0)

Cells were grown in 12-well plates with an initial plating density of 100,000 cells per well for 2 days. Cold C22:0 (4 µM) and 13,14-3H C22:0 (1 µCi, Anawa) were mixed with the media and added to the cells for 20 h. Media was collected and processed according to Ma et al. (2020) Wanders et al. (1995) with minor modifications. Radioactivity was quantified in the media and water fractions transferred to the scintillation vials after 3 days using a scintillation counter. Cold samples and cell-free samples were used as controls and for background subtraction. Peroxisome-deficient cells were used as a control to distinguish between peroxisomal and mitochondrial fatty acid oxidation.

**Table 2. Primers for rtPCR**

| | |
|---|---|
| NT5F | 5′-GTGGGAATCGTTGGATACACTTCC-3′ |
| NT5R | 5′-CAAAACCCGAATGTCCCAGTGC-3′ |
| NESF | 5′-GGGCCTACAGAGCCAGATCG-3′ |
| NESR | 5′-CTGAAAGCTGAGGGAAGTCTTGG-3′ |
| NeuNF | 5′-CACACCAGCACAGACCCACC-3′ |
| NeuNR | 5′-GATGTTGGAGACGTGTAGCCGC-3′ |
| MAP2F | 5′-GGGAGGATGAAGAGGGTGCC-3′ |
| MAP2R | 5′-CAGGACTGCTACAGCCTCAGC-3′ |
| PAX6F | 5′-GGAATCAGAGAAGACAGGCCAGC-3′ |
| PAX6R | 5′-CCATGGTGAAGCTGGGCATAG-3′ |
| GAPDHF | 5′-GGGGAAGGTGAAGGTCGGAGTC-3′ |
| GAPDHR | 5′-GTGCCATGGAATTTGCCATGGG-3′ |
| PEX5F | 5′-CCTGCAGGACCAGAATGCAC-3′ |
| PEX5R | 5′-CTGAGTTACATCCACAGCATCTCC-3′ |
| PEX19F | 5′-CAGCACCCCCTTCTACCACC-3′ |
| PEX19R | 5′-GGAACTGCTCCACCAGGTG-3′ |
| PEX7F | 5′-ACAACGAACATGTCCTCATCACC-3′ |
| PEX7R | 5′-GAGCCAGACACCACAAGCTG-3′ |
| CATF | 5′-GCATGCTAAAGGAGCAGGGG-3′ |
| CATR | 5′-GTCAGCTGAACCCGATTCTCC-3′ |
| BAATF | 5′-CCTCCTTGGCCTTGGCTTACC-3′ |
| BAATR | 5′-CTCCTTGACATACAGAGACTACCCC-3′ |
| ACOX2F | 5′-GTGGACATGGCAAGAACAGCC-3′ |
| ACOX2R | 5′-ACTGGAACAGGCGTTCGTAGAC-3′ |
| DBPF | 5′-CAAAGTTGGGTCTTCTGGGCC-3′ |
| DBPR | 5′-GCCACATACTCTGGCTTCAGGG-3′ |
| SCPxF | 5′-GTATGGCCTGCAATCCAAAGC-3′ |
| SCPxR | 5′-GTGTCAGGCCAGATTTCTCATAGCA-3′ |
| AMACRF | 5′-TTTGCTGGTGGTGGCCTTATG-3′ |
| AMACRR | 5′-TCGAGGTGCTTCCCACAGAC-3′ |
| ACAA1F | 5′-GAATCTGAGGCCGGAACAGC-3′ |
| ACAA1R | 5′-ACCTGCTATGCTGGCCACTG-3′ |
| ACOX1F | 5′-GTCAGGTCCTTCCTTGTGGGAG-3′ |
| ACOX1R | 5′-GTATGCGCCCACAAACTGGAAG-3′ |
| PHYHF | 5′-CGAAGGTCCAGGATTTCCAGG-3′ |
| PHYHR | 5′-TAGTGCAGGTCCTGGTGCAG-3′ |
| DAOF | 5′-CTGACCCCAACAACCCACAG-3′ |
| DAOR | 5′-CCGAAATCCCAGAACTGTGTCC-3′ |
| PEX3F | 5′-CAGAAATCTCGTTGAGCAGCATAAGTC-3′ |
| PEX3R | 5′-CTAAAATCTGGGCTTTCCAACATGTCTC-3′ |
| PRKCAF | 5′-CCTGCTGGCAACAAAGTCATC-3′ |
| PRKCAR | 5′-CTTCCTGTCGGCAAGCATC-3′ |
| PRKCBF | 5′-GACGTGGAGTGCACTATGGTG-3′ |
| PRKCBR | 5′-GCCGACTTGCTGGATGTGA-3′ |
| PRKCGF | 5′-GACGACGATGTGGACTGCAC-3′ |
| PRKCGR | 5′-TCTCCCCCGGTGACGTACTC-3′ |

**Table 2. Primers for rtPCR (Continued)**

| | |
|---|---|
| PRKCDF | 5′-TGTACGAGATGCTCATTGGCC-3′ |
| PRKCDR | 5′-CACTCCCAGCCTCTTGGTTG-3′ |
| PRKCEF | 5′-CGGTTCTATGCTGCAGAGGTC-3′ |
| PRKCER | 5′-ATCCCTTCCTTGCACATCCC-3′ |
| PRKCHF | 5′-AGCTCGCTTCTATGCTGCAG-3′ |
| PRKCHR | 5′-TGCAAATCCCCTCCTTGCAC-3′ |
| PRKCQF | 5′-GGGACACCTGACTACATCGC-3′ |
| PRKCQR | 5′-TGGAGTGGAAGAGCTCCTCC-3′ |
| PRKCIF | 5′-GCACAGACCGAATATGGGGAC-3′ |
| PRKCIR | 5′-GTGCATGGTCAGAATGCATGG-3′ |
| PRKCZF | 5′-GTTCTCCTGGTGCGGTTGAAG-3′ |

rtPCR, real-time PCR.

## Microscopy

For live-cell imaging, 4-well microscope glass-bottom plates (IBIDI) or Cellview cell culture dishes (Greiner Bio One) were used. Alternatively, cells were grown on glass slides (Marienfeld). Confocal images and movies were acquired using an SP8 (Leica) confocal microscope equipped with a temperature and $CO_2$ incubator, using a 60× PlanApo VC oil objective NA 1.40. We used 406-, 488-, 561-, and 640-nm lasers. We used ProLong mounting media (Thermo Fischer Scientific). Peroxisome tracking in the movies was done using the TrackMate Fiji plugin (Tinevez et al., 2017). Image processing was performed using Fiji (ImageJ) software, using cropping tool and LUT adjustments.

## Quantification and statistical analysis

Three or more biological repeats were performed to obtain the data. P values were calculated by two-tailed Student's *t* test or one-way ANOVA for samples following normal distribution determined by the Shapiro–Wilks test. The equality of variances was verified by the Brown–Forsythe or F test. Mann–Whitney (2 groups) or Kruskal–Wallis (multiple groups) tests were used for samples that did not follow a normal distribution. The sample sizes were not predetermined. Refer to the Table S2 for the details on statistical analysis.

## Online supplemental material

Fig. S1 shows the supporting data for the screen Fig. 1. Fig. S2 shows the screen confocal microscopy images. Fig. S3 shows the screen confocal microscopy images. Fig. S4 shows the supporting data for Figs. 3 and 4. Fig. S5 shows the co-immunoprecipitation of peroxisome division factors. Fig. S6 shows the supporting data for neuronal differentiation Fig. 5. Videos 1, 2, 3, 4, 5, and 6 show peroxisomes and ER in live human U2OS cells in control and PKC inhibition conditions (three independent movies for each condition). Table S1 shows the kinase inhibitor screen and contains details on the small molecules used for the screening and the peroxisome density for each treatment with highlighted significant differences. Table S2 shows statistical analysis and contains information on statistical tests used in

this manuscript, including repeats, sample size, normality tests, and parameter values.

### Data availability
The generated data are available in the published article and its online supplemental material.

## Acknowledgments
We thank Prof. Gisou van der Goot and Prof. Giovanni D'Angelo and members of their labs for their support. We thank the Gene Expression Core Facility (EPFL), Dr. Adrien Schmid, and the Bioimaging and Optics Core Facility (PTBIOP, EPFL). We thank the Imaging and Microscopy Centre Facility, University of Southampton. We thank Prof. Ronald J.A. Wanders for the advice on radioactive measurements.

T. Amen was funded by the HFSP Long-term Fellowship (LT000559/2021-L), EPFL Faculty, the University of Southampton, and the Wessex Medical Research Innovation Grant. A. Borisyuk was funded by the HFSPO Scientists for Scientists Initiative.

Author contributions: A. Borisyuk: formal analysis and investigation. C.M. Howman: investigation. S. Pattabiraman: formal analysis and investigation. D. Kaganovich: conceptualization, funding acquisition, methodology, project administration, supervision, and writing—review and editing. T. Amen: conceptualization, data curation, formal analysis, funding acquisition, investigation, methodology, project administration, resources, software, supervision, validation, visualization, and writing—original draft, review, and editing.

Disclosures: The authors declare no competing interests exist.

Submitted: 8 May 2025

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

# Supplemental material

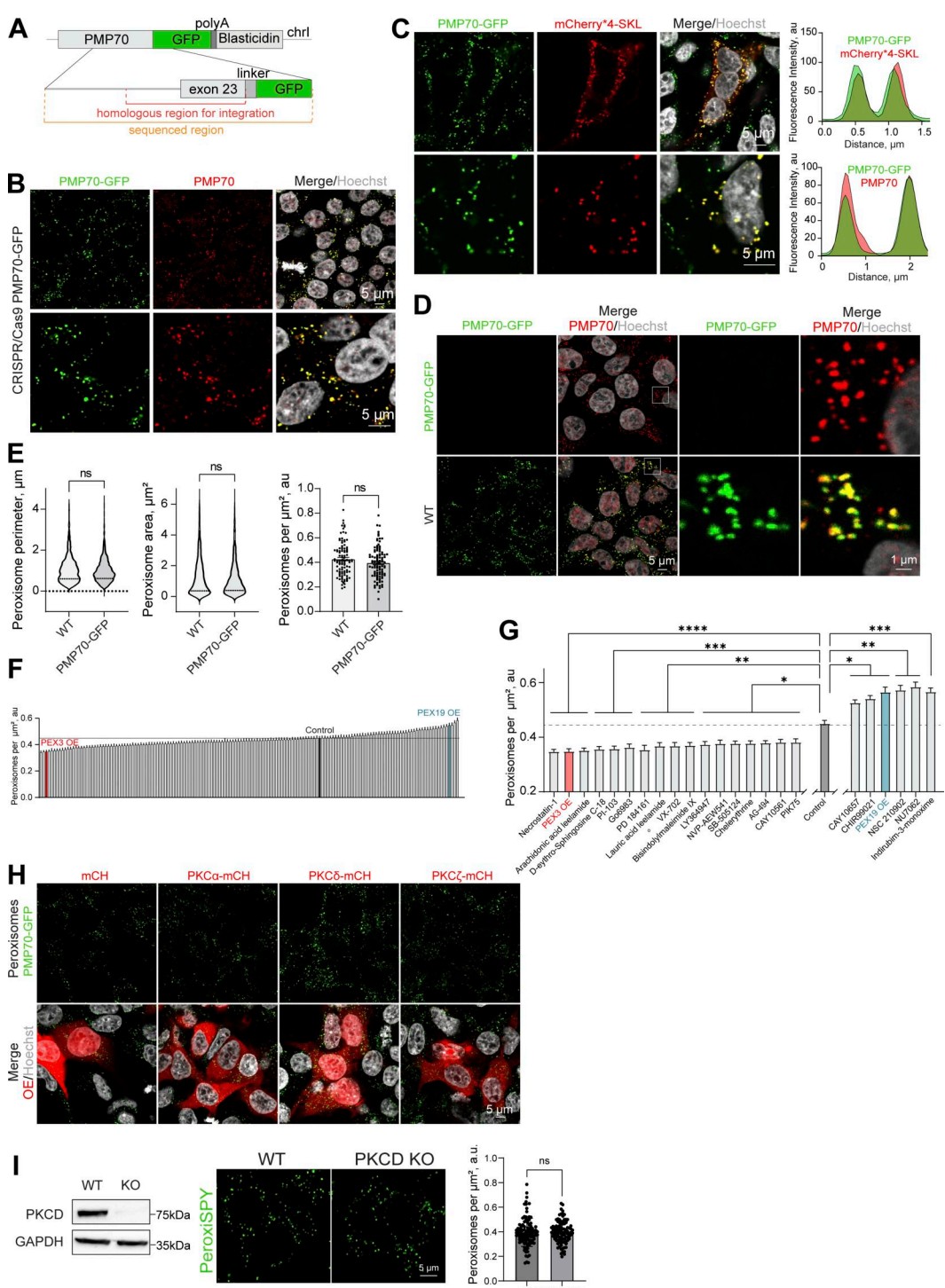

Figure S1.    **Signaling regulators of peroxisome abundance. (A)** Schematic of the genomic DNA region corresponding to the end of the human PMP70 open reading frame endogenously tagged with -GFP-polyA-Blasticidin. **(B and C)** Confocal microscopy of the peroxisomal import marker mCherry*4-SKL expressed in HEK293T PMP70-GFP, and (B) confirmation of the PMP70-GFP peroxisomal localization. Fluorescence intensity profiles through single peroxisomes are shown. Scale bar: 5 μm. **(D and E)** Comparison of WT and PMP70-GFP–tagged HEK293T. Confocal images and quantification of perimeter (N = 1,000), area (N = 1,000), and density of peroxisomes (N = 100) are shown, mean ± SEM, Mann–Whitney. **(F and G)** Quantification of the number of peroxisomes per square micron of the cytoplasm in the HEK293T cells treated with 1 μM of the indicated small molecule, mean ± SEM, *P < 0.05, **P < 0.01, ***P < 0.001, ****P < 0.0001, Kruskal–Wallis test. **(G)** Significant screen hits are plotted, and (F) complete screen. Refer to Figs. S2 and S3 for the confocal images, N = 100 for each molecule. **(H)** Confocal microscopy of HEK293T CRISPR/Cas9 PMP70-GFP cells overexpressing (OE) PKCα-mCherry, PKCδ-mCherry, or PKCζ-mCherry. Scale bar: 5 μm. See zoomed in images in Fig. 2 B. **(I)** Western blot of PKCδ in WT and CRISPR/Cas9 KO HEK293T cells. Confocal microscopy (PeroxiSPY555 0.5 μM 10 min), and quantification shows the number of peroxisomes per square micron of the cytoplasm in the 2D confocal image, mean ± SEM, N = 100 pooled from three biological repeats, ns—nonsignificant, Mann–Whitney. Representative images are shown. Scale bar: 5 μm; inlet: 1 μm. Source data are available for this figure: SourceData FS1.

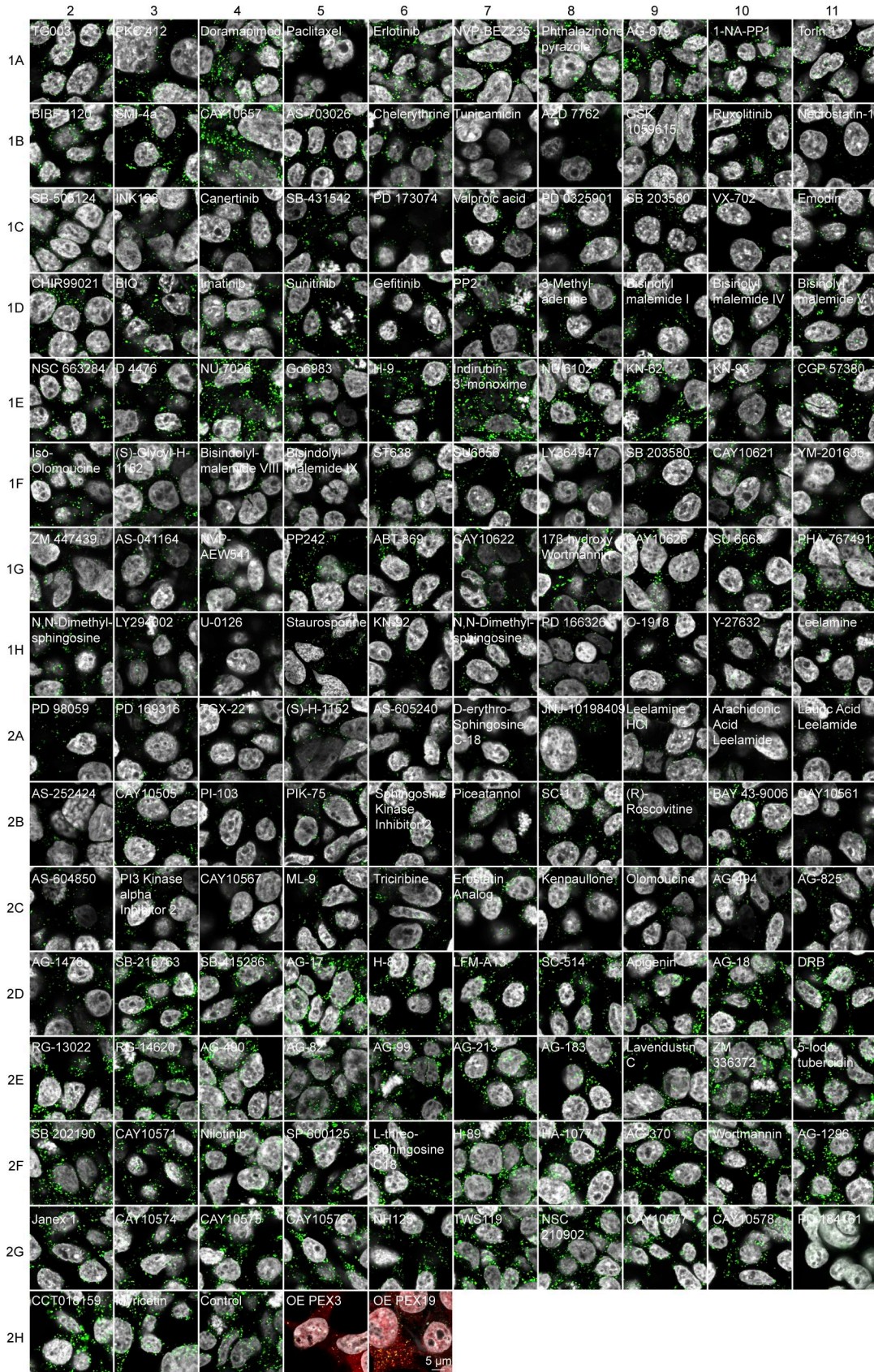

Figure S2.  **Kinase Inhibitor Screen.** Confocal microscopy of the HEK293T CRISPR/Cas9 PMP70-GFP cells incubated for 2 days with 1 µM of the indicated small molecule. Note that endogenous levels of PMP70 vary between treatments, as do the numbers of peroxisomes. Scale bar: 5 µm. See Fig. 1 C for the separate channels of control images.

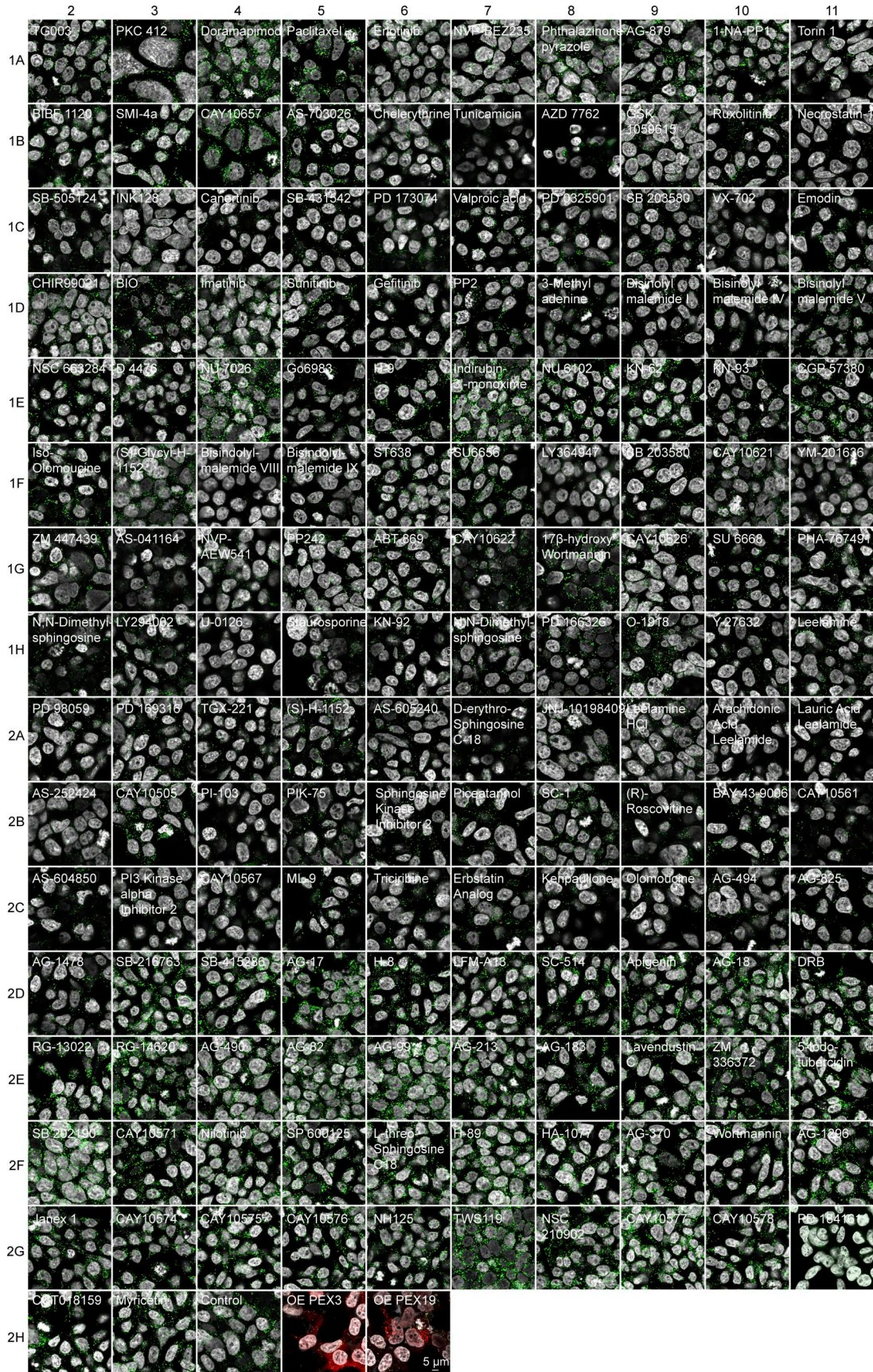

Figure S3. **Kinase Inhibitor Screen.** Confocal microscopy of the HEK293T CRISPR/Cas9 PMP70-GFP cells incubated for 2 days with 1 µM of the indicated small molecule. Note that endogenous levels of PMP70 vary between treatments, as do the numbers of peroxisomes. Scale bar: 5 µm.

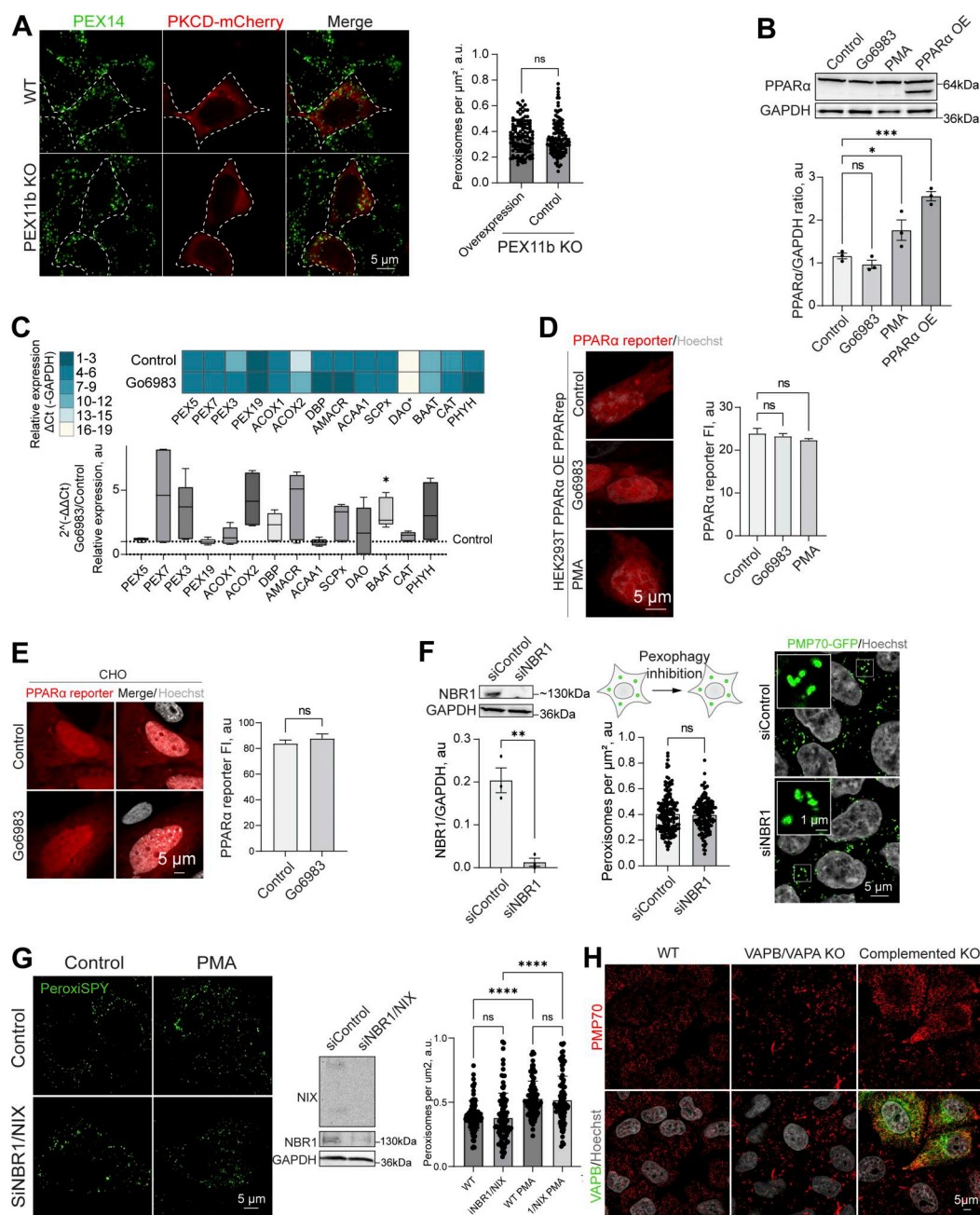

Figure S4. **PPAR response and pexophagy during PKC regulation. (A)** Confocal microscopy of peroxisomes in WT and PEX11b KO HEK293T overexpressing PKCδ-mCherry overexpressing (OE). Peroxisomes were stained with anti-PEX14 antibody. Scale bar: 10 µm. Quantification shows the number of peroxisomes per square micron of the cytoplasm in PEX11b KO cells with and without overexpression of PKCδ-mCherry, mean ± SEM, N = 100 pooled from three biological repeats, ns—nonsignificant, Mann–Whitney. **(B)** Western blot of PPARα in HEK293T cells in control PMA (0.5 µM for 1 day) and Go6983 (5 µM for 2 days) conditions. Quantification shows the ratio of PPARα to loading control, mean ± SEM, N = 3, ***P < 0.001, *P < 0.05, one-way ANOVA. **(C)** Quantitative PCR of the peroxisomal genes in HEK293T cells in control and Go6983 5 µM conditions (2-day treatment). Quantification shows color coded relative expression levels calculated as a ΔCt (peroxisomal gene—GAPDH expression reference) and a ratio of Go6983/control expression, expressed as 2$^{-\Delta\Delta Ct}$, mean ± SEM, N = 4, *P < 0.05, Kruskal–Wallis test. **(D and E)** Confocal microscopy of PPAR mCherry reporter in HEK293T cells expressing PPARα (D) and CHO (E) cells in control, Go6983 (5 µM for 2 days), or PMA conditions (0.5 µM for 1 day). Nuclei were stained with Hoechst (10 µg/ml). Representative images are shown. Scale bar: 5 µm. Quantification shows average fluorescence intensity per cell, N = 50, ****P < 0.0001, Kruskal–Wallis test. **(F)** Western blot and confocal microscopy of NBR1 and control silencing in HEK293T CRISPR/Cas9 PMP70-GFP cells. Nuclei were stained with Hoechst (10 µg/ml). Representative images are shown. Scale bar: 5 µm; inlet: 1 µm. Quantification of the western blot shows the NBR1/GAPDH ratio quantified by the lane intensity on the western blot, mean ± SEM, N = 3, unpaired t test. Quantification shows the number of peroxisomes per square micron of the cytoplasm in the 2D confocal image, mean ± SEM, N = 150 pooled from three biological repeats, Mann–Whitney. **(G)** Western blot and confocal microscopy of NBR1/NIX and control silencing in HEK293T cells. Nuclei were stained with Hoechst (10 µg/ml). Representative images are shown. Scale bar: 5 µm. Quantification shows the number of peroxisomes per square micron of the cytoplasm in the 2D confocal image, mean ± SEM, N = 100 pooled from three biological repeats, **** - -<0.0001, one-way ANOVA. **(H)** Confocal microscopy of peroxisomes stained with anti-PMP70 antibody in WT and VAPB/VAPA KO HeLa cells in control and GFP-VAPB overexpression conditions. Source data are available for this figure: SourceData FS4.

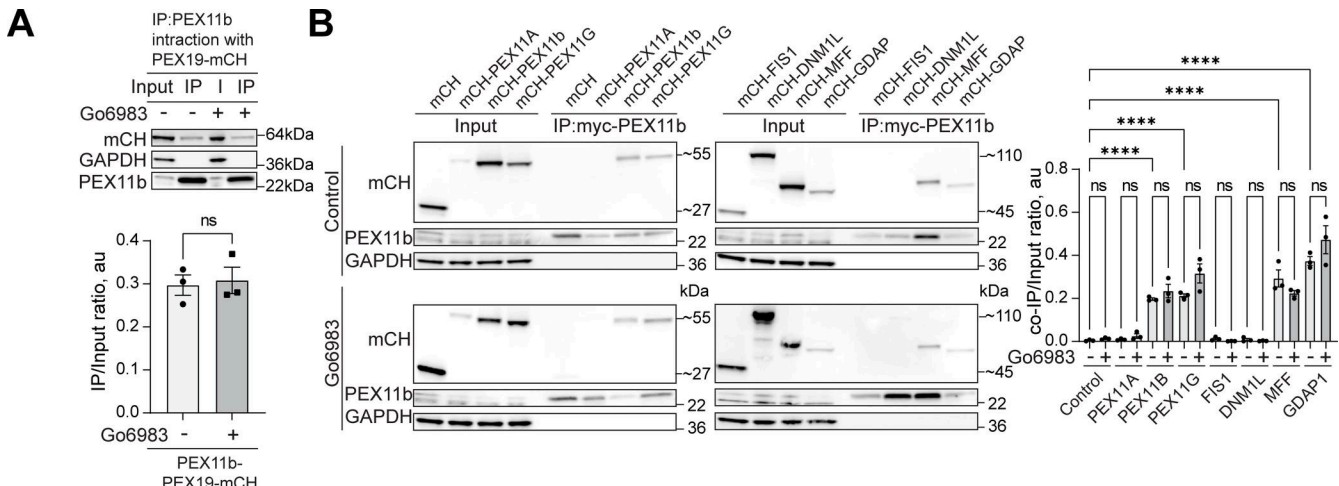

Figure S5. **Interaction of peroxisome division factors during PKC inhibition. (A)** Co-immunoprecipitation of mycPEX11b in control and PKC inhibition conditions (Go6983 5 µM, 24 h). Western blot of mycPEX11b and mCherry-PEX19 in the WT HEK293T cells. Quantification shows the IP to input (I) ratio, mean ± SEM, $N$ = 3. **(B)** Co-immunoprecipitation of mycPEX11b and peroxisome division factors in control and PKC inhibition conditions (Go6983 5 µM, 24 h). Western blot of mycPEX11b and mCherry-, mCherry-PEX11A, PEX11b, PEX11G, FIS1, DNM1L, MFF, or GDAP1 in the WT HEK293T cells. Quantification shows the IP to input (I) ratio, mean ± SEM, $N$ = 3, ****$P$ < 0.0001. Source data are available for this figure: SourceData FS5.

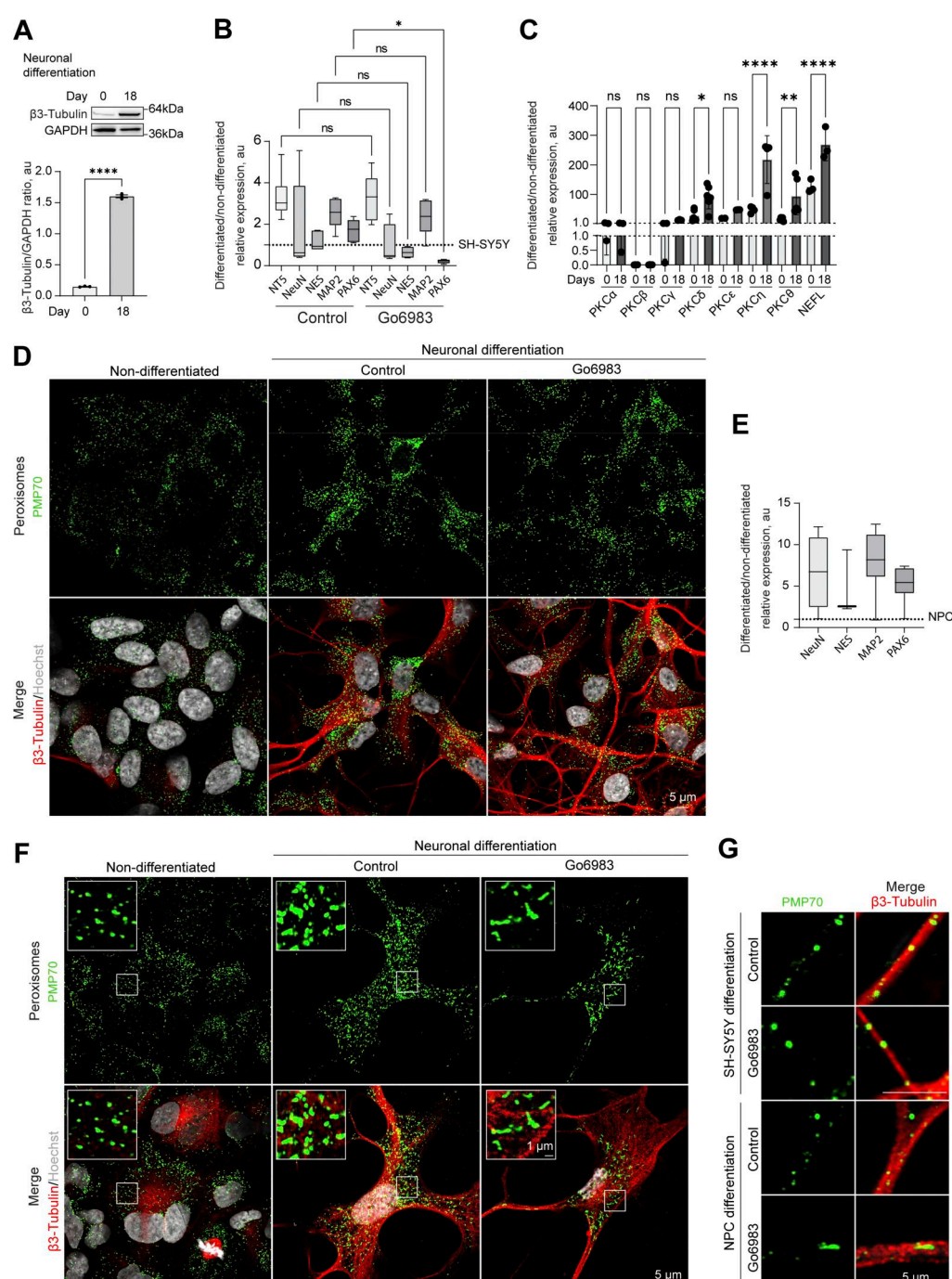

Figure S6. **Positive regulation of neuronal peroxisome abundance by PKC. (A)** Western blot of β-3 tubulin in the non-differentiated and 18-day differentiated SH-SY5Y. Quantification shows the ratio of and β-3 tubulin to GAPDH, mean ± SEM, $N = 3$, ****$P < 0.0001$, unpaired $t$ test. **(B and C)** Quantitative PCR of the neuronal markers and PKC isoforms in the (B) 18-day differentiated SH-SY5Y cells in control and Go6983 1 μM conditions; Go6983 was added on days 10–18 of the differentiation protocol. Quantification shows the relative expression of differentiated/non-differentiated markers, expressed as $2^{-\Delta\Delta Ct}$, mean ± SEM, $N = 4$, *$P < 0.05$, Kruskal–Wallis test, and (C) comparison of day 0 and 18 expression, mean ± SEM, $N = 4$, *$P < 0.05$, **$P < 0.01$, ****$P < 0.0001$, one-way ANOVA. **(D)** Confocal microscopy of peroxisomes in the non-differentiated and 18-day differentiated SH-SY5Y cells in control and Go6983 1 μM conditions. Go6983 was added on days 10–18 of the differentiation protocol. Peroxisomes were visualized with the PMP70 antibody, nuclei were stained with Hoechst (10 μg/ml), and neuronal differentiation was visualized with β-3 tubulin antibody. Representative images are shown. Scale bar: 5 μm. See zoomed in images in Fig. 5 F. **(E)** Quantitative PCR of the neuronal markers in the 12-day differentiated NPCs. Quantification shows the relative expression of differentiated/non-differentiated markers, expressed as $2^{-\Delta\Delta Ct}$, mean ± SEM. **(F)** Confocal microscopy of peroxisomes in the non-differentiated and 12-day differentiated NPCs in control and Go6983 1 μM conditions. Go6983 was added on days 1–12 of the differentiation protocol. Peroxisomes were visualized with the PMP70 antibody, nuclei were stained with Hoechst (10 μg/ml), and neuronal differentiation was visualized with β-3 tubulin antibody. Representative images are shown. Scale bar: 5 μm; inlet: 1 μm. **(G)** Confocal microscopy of peroxisomes in the neuronal terminals of the 18-day differentiated SH-SY5Y and 12-day differentiated NPCs in control and Go6983 1 μM conditions. Peroxisomes were visualized with the PMP70 antibody, and neuronal terminals were visualized with β-3 tubulin antibody. Representative images are shown. Scale bar: 5 μm. Source data are available for this figure: SourceData FS6.

Video 1.   **Peroxisomes and ER dynamics in control conditions.** Live cell time-lapse confocal microscopy of ER (ER-mCherry overexpression, red) and peroxisomes (GFP-SKL overexpression, green) in U2OS cells in control (Videos 1, 2, and 3) conditions, three frames per second display rate, and frames are collected as one frame per 2 s. See Fig. 4 E for video stills.

Video 2.   **Peroxisomes and ER dynamics in control conditions.** Live cell time-lapse confocal microscopy of ER (ER-mCherry overexpression, red) and peroxisomes (GFP-SKL overexpression, green) in U2OS cells in control (Videos 1, 2, and 3) conditions, three frames per second display rate, and frames are collected as one frame per 2 s. See Fig. 4 E for video stills.

Video 3.   **Peroxisomes and ER dynamics in control conditions.** Live cell time-lapse confocal microscopy of ER (ER-mCherry overexpression, red) and peroxisomes (GFP-SKL overexpression, green) in U2OS cells in control (Videos 1, 2, and 3) conditions, three frames per second display rate, and frames are collected as one frame per 2 s. See Fig. 4 E for video stills.

Video 4.   **Peroxisomes and ER dynamics in PKC inhibition conditions.** Live cell time-lapse confocal microscopy of ER (ER-mCherry overexpression, red) and peroxisomes (GFP-SKL overexpression, green) in U2OS cells in Go6983 1 μM 24-h (Videos 4, 5, and 6) conditions, three frames per second display rate, and frames are collected as one frame per 2 s. See Fig. 4 E for video stills.

Video 5.   **Peroxisomes and ER dynamics in PKC inhibition conditions.** Live cell time-lapse confocal microscopy of ER (ER-mCherry overexpression, red) and peroxisomes (GFP-SKL overexpression, green) in U2OS cells in Go6983 1 μM 24-h (Videos 4, 5, and 6) conditions, three frames per second display rate, and frames are collected as one frame per 2 s. See Fig. 4 E for video stills.

Video 6.   **Peroxisomes and ER dynamics in PKC inhibition conditions.** Live cell time-lapse confocal microscopy of ER (ER-mCherry overexpression, red) and peroxisomes (GFP-SKL overexpression, green) in U2OS cells in Go6983 1 μM 24-h (Videos 4, 5, and 6) conditions, three frames per second display rate, and frames are collected as one frame per 2 s. See Fig. 4 E for video stills.

**Provided online are Table S1 and Table S2. Table S1 contains the details on the small molecules used for the screening and the peroxisome density for each treatment with highlighted significant differences. Table S2 contains the information on statistical tests used in this manuscript, including repeats, sample size, normality tests, and parameter values.**

