## [Peer Review File · The Journal of Cell Biology]

Protein Kinase C promotes peroxisome biogenesis and peroxisome-endoplasmic reticulum interaction

Anya Borisyuk, Charlotte Howman, Sundararaghavan Pattabiraman, Daniel Kaganovich, and Triana Amen

Corresponding Author(s): Triana Amen, University of Southampton

Review Timeline:

Submission Date:	2025-05-08
Editorial Decision:	2025-05-16
Revision Received:	2025-06-09

Monitoring Editor: John Aitchison

Scientific Editor: Dan Simon

Transaction Report:

DOI: <https://doi.org/10.1083/jcb.202505040>

Revision 0

Review #1

1. Evidence, reproducibility and clarity:

Evidence, reproducibility and clarity (Required)

Using a compound kinase inhibitor screen, Borisyyuk et al. identified Protein Kinase C inhibitors which reduce the number of peroxisomes in human HEK cells, fibroblasts and neurons. To validate the peroxisome-specific regulatory effects of PKC they overexpressed several isoforms of PKC and found that only one overexpressed isoform PKCdelta could trigger peroxisome proliferation. Moreover, incubation of cells with known PKC activating compounds, e.g. phorbol esters (PMA), resulted in increasing number of peroxisomes per cell. Preliminary data let the authors suggest that PKC regulation does neither activate the transcriptional regulator PPARalpha nor inhibits selective peroxisomal autophagic degradation dependent of the NBR receptor. A role for PKC to regulate the division of mature peroxisomes is indicated by the observation that in the absence of the division factor PEX11b the abundance of peroxisomes was not affected by PKC inhibition. The data further suggests that PEX11b is not the phosphorylation target of the PKC signalling pathway. A good candidate for phosphorylation is glycogen synthase kinase beta (GSK3 β) which negatively regulates the contact of peroxisomes and ER via ACBD5/VAPB tethering (as recently shown by the Schrader group). Finally, the authors show the existence of the novel transduction cascade in differentiating neurons in culture.

The validation of PKC and identification of other constituents of the regulation of peroxisome proliferation is largely carried out using inhibitors or activators controlling a large variety of cellular responses. Several of the main conclusions based on the application of inhibitors and activators should be supported by additional experiments which directly target the PKCdelta isoform.

****Major concern:****

The suggested additional control experiments should be considered to support the following conclusions.

1. Protein Kinase C delta positively regulates peroxisome abundance

After the inhibitor screening, the inhibited kinase of the PKC family was identified. To this end, they overexpressed several kinases of this family but only the PKCdelta isoform shows an

increase in the number of peroxisomes (Fig.2). The next logical question would be whether the identified PKCdelta isoform interferes with peroxisome proliferation only indirectly or in combination with other kinases. To analyze this, the authors should downregulate the identified kinase isoform using siRNA and, as with the inhibitor, see a reduction in peroxisome number.

2. Protein Kinase C induces PEX11b-dependent peroxisome formation

The proliferation of peroxisomes is PEX11b-dependent. The authors observed no further reduction in the number of peroxisomes in the absence of PEX11b and incubation with a PKC inhibitor (Fig.3) From this they conclude that PEX11b-dependent peroxisome proliferation is induced by PKC. This may be correct, but in addition they should show that overexpression of PKCdelta in the PEX11b knockout cell line does not lead to an increase in peroxisomes.

3. PKC regulates peroxisome-ER contact sites through GSK3 β inhibition

The authors confirm previous studies that a lack of ER-peroxisome contact leads to a reduced number of peroxisomes. Their data (Fig 4C-F) suggest that this VAP-dependent interaction is regulated by PKC. In their model, PKC inactivated with inhibitors (Go6983) triggers an active form of GSK β , which in turn detaches peroxisomes from the ER. In fact, the inactivating phosphorylation of GSK3 β is suppressed by Go6983 and promoted by PMA (Fig. 4G). Again, siRNA knockdown or and overexpression of PKCdelta would be more appropriate than using the inhibitors and activators, both of which can potentially cause numerous indirect effects.

4. Protein kinase C regulates peroxisome abundance during neuronal differentiation

In the last section, they show that the general PKC inhibitor Go6983 reduces the number of peroxisomes in cell culture, albeit only slightly (Fig.5 E,F), and thus possibly also influences neuronal differentiation in humans. In this experimental setup, it is very difficult to show the specific effect of the PKCdelta isoform by siRNA-induced inactivation, but I would like to know whether overexpression of PKCdelta in these cultures reverses the effects. If the model shown is correct, another interesting question would be whether the expression of this isoform is upregulated in neuronal cells upon differentiation (see also minor concerns, point 3). For the correlation of the model with neuronal differentiation, it would also be interesting to see whether specific GSK3 inhibitors, as shown for HEK cells (Fig. S1G), have a neuronal effect.

****Minor points:****

1. Why were overexpression of PEX19 and PEX3 used as positive and negative controls, respectively, in the inhibitor screen? I was not aware that overexpression of PEX3 in human cell lines leads to a reduction in peroxisome numbers. As far as I know, the cited references refer to yeast experiments, i.e. Höhfeld et al. In addition, the controls used in Figure Fig 1 D,E are missing.

2. Page 4, line 35 the authors wrote: To also rule out a role for pexophagy, we silenced the NBR1

receptor, which did not affect peroxisome abundance over a two day period (Figure S4F, S4G), confirming that PKC does not upregulate peroxisomes by inhibiting their degradation. (Fig S4 F,G). The essential importance of this receptor for the selective degradation of peroxisomes has been shown previously (Ref 89). To understand the authors' conclusion, one should know under which conditions this experiment was performed, i.e. with or without Go6983 inhibitor ?

3. The authors postulate that PKC activity is particularly high in the rat brain. They refer to a reference 101, (Kikkawa et al), which dates back to 1982 and examines the enzyme localization in the rat brain. The reference on the expression of different PKC isoforms, especially PKCdelta, should be updated.

4. To better understand the results of the inhibitor screen and to use these as an open data source, a supplemental table should be generated containing name of the compounds, their specificity as well as peroxisome count.

2. Significance:

Significance (Required)

The approach to identify the kinases involved in peroxisome biogenesis is novel and straight forward. The presentation of the data as well as the accompanying text are very clear and suitable to arouse the interest of a large readership. The original data set of the screening (see also Minor point 4) can be used as a data source in future work.

The data are of great interest not only for specialists in the field. The identification of PKC significantly contributes to our understanding of the regulation of peroxisome biogenesis.

3. How much time do you estimate the authors will need to complete the suggested revisions:

Estimated time to Complete Revisions (Required)

(Decision Recommendation)

Between 1 and 3 months

4. Review Commons values the work of reviewers and encourages them to get credit for their work. Select 'Yes' below to register your reviewing activity at Web of Science Reviewer Recognition Service (formerly Publons); note that the content of your review will not be visible on Web of Science.

Yes

Review #2

1. Evidence, reproducibility and clarity:

Evidence, reproducibility and clarity (Required)

****Summary:****

This paper from Dr. Amen and colleagues uses a kinase inhibitor screen to identify potential regulators of peroxisome abundance in mammalian cells. They find several promising hits in the screen some of which link to the kinase PKC, a multifunctional kinase involved in a huge number of cellular processes. After exploring the possible mechanisms by which PKC might be regulating peroxisome abundance they eventually link PKC activity to recent work showing that GSK3-beta can regulate peroxisome connections to the ER. They also found a direct GSK3-beta inhibitor in the screen which further supports this concept. As GSK3-beta is a known PKC target this nicely links these two processes and solidifies/confirms some of the previous work in this area, allowing PKC to now be placed in a hypothetical model upstream of GSK3-beta in the pathway. Other work in the paper seems to rule out a link between PKC and pexophagy, activation of peroxisomal genes and modulation of the peroxisome biogenesis factor Pex11. To show a role for PKC they then link this to cell differentiation, when both peroxisomes and PKC activity are known to be increased and provide data suggesting that the peroxisomal increase during differentiation is dependent on PKC activity.

****Major comments:****

- Are the key conclusions convincing?

The conclusion that modulation of PKC alters peroxisome abundance is convincing.

The conclusion that PKC acts upstream of GSK3-beta to regulate the ACBD5-VAPB axis and thus regulate peroxisome-ER connections is convincing. That this is then directly linked to peroxisome abundance changes is still somewhat less convincing but could be built upon.

The conclusion that the increase in peroxisomal abundance during differentiation is PKC dependent is convincing.

The other data surrounding these key points is less convincing and some of the experiments

around Pex11 beta are a little confusingly presented, their relevance is less clear and at times this element detract from the other key conclusions.

- Should the authors qualify some of their claims as preliminary or speculative, or remove them altogether?

The narrative that the authors have around Pex11 is confusing, as none of the data they present seems to show any clear link to Pex11.

"We established a mechanistic connection between PKC activation and peroxisome biogenesis through PEX11b peroxisome formation".

This does not actually seem to fit your data. Your data fits with regulation of peroxisome-ER tethering regulation via GSK3-beta being potentially linked to peroxisome biogenesis and that PKC acts upstream of GSK3-beta in this pathway. In the model the Pex11 part of the pathway is downstream from this event and you don't show any link to Pex11 in any of your data - in fact the opposite - it seems to suggest Pex11 is not involved. The fact that in the Pex11 KO you do not see a change when you inhibit PKC would also fits with this. Pex11 is essential for this process and it is downstream of the PKC-GSK3-Beta-ACBD5-VAP part of the pathway. So perhaps no matter what you do with PKC, Pex11 will still be critical for peroxisome biogenesis.

I can see that a huge amount of work has been invested in the Pex11 link, so to remove it entirely seems unnecessary, but I think this aspect makes the current manuscript confusing and so it needs to be clarified.

- Would additional experiments be essential to support the claims of the paper?

If the main claims of the paper are clearly defined as I have set out above then these would be mostly supported by the current data.

If you want to really show that Pex11 beta is downstream of the PKC-GSK3-beta-ACBD5-VAPB step in the mechanism and make this point, you could consider the following:

The main phenotype of Pex11beta KO cells is elongated peroxisomes with a slightly reduced number (see example here <https://doi.org/10.3389/fcell.2020.577637>) which you presumably see in your own Pex11beta KO cell line. You also show a slightly reduced abundance in Figure 3G when comparing WT with Pex11 KO cells but do not discuss elongation. These elongations could potentially be due to membrane lipids still being provided by the ER, allowing initial peroxisome elongation prior to division but then a defect in the Pex11-driven part of the process (membrane deformation, interaction with MFF/Fis1, DLP1 activation etc) resulting in some elongation but inefficient division. Hence, a smaller number of elongated peroxisomes are observed. If PKC is driving interaction between ACBD5-VAPB, generating PO-ER contacts, allowing membrane growth of peroxisomes, upstream of Pex11beta. Then if you inhibit PKC you might expect to see less elongations in the Pex11 beta KO cells. The same number, but less elongated structures. So

you could assess this.

- Are the suggested experiments realistic in terms of time and resources?

The authors already have all the tools and may even have the images to be able to do this analysis.

- Are the data and the methods presented in such a way that they can be reproduced?

Yes, in general data are well presented with clear schematics.

- Are the experiments adequately replicated and statistical analysis adequate?

Yes, in general. However, one point to consider is that throughout the paper, N numbers are used, based on number of cells counted yet the number of actual experimental repeats is unclear. Should N be number of areas or number of experimental repeats? How many biological repeats has this come from? If none then any experimental variation (which can be considerable) is not being considered and could leave to false positives. Consider with reference to this paper: <https://doi.org/10.1083/jcb.202401074>

****Minor comments:****

- Specific experimental issues that are easily addressable.

To further convince the reader that the impacts you observe are not PMP70 specific, you could consider to another peroxisome marker to backup PMP70 work. This could be Pex11 - as you have also have a Pex11-FP reporter line used in Fig 3E.

- Are prior studies referenced appropriately?

In general yes but at times the references cited to evidence specific points do not clearly justify or explain these points or the evidence for your point is not cited.

E.g. Page 6 line 43

Why do you reference (117) the paper here from the Nunnari lab on ER-mitochondria contact sites. What does this imply?

E.g. "metabolic control of peroxisome abundance (MCPA)"

Where has this come from - it is not clear from the references given here (23, 60-63). Please give the reader a little more help and evidence here as from the citations you have given it is hard to know what to make of this point. Consider rephrasing, re-citing or giving a bit more context on how the data from these papers leads to this MCPA - it is not clear how some of them are related. 23- The Chang et al 1999 paper shows altered peroxisome number/morphology in beta oxidation mutant but this is not the same as a regulatory system. 60 - is based on rodent PPARs, 61 is a very old review on PPARs, 62 is about the discovery of the Pex11 gene, 63 - a review on membrane contact sites, the relevance to this point is not clear?

E.g. "Both peroxisome biogenesis mechanisms are initiated by several non-mutually exclusive pathways: transcriptional peroxisome proliferator receptor alpha (PPAR α) increases the level of division factors, peroxisome tethering to the endoplasmic reticulum (ER) through a recently defined membrane contact site that is formed between acyl-coenzyme A-binding domain protein 5 (ACBD5) and the ER protein vesicle-associated membrane protein-associated protein (VAPB)58, 59 46 47 potentially sources the membrane, and an increase in peroxisome function in the presence of 48 peroxisomal substrates promotes peroxisome formation via unknown mechanisms."

This sentence is very long and difficult to follow and citation use is confusing. We suggest you rephrase.

E.g. "transcriptional peroxisome proliferator receptor alpha (PPAR α) increases the level of division factors"

As this is not so widely known for nons-specialists can you give some more details here, cite the evidence for this in human cells? Which division factors?

E.g. "and an increase in peroxisome function in the presence of peroxisomal substrates promotes peroxisome formation via unknown mechanisms".

Again, as above, this is not completely clear so please cite here the evidence for this in human cells, and ideally provide some examples of this.

E.g. "The regulation of peroxisome abundance by pexophagy may be more relevant for clearance of peroxisomes, when excess peroxisomes are degraded, usually after a bout of upregulation 46, 81, 82, 89, 116"

I think this is an interesting idea but is there really any evidence for this in these studies? Or is it more likely pexophagy is usually employed during wider autophagy response, for overall cell survival in response to stress?

- Do you have suggestions that would help the authors improve the presentation of their data and conclusions?

General points

Intro - needs a bit more specific detail and examples as well as some clarity in some of the sections to explain the rationale for the experiments and what is already known from the literature. In the Introduction - it would be good for the reader to have a very clear and specific example in human cells of the "transient "on demand" need for peroxisomal metabolism and specifically biogenesis. This is very obvious in various yeast, rodents and other organisms but potentially less clear in humans.

I also suggest a thorough proofread - quite a few mistakes and sentences which could be more coherent. E.g. Slight typo on page 2 line 42 - 53-5, Page 3 - line 6 - typo. Figure S1 legend. Line 12 and 13 - should be G and F?

Please detail statistical analyses and experimental number used in the figure legends.

Specific minor points:

Please explain briefly why Pex19 and Pex3 were used as controls in the screen and how this works, this is unclear for the non-specialist. Please specify what was the cut-off for the screen for compounds which altered PO density, just statistically significantly different?

Figure 3 - Why have PMP70 levels not been restored by Day 4 of the complementation assay (Fig3C?). Please comment

Page 4 line 22 - this is a potentially important result. What is the anticipated/known time for peroxisome division - how does this relate to your 2 hours for PKC activation (Fig 3E). Can you first see spherical and then elongated peroxisomes when you do this?

Explain with a little bit more detail the de novo assay for the non-specialist.

Page 4 line 29 "we constructed a CRISPR/Cas9 PEX11b KO" in our GFP-PMP70 HEK293 background (I assume)?

"The ratio of phosphorylated to the overall identified peptides was 4.4% for Ser53". Can the authors explain this?

The peptide competition assay needs to be explained more clearly. It is hard to follow. This is an in-vitro assay with purified PKC? Using different Pex11 peptides and also a known PKC substrate. What are the substrates? Were any non PKC peptides included - do they all have the consensus sequence - if so then what does this tell us?

"PEX11b S53D overexpression, and that of the WT and S53A, had no effect on peroxisome number (after correction for PEX11b expression level)"

What does this mean after correction for Pex11b expression level? It is already known that Pex11 overexpression may not impact the number but it will likely result in more elongated peroxisomes. It is a little unclear what the expected outcome of these experiments would be?

Page 5 Line 18 - There is no concluding sentence and so what do we then take from this? You have found some phosphosites on Pex11-beta. Which could in theory be PKC sites as in vitro PKC can phosphorylate this site - but they do not do anything when you mutate them and you do not show they are PKC sites in cells.

Overall - data that Pex11 is a substrate for PKC remains unclear. This leads to the reader questioning what to take from this whole section on Pex11. I strongly suggest addressing this in a revised version.

Figure 4 - Just a comment: In the VAPA/VAPB double KO cells the peroxisome numbers/density may be reduced but there are also very significant changes in their morphology. Due to the wide-ranging roles of the VAPs in multiple different organelle contact sites in these KO cells, you would expect very significant changes in (all?) organelle morphology and wide-scale disruption to lipid transport and membrane lipid content. See here for an example of VAPA/VAPB double KO in endothelial cells. <https://doi.org/10.1016/j.jsbmb.2023.106349>

Is this therefore the best system to investigate changes in organelle morphology, linked to PKC?

Figure 5 - I was surprised that PKC inhibition seems to have no impact on differentiation in your assays? PKC is supposed to be a key player in differentiation and yet its inhibition seems to have no impact on the process. I am not familiar with all the work on this but has a similar experiment not been performed previously? I note the authors only use minimal differentiation markers. Should more markers of differentiation be considered?

The NBR1 silencing experiment in Fig S4, exploring pexophagy. I was confused by what this actually shows? This seems to go against the data in the Kim paper, albeit a different cell type, in which they showed that 2 days silencing NBR1 caused a big change in pexophagy observed <https://doi.org/10.1242/jcs.114819>. How can the authors explain this?

2. Significance:

Significance (Required)

- Describe the nature and significance of the advance (e.g. conceptual, technical, clinical) for the field.

The major nature of the advance is that it solidifies/confirms some of the concepts on how peroxisomes can multiply, by using their connections to the ER to promote peroxisome elongation and biogenesis. As the authors have done in Figure 5K we can now potentially add PKC upstream of GSK3-beta in this pathway of regulating the VAP-ACBD5 tethers. From this study a role of the PO-ER tethers can be more firmly linked to peroxisome elongation and biogenesis, although direct evidence is still lacking. We can now potentially link a role for peroxisomes in other PKC related processes, as the authors have done for the process of differentiation. So, some nice observations and connections with some advances in knowledge. As GSK3-beta is a well-known PKC target and GSK3-beta is already known to regulate the PO-ER tethers this of course reduces the overall significance of the work. However, it is much more significant that the PKC regulation data has come from an unbiased screening process rather than a biased, candidate-testing approach. The screen also provides several other regulatory mechanisms to pursue which can be followed up in future work.

There is solid data linking PKC to regulation of GSK3-beta and the ACBD5-VAPB axis. The rest of the work and the discussion around Pex11 is also solid negative data but this does not really add to our understanding. However, as very little is known about regulation of peroxisomes then this

study represents a great starting point which can be built on in future work looking at PKC.

- Place the work in the context of the existing literature (provide references, where appropriate).

This I have mostly done above. It fits with existing literature and is an incremental increase in knowledge with good potential for future work in this area of PKC regulation.

- State what audience might be interested in and influenced by the reported findings.

This will of course appeal to peroxisome researchers but also audiences interested in regulation of organelles and PKC signalling. This is a very nice study, with a well-utilised initial screen and if the authors address some of the issues I have highlighted to make the outcomes more clear then this will be clearly be worthy of publication in a cell biology-focused journal.

- Define your field of expertise with a few keywords to help the authors contextualize your point of view. Indicate if there are any parts of the paper that you do not have sufficient expertise to evaluate

Organelles, membrane contact sites, lipid metabolism

3. How much time do you estimate the authors will need to complete the suggested revisions:

Estimated time to Complete Revisions (Required)

(Decision Recommendation)

Cannot tell / Not applicable

4. Review Commons values the work of reviewers and encourages them to get credit for their work. Select 'Yes' below to register your reviewing activity at Web of Science Reviewer Recognition Service (formerly Publons); note that the content of your review will not be visible on Web of Science.

No

Review #3

1. Evidence, reproducibility and clarity:

Evidence, reproducibility and clarity (Required)

This manuscript uses a kinase inhibitor screen in human HEK293T cells to identify inhibitors that reduce or increase peroxisome abundance (peroxisome numbers/ μm^2). The authors found that 16 kinase inhibitors reduced, and 5 upregulated, the number of peroxisomes in HEK293T cells. They then undertook a secondary screen of the hits in untagged human fibroblasts, validating 10 compounds that reduced and 2 that upregulated the number of peroxisomes. Positive regulators (wherein inhibition of the kinase reduces peroxisome number) included: PKC, TGF β R, MEK1/2, ERK2, PDHK, IGF1R, and ALK4/7. Negative regulators (wherein inhibition of the kinase increases peroxisome numbers) were CK2 and IKK β .

Further work involved validation of the role of PKC in the regulation of peroxisome abundance. Because there are multiple classical (cPKC: α , β , and γ), novel (nPKC δ), and atypical (aPKC: ζ) PKC isoforms, the authors tried to see if any inhibitor tested was specific to one PKC isoform and found that this was not the case. Instead, they turned to overexpression of individual PKC isoforms and found that peroxisome proliferation was specific to PKC α overexpression. Using the phorbol ester, PMA, which activates all PKC isoforms, peroxisome abundance was enhanced in untagged human fibroblasts. The authors interpret these data as an indication of PKC as a positive regulator of peroxisome abundance. Consistent with this idea, inhibition of PKC isoforms in primary human fibroblasts reduced fatty acid oxidation relative to control.

The authors assessed the de novo formation of peroxisomes (by inducing PEX19 in PEX19-KO cells) in the presence and absence of PKC inhibitors and found that PKC was not involved de novo biogenesis. Instead, it affected the PEX11 δ -mediated division of peroxisomes. PEX11 δ KO cells had reduced peroxisome abundance as was previously shown, but they did not respond to PKC inhibition by further reduction of peroxisome numbers, suggesting PEX11 δ requirement for the action of the inhibitor. They ruled out that PKC acts through PPAR α activation. They also ruled out increased pexophagy as the cause of reduced peroxisome abundance in the presence of PKC inhibitors, by showing that depletion of one of the pexophagy receptors, NBR1, had no effect on peroxisome abundance.

Since yeast Pex11 is known to be activated for peroxisome division via phosphorylation, the PKC-phosphorylation site (Ser53 and other mapped sites of phosphorylation) in PKC was mutated but this was not a factor. However, PKC was found to regulate peroxisome-ER contact sites through inhibition of GSK3 δ , which had been shown previously by the Schrader lab to negatively regulate ER-peroxisome contact via phosphorylation of peroxisomal ABCD5 and its interaction with VAPB. Finally, to assess the physiological relevance of this mode of regulation of

peroxisome abundance, differentiating human neuronal cells (human neuroblastoma SH-SY5Y) were examined. PKC activity and peroxisome abundance were significantly increased in differentiating cells. Similarly, an increase in PKC activity and peroxisome number was observed during the differentiation of iPSC to neuronal progenitor cells. Inhibition of peroxisome proliferation with PKC inhibitor did not prevent neuronal differentiation of the SH-SY5Y cells even though peroxisomes were reduced.

****Referees cross-commenting****

this session contains comments of Rev 2 and Rev 3

Rev 3

Point #3, reviewer 3 may have been missed by the other reviewers. The modulation of ER-peroxisome contacts may have nothing to do with the increase in peroxisome abundance or be indirect. Until Drp1 (point2, reviewer 3), the more likely target of GSK3beta is tested, the data as presented maybe right or wrong. I agree with reviewer 2 that it is not worth chasing after Pex11beta with better experiments (e.g reviewer 1, point 2). The paper's emphasis on Pex11b-dependence could just as easily be Drp1-dependence (which acts in the same pathway as Pex11beta)

Rev 2

Yes, I am not so familiar with this role of GSK3-beta (this protein has so many roles!) and I had missed that Drp1 was also a PKC and GSK3 target.

However:

"Phosphorylation of Drp1 at S693 by GSK3 negatively affects Drp1 function".

If this is accurate and I understand correctly then inhibition of GSK3-beta may then be expected to give less Drp1 S693 phosphorylation and therefore effectively ACTIVATE Drp1? Or at

least reduce its inhibition? If the decreased peroxisome abundance seen here with the inhibitors is Drp1 dependent then would you expect less peroxisomes/peroxisome density due to activation of Drp1? In my mind this does not fit. It has been well published that loss of Drp1 activity results in elongated peroxisomes and mitochondria but I am not aware of work looking at the outcome of an activation of Drp1 on peroxisome number/elongation.

Rev 3

Drp1 phosphorylation at different sites can be activating or inhibiting, so it is hard to predict the outcome regarding peroxisome abundance. As stated in point 2, Drp1 phosphorylation is responsive to PKC- δ activation and inhibition, while also being involved in peroxisome fission, making it a reasonable target - more so than Pex11 β .

2. Significance:

Significance (Required)

The strength of this study is that it shows that PKC activation, and GSK3 β inhibition, are involved, both in vitro and physiologically, in the regulation of peroxisome abundance in human cells and this is driven primarily by affecting PEX11 β -mediated peroxisome division and not by affecting de novo peroxisome biogenesis and possibly pexophagy. The somewhat weaker conclusion is that ER-peroxisome contacts regulate peroxisome division. The manuscript should be of broad interest to cell biologists and scientists in the peroxisome field. The manuscript does have several issues that would need to be addressed before being considered to publication, but this should be encouraged.

My expertise is in peroxisome homeostasis and mechanisms of biogenesis and turnover.

****Major****

1. NBR1 is not the only pexophagy receptor (PMID: 36810161) so one should be cautious about the assumption that if NBR1 is knocked down, then pexophagy is completely inhibited (Fig. S4).

2. There is no direct evidence in mammalian cells that PEX11 β phosphorylation drives peroxisome division, as it does in some but not all yeast. So, it is not surprising that PEX11 β phosphorylation did not affect peroxisome proliferation by PKC. However, a more direct test would be for PKC activation of Drp1, the dynamin-like protein involved in peroxisome and mitochondrial division. PKC δ induces phosphorylation of Drp1. Additionally, PKC δ inhibitor (Go 6983) or PKC δ siRNA reversed phosphorylation of Drp1 (PMID: 35490835). So, the effect of PKC

activation or inhibition on Drp1 phosphorylation (especially at S579 which is believed to activate Drp1) should be examined. Notably, phosphorylation of Drp1 at S693 by GSK3 negatively affects Drp1 function (PMID: 23185298). If this hypothesis about Drp1 as the target of PKC is true, there need not be any direct involvement of contact sites, which may be coincidental rather than causal.

3. One concern in the data interpretation regarding the direct importance of ER-peroxisome contacts is that the VAP KO is used. VAPA/B make contacts between the ER and many organelles, making it challenging to assign the entire phenotype solely to the loss of ER-peroxisome contacts. It would be much more informative to knock out ACBD5, which should affect mostly peroxisome contacts. Since the effect of PKC activation or inhibition on VAP contacts with other organelles was not measured, it would be an overinterpretation to assign the peroxisome abundance phenotypes solely to ER-peroxisome contacts. Stronger evidence (e.g. ACBD5 KO) should be used to implicate primarily peroxisome-ER contacts.

****Minor****

1. The quantitation upon which this screen is based is poorly described and possibly limited. It is unclear how the cell segmentation was performed and whether Z-stack images were converted to MIPs for the analyses. It would also be useful to know what a give peroxisome/area means - is this the mean or median for a cell or a population treated a certain way? These details are important for reproducibility.

2. In general, the experimental details are very sparse, impacting reproducibility. All drug concentration used, as well as antibody dilutions and promoters used for overexpression of PKC isoforms and PEX genes should be specified consistently.

3. In yeast, GFP-tagging of Pex1 blocks its function in peroxisome division but not its peroxisomal location. Were any experiments done (e.g. comparison of peroxisome abundance in WT versus complemented PEX11 KO cells) to show that myc-PEX11 is fully functional because if not, both WT and mutants may be insensitive to PKC modulation.

3. How much time do you estimate the authors will need to complete the suggested revisions:

Estimated time to Complete Revisions (Required)

(Decision Recommendation)

Between 1 and 3 months

4. Review Commons values the work of reviewers and encourages them to get credit for their work. Select 'Yes' below to register your reviewing activity at Web of Science Reviewer Recognition Service (formerly Publons); note that the content of your review will not be visible

on Web of Science.

Yes

Manuscript number: RC- 2025-02877R

Corresponding author(s): Daniel, Kaganovich; Triana, Amen

1. General Statements [optional]

We thank all the reviewers for their constructive comments.

Reviewer #1 (Evidence, reproducibility and clarity (Required)):

Using a compound kinase inhibitor screen, Borisjuk et al. identified Protein Kinase C inhibitors which reduce the number of peroxisomes in human HEK cells, fibroblasts and neurons. To validate the peroxisome-specific regulatory effects of PKC they overexpressed several isoforms of PKC and found that only one overexpressed isoform PKCdelta could trigger peroxisome proliferation. Moreover, incubation of cells with known PKC activating compounds, e.g. phorbol esters (PMA), resulted in increasing number of peroxisomes per cell. Preliminary data let the authors suggest that PKC regulation does neither activate the transcriptional regulator PPARalpha nor inhibits selective peroxisomal autophagic degradation dependent of the NBR receptor. A role for PKC to regulate the division of mature peroxisomes is indicated by the observation that in the absence of the division factor PEX11b the abundance of peroxisomes was not affected by PKC inhibition. The data further suggests that PEX11b is not the phosphorylation target of the PKC signalling pathway. A good candidate for phosphorylation is glycogen synthase kinase beta (GSK3 β) which negatively regulates the contact of peroxisomes and ER via ACBD5/VAPB tethering (as recently shown by the Schrader group). Finally, the authors show the existence of the novel transduction cascade in differentiating neurons in culture.

The validation of PKC and identification of other constituents of the regulation of peroxisome proliferation is largely carried out using inhibitors or activators controlling a large variety of cellular responses. Several of the main conclusions based on the application of inhibitors and activators should be supported by additional experiments which directly target the PKCdelta isoform.

[Response] We thank this reviewer for their constructive comments. Please see the point-by-point responses below.

Major concern:

The suggested additional control experiments should be considered to support the following conclusions.

1. Protein Kinase C delta positively regulates peroxisome abundance

After the inhibitor screening, the inhibited kinase of the PKC family was identified. To this end, they overexpressed several kinases of this family but only the PKCdelta isoform shows an

increase in the number of peroxisomes (Fig.2). The next logical question would be whether the identified PKCdelta isoform interferes with peroxisome proliferation only indirectly or in combination with other kinases. To analyze this, the authors should downregulate the identified kinase isoform using siRNA and, as with the inhibitor, see a reduction in peroxisome number.

[Response] To address whether PKCdelta isoform is responsible for the phenotypes that we observed we constructed a CRISPR/Cas9 knockout of PKCdelta in HEK293T cells (Figure S1I). We observed no significant decrease in the number of peroxisomes in the KO compared to WT cells, ruling out PKCdelta as the only regulator of peroxisome number (Figure S1I). It is important to note that human cells express many isoforms of PKC with reported overlapping roles, therefore it would necessitate a detailed inquiry into what isoforms or their combinations maybe contributing to the effects we observed. We intend to explore it in the future work, and we included an experiment on the expression of different PKC isoforms in neuronal differentiation (see below).

2. Protein Kinase C induces PEX11b-dependent peroxisome formation

The proliferation of peroxisomes is PEX11b-dependent. The authors observed no further reduction in the number of peroxisomes in the absence of PEX11b and incubation with a PKC inhibitor (Fig.3) From this they conclude that PEX11b-dependent peroxisome proliferation is induced by PKC. This may be correct, but in addition they should show that overexpression of PKCdelta in the PEX11b knockout cell line does not lead to an increase in peroxisomes.

[Response] We overexpressed PKCD in PEX11bKO cells and observed no increase in peroxisome number, indicating that PKCD regulation requires PEX11b-dependent peroxisome division (Figure S4A).

3. PKC regulates peroxisome-ER contact sites through GSK3 β inhibition

The authors confirm previous studies that a lack of ER-peroxisome contact leads to a reduced number of peroxisomes. Their data (Fig 4C-F) suggest that this VAP-dependent interaction is regulated by PKC. In their model, PKC inactivated with inhibitors (Go6983) triggers an active form of GSK β , which in turn detaches peroxisomes from the ER. In fact, the inactivating phosphorylation of GSK3 β is suppressed by Go6983 and promoted by PMA (Fig. 4G). Again, siRNA knockdown or and overexpression of PKCdelta would be more appropriate than using the inhibitors and activators, both of which can potentially cause numerous indirect effects.

[Response] As suggested, we measured the interaction between ACBD5 and VAPB using co-immunoprecipitation in the presence of PKCD by overexpressing PKCD-mCherry. We found increased interaction between ACBD5 and VAPB when overexpressing PKCD, confirming our finding that PKC regulates peroxisome-ER contact sites (Figure 4G).

4. Protein kinase C regulates peroxisome abundance during neuronal differentiation

In the last section, they show that the general PKC inhibitor Go6983 reduces the number of peroxisomes in cell culture, albeit only slightly (Fig.5 E,F), and thus possibly also influences neuronal differentiation in humans. In this experimental setup, it is very difficult to show the specific effect of the PKCdelta isoform by siRNA-induced inactivation, but I would like to know whether overexpression of PKCdelta in these cultures reverses the effects. If the model shown is correct, another interesting question would be whether the expression of this isoform is upregulated in neuronal cells upon differentiation (see also minor concerns, point 3). For the

correlation of the model with neuronal differentiation, it would also be interesting to see whether specific GSK3 inhibitors, as shown for HEK cells (Fig. S1G), have a neuronal effect.

[Response] We attempted to overexpress PKCD-mCherry throughout neuronal differentiation. However, we didn't find neuronal cells overexpressing PKCD on day 18. This could be due to the toxicity of high levels of PKCD over long periods of time. In the future we will attempt to create a PKCD KO in neuronal cells with the validated constructs to confirm these findings. To further confirm the importance of PKC isoforms in neuronal differentiation we did an unbiased qPCR analysis of PKC expression during neuronal differentiation (Figure S7C). We found that most of the novel PKCs were upregulated, including PKCdelta, PKCeta and PKCtheta . We will explore whether other novel isoforms besides PKCdelta regulate peroxisome number in the future work.

Minor points:

1. Why were overexpression of PEX19 and PEX3 used as positive and negative controls, respectively, in the inhibitor screen? I was not aware that overexpression of PEX3 in human cell lines leads to a reduction in peroxisome numbers. As far as I know, the cited references refer to yeast experiments, i.e. Höhfeld et al. In addition, the controls used in Figure Fig 1 D,E are missing.

[Response] PEX3 was shown to lead to peroxisome degradation when overexpressed in mammalian cells by Yamashita et al. (doi: [10.4161/auto.29329](https://doi.org/10.4161/auto.29329)) due to peroxisome degradation, therefore we used it as a negative control. We didn't find a reference for overexpression of PEX19 in human cells with quantification of peroxisome number. In our hands it consistently led to increase in peroxisome number, we cited yeast work in which biogenesis factor was first isolated and shown to affect peroxisome number.

We didn't use overexpression in Figures 1D, E as transfection of primary human fibroblasts is very difficult and does not lead to a number of cells required for statistical analysis. In these figures we compared peroxisome number to an untreated control condition and scored significant increase or decrease in peroxisome number.

2. Page 4, line 35 the authors wrote: To also rule out a role for pexophagy, we silenced the NBR1 receptor, which did not affect peroxisome abundance over a two day period (Figure S4F, S4G), confirming that PKC does not upregulate peroxisomes by inhibiting their degradation. (Fig S4 F,G). The essential importance of this receptor for the selective degradation of peroxisomes has been shown previously (Ref 89). To understand the authors' conclusion, one should know under which conditions this experiment was performed, i.e. with or without Go6983 inhibitor ?

[Response] This experiment was performed in control conditions. The idea was that if PKC acts through pexophagy then pexophagy regulation alone should show significant increases or decreases in peroxisome number, which we didn't see in control conditions. In this revision we used silencing of NBR1 and NIX (another pexophagy receptor indicated to by Reviewer 3, PMID: 36810161) in both control and PKC activation conditions. We didn't find significant differences when pexophagy was inhibited in these conditions (similar upregulation of peroxisomes in PKC activation) from which we concluded that PKC does not act through pexophagy regulation (Figure S4G).

3. The authors postulate that PKC activity is particularly high in the rat brain. They refer to a reference 101, (Kikkawa et al), which dates back to 1982 and examines the enzyme localization in the rat brain. The reference on the expression of different PKC isoforms, especially PKCdelta, should be updated.

[Response] We updated the reference on PKC in the brain, by adding references that refer to PKCdelta neuronal populations in the brain that control feeding, alcohol use, and fear (doi.org/10.1038/nn.3767, <https://www.nature.com/articles/nature09553>, <https://www.science.org/doi/10.1126/sciadv.abg9045>, PMID: [20691763](https://pubmed.ncbi.nlm.nih.gov/20691763/)). Since there is lack of comparative expression studies of PKC expression during neuronal development, we decided to analyze PKC expression in SH-SY5Y cells that we are using for neuronal differentiation experiments. Real-time qPCR confirmed upregulation of several PKC isoforms when comparing non-differentiated and 18 day differentiated cultures (Figure S7C), including PKC delta, eta, and theta. These results indicate that other novel isoforms may also be contributing to the increased PKC activity and peroxisome proliferation, a subject for further studies.

4. To better understand the results of the inhibitor screen and to use these as an open data source, a supplemental table should be generated containing name of the compounds, their specificity as well as peroxisome count.

[Response] We include an updated table with the name of the compound, compound specificity to kinases, and peroxisome counts (Supplementary Table 1: Kinase Inhibitor Screen).

Reviewer #1 (Significance (Required)):

The approach to identify the kinases involved in peroxisome biogenesis is novel and straight forward. The presentation of the data as well as the accompanying text are very clear and suitable to arouse the interest of a large readership. The original data set of the screening (see also Minor point 4) can be used as a data source in future work.

The data are of great interest not only for specialists in the field. The identification of PKC significantly contributes to our understanding of the regulation of peroxisome biogenesis.

[Response] We thank this reviewer for their positive assessment of our work.

Reviewer #2 (Evidence, reproducibility and clarity (Required)):

Summary:

This paper from Dr. Amen and colleagues uses a kinase inhibitor screen to identify potential regulators of peroxisome abundance in mammalian cells. They find several promising hits in the screen some of which link to the kinase PKC, a multifunctional kinase involved in a huge number of cellular processes. After exploring the possible mechanisms by which PKC might be regulating peroxisome abundance they eventually link PKC activity to recent work showing that

GSK3-beta can regulate peroxisome connections to the ER. They also found a direct GSK3-beta inhibitor in the screen which further supports this concept. As GSK3-beta is a known PKC target this nicely links these two processes and solidifies/confirms some of the previous work in this area, allowing PKC to now be placed in a hypothetical model upstream of GSK3-beta in the pathway. Other work in the paper seems to rule out a link between PKC and pexophagy, activation of peroxisomal genes and modulation of the peroxisome biogenesis factor Pex11. To show a role for PKC they then link this to cell differentiation, when both peroxisomes and PKC activity are known to be increased and provide data suggesting that the peroxisomal increase during differentiation is dependent on PKC activity.

[Response] We thank this reviewer for their thoughtful and constructive comments and suggestions. We addressed all the comments below.

Major comments:

- Are the key conclusions convincing?

The conclusion that modulation of PKC alters peroxisome abundance is convincing. The conclusion that PKC acts upstream of GSK3-beta to regulate the ACBD5-VAPB axis and thus regulate peroxisome-ER connections is convincing. That this is then directly linked to peroxisome abundance changes is still somewhat less convincing but could be built upon. The conclusion that the increase in peroxisomal abundance during differentiation is PKC dependent is convincing. The other data surrounding these key points is less convincing and some of the experiments around Pex11 beta are a little confusingly presented, their relevance is less clear and at times this element detract from the other key conclusions.

[Response] In light of the comments regarding the lack of evidence that the PKC regulation of peroxisome-ER contact sites promotes peroxisome division we restructured the manuscript. – We think that more evidence and a direct investigation is required to link contact sites and peroxisome number. Although it is likely that contact sites are required to provide the membrane for the subsequent growth and division of peroxisome, this paper does not intend to bridge this knowledge gap. Showing that modulation of PKC alters peroxisome number and regulates peroxisome-ER contact site is what all reviewers thought as convincing evidence. Therefore, we changed the title to reflect that. We also put PEX11b data in supplementary (former Figure 3I and J) as it is potentially interesting but does not add to the conclusions, while leaving the negative PEX11b results in the manuscript.

- Should the authors qualify some of their claims as preliminary or speculative, or remove them altogether?

The narrative that the authors have around Pex11 is confusing, as none of the data they present seems to show any clear link to Pex11.

"We established a mechanistic connection between PKC activation and peroxisome biogenesis through PEX11b peroxisome formation".

Full Revision

This does not actually seem to fit your data. Your data fits with regulation of peroxisome-ER tethering regulation via GSK3-beta being potentially linked to peroxisome biogenesis and that PKC acts upstream of GSK3-beta in this pathway. In the model the Pex11 part of the pathway is downstream from this event and you don't show any link to Pex11 in any of your data - in fact the opposite - it seems to suggest Pex11 is not involved. The fact that in the Pex11 KO you do not see a change when you inhibit PKC would also fit with this. Pex11 is essential for this process and it is downstream of the PKC-GSK3-Beta-ACBD5-VAP part of the pathway. So perhaps no matter what you do with PKC, Pex11 will still be critical for peroxisome biogenesis. I can see that a huge amount of work has been invested in the Pex11 link, so to remove it entirely seems unnecessary, but I think this aspect makes the current manuscript confusing and so it needs to be clarified.

[Response] We agree with the Reviewer. We now moved the majority of PEX11b experiments to supplementary figures and renamed the result section which now focuses on PEX11b and DNM1L (see below).

- Would additional experiments be essential to support the claims of the paper?

If the main claims of the paper are clearly defined as I have set out above then these would be mostly supported by the current data.

If you want to really show that Pex11 beta is downstream of the PKC-GSK3-beta-ACBD5-VAPB step in the mechanism and make this point, you could consider the following:

The main phenotype of Pex11beta KO cells is elongated peroxisomes with a slightly reduced number (see example here <https://doi.org/10.3389/fcell.2020.577637>) which you presumably see in your own Pex11beta KO cell line. You also show a slightly reduced abundance in Figure 3G when comparing WT with Pex11 KO cells but do not discuss elongation. These elongations could potentially be due to membrane lipids still being provided by the ER, allowing initial peroxisome elongation prior to division but then a defect in the Pex11-driven part of the process (membrane deformation, interaction with MFF/Fis1, DLP1 activation etc) resulting in some elongation but inefficient division. Hence, a smaller number of elongated peroxisomes are observed. If PKC is driving interaction between ACBD5-VAPB, generating PO-ER contacts, allowing membrane growth of peroxisomes, upstream of Pex11beta. Then if you inhibit PKC you might expect to see less elongations in the Pex11 beta KO cells. The same number, but less elongated structures. So you could assess this.

[Response] We analyzed peroxisome size (area in square microns) in WT and PEX11b KO cells in control and PKC inhibition conditions. PKC inhibition resulted in a significant increase in the size of peroxisomes in both WT and PEX11b KO. We cannot explain the results of the increase, however in VAPA/B KO cells that lost contact sites we also see a significant enlargement of peroxisomes. More experiments and super-resolution techniques are needed to understand this phenomenon and to resolve potential clustering issues. Therefore, we did not include this experiment in the paper to avoid more confusion, but we insert it here for the reviewer:

Figure for the reviewer. Graph shows the size of peroxisomes in HEK293T WT and PEX11b KO cells in control and PKC inhibition conditions (Go6983, 10uM, 24h), mean± SEM, N=5-10 biological repeats, each biological repeat is an average of peroxisome size in 100 cells , **** - $p < 0.0001$, *** - $p < 0.001$, ns – nonsignificant.

To clarify the issue of whether PKC acts upstream peroxisome division, we performed a similar experiment using MFF KO. MFF KO has significantly elongated peroxisomes. We quantified the length of the organelle in the control and PKC inhibition conditions and found a significant reduction in the elongation when PKC is inhibited (Figure 3I, J). We also performed several experiments on the DNM1L KO complemented with phosphomutants of DNM1L in different conditions (see Figures S7, S8 and comments for Reviewer 3). However, we didn't find the difference when comparing control and PKC modulation, or phosphomutants, indicating that PKC acts upstream of peroxisome division.

Additionally, we added an experiment showing that overexpression of PKCD increases the contact site formation assessed by co-IP of VAPB-ACBD5 (Figure 4G). Together the data points toward contact site regulation by PKC, rather than regulation of peroxisome division.

- Are the suggested experiments realistic in terms of time and resources?

The authors already have all the tools and may even have the images to be able to do this analysis.

- Are the data and the methods presented in such a way that they can be reproduced?

Yes, in general data are well presented with clear schematics.

- Are the experiments adequately replicated and statistical analysis adequate?

Yes, in general. However, one point to consider is that throughout the paper, N numbers are

used, based on number of cells counted yet the number of actual experimental repeats is unclear. Should N be number of areas or number of experimental repeats? How many biological repeats has this come from? If none then any experimental variation (which can be considerable) is not being considered and could leave to false positives. Consider with reference to this paper: <https://doi.org/10.1083/jcb.202401074>

[Response] We performed 3 biological repeats for each experiment, and the number of areas or cells was pooled from all the biological repeats. We added biological repeats information into the statistical analysis section, and supplementary table 3 – Statistical analysis. We also indicated in legends all the biological repeats to avoid confusion.

Minor comments:

- Specific experimental issues that are easily addressable.

To further convince the reader that the impacts you observe are not PMP70 specific, you could consider to another peroxisome marker to backup PMP70 work. This could be Pex11 - as you have also have a Pex11-FP reporter line used in Fig 3E.

[Response] In revision we added other protein and non-protein markers including PEX14 and PeroxiSPY (Figures S4A, S4G, S7, S8) in addition to PEX11b used in Fig.3E.

- Are prior studies referenced appropriately?

In general yes but at times the references cited to evidence specific points do not clearly justify or explain these points or the evidence for your point is not cited.

E.g. Page 6 line 43

Why do you reference (117) the paper here from the Nunnari lab on ER-mitochondria contact sites. What does this imply?

[Response] This reference was meant as an example of how the known organelle contact can promote organelle division, evidencing the possibility that these two mechanisms can be linked experimentally. Since the sentence was meant as a future research direction, we removed this reference.

E.g. "metabolic control of peroxisome abundance (MCPA)"

Where has this come from - it is not clear from the references gives here (23, 60-63). Please give the reader a little more help and evidence here as from the citations you have given it is hard to know what to make of this point. Consider rephrasing, re-citing or giving a bit more context on how the data from these papers leads to this MCPA - it is not clear how some of them are related. 23- The Chang et al 1999 paper shows altered peroxisome number/morphology in beta oxidation mutant but this is not the same as a regulatory system. 60 - is based on rodent PPARs, 61 is a very old review on PPARs, 62 is about the discovery of the Pex11 gene, 63 - a review on membrane contact sites, the relevance to this point is not clear?

[Response] Thank you for indicating that. We updated the text, removing the sentence and rearranging the paragraph (see below).

E.g. "Both peroxisome biogenesis mechanisms are initiated by several non-mutually exclusive pathways: transcriptional peroxisome proliferator receptor alpha (PPAR α) increases the level of division factors, peroxisome tethering to the endoplasmic reticulum (ER) through a recently defined membrane contact site that is formed between acyl-coenzyme A-binding domain protein 5 (ACBD5) and the ER protein vesicle-associated membrane protein-associated protein (VAPB)58, 59 46 47 potentially sources the membrane, and an increase in peroxisome function in the presence of 48 peroxisomal substrates promotes peroxisome formation via unknown mechanisms."

This sentence is very long and difficult to follow and citation use is confusing. We suggest you rephrase.

[Response] We rephrased and split it into several sentences to improve readability. The references were updated to indicate specific findings of PPARalpha role in upregulating PEX11a, ACOX1, ABCD2 and 3 ((doi.org/10.1371/journal.pone.0006796, doi: 10.1155/2010/612089). We also added a citation for peroxisomal substrates increasing peroxisome abundance ([10.1016/j.devcel.2024.06.020](https://doi.org/10.1016/j.devcel.2024.06.020)) and how PEX11b and to a lesser extent PEX11a regulate peroxisome formation (doi.org/10.1074/jbc.273.45.29607).

E.g. "transcriptional peroxisome proliferator receptor alpha (PPAR α) increases the level of division factors"

As this is not so widely known for nons-specialists can you give some more details here, cite the evidence for this in human cells? Which division factors?

[Response] We added appropriate references for the upregulation of PEX11a by PPAR α in human and rodent cells (doi.org/10.1371/journal.pone.0006796, doi: 10.1155/2010/612089)

E.g. "and an increase in peroxisome function in the presence of peroxisomal substrates promotes peroxisome formation via unknown mechanisms".

Again, as above, this is not completely clear so please cite here the evidence for this in human cells, and ideally provide some examples of this.

[Response] We rephrased this section and added citation how very long-chain fatty acids can promote peroxisome abundance through PPAR response ([10.1016/j.devcel.2024.06.020](https://doi.org/10.1016/j.devcel.2024.06.020)).

E.g. "The regulation of peroxisome abundance by pexophagy may be more relevant for clearance of peroxisomes, when excess peroxisomes are degraded, usually after a bout of upregulation 46, 81, 82, 89, 116"

I think this is an interesting idea but is there really any evidence for this in these studies? Or is it more likely pexophagy is usually employed during wider autophagy response, for overall cell survival in response to stress?

[Response] Thank you for indicating that, indeed it would be very relevant for starvation responses. We modified the sentence to reflect that. All the textual modifications are highlighted in the manuscript.

- Do you have suggestions that would help the authors improve the presentation of their data and conclusions?

General points

Intro - needs a bit more specific detail and examples as well as some clarity in some of the sections to explain the rationale for the experiments and what is already known from the literature. In the Introduction - it would be good for the reader to have a very clear and specific example in human cells of the "transient "on demand" need for peroxisomal metabolism and specifically biogenesis. This is very obvious in various yeast, rodents and other organisms but potentially less clear in humans.

[Response] We now added a specific example of transient reversible increase in peroxisome number upon ethanol treatment in liver cells and tissues (<https://doi.org/10.1091/mbc.E24-06-0252>).

I also suggest a thorough proofread - quite a few mistakes and sentences which could be more coherent. E.g. Slight typo on page 2 line 42 - 53-5, Page 3 - line 6 - typo. Figure S1 legend. Line 12 and 13 - should be G and F?

[Response] Thank you for indicating that. We corrected the mistakes and proofread the text.

Please detail statistical analyses and experimental number used in the figure legends.

[Response] We added statistical analyses in the figure legends. Full information is in the Supplementary Table – Statistical Analysis.

Specific minor points:

Please explain briefly why Pex19 and Pex3 were used as controls in the screen and how this works, this is unclear for the non-specialist. Please specify what was the cut-off for the screen for compounds which altered PO density, just statistically significantly different?

[Response] PEX3 was indicated to lead to peroxisome degradation when overexpressed in mammalian cells by Yamashita et al. (doi: [10.4161/auto.29329](https://doi.org/10.4161/auto.29329)) due to peroxisome degradation, therefore we used it as a negative control. We didn't find a reference for overexpression of PEX19 in human cells, in our hands it consistently led to increase in peroxisome number, we cited yeast work in which biogenesis factor was first isolated and shown to affect peroxisome number. We added an explanation in the Results section.

We used statistical significance to determine the "hits" from the screen. We included a table with all the values and kinase information in supplementary (table 3).

Figure 3 - Why have PMP70 levels not been restored by Day 4 of the complementation assay (Fig3C?). Please comment

[Response] We used transient transfection in our experiments. The blot shows lysates that contain a mix of transfected and untransfected cells. The low restoration reflects the transfection efficiency combined with outgrowing of cells for days after transfection. We think that in transfected cells PMP70 levels were restored fully as visible in the images (Figure 3D).

Page 4 line 22 - this is a potentially important result. What is the anticipated/known time for peroxisome division - how does this relate to your 2 hours for PKC activation (Fig 3E). Can you first see spherical and then elongated peroxisomes when you do this?

Explain with a little bit more detail the de novo assay for the non-specialist.

[Response] We saw full restoration of peroxisomes in complemented cells over the course of 4 days during de novo formation (when peroxisomes form, they likely start dividing). Peroxisome number doubles upon cell division, HEK293T cells divide approximately every 12 hours and shortly after division cells will have a number of peroxisomes comparable to the mother cell. The anticipated time for peroxisome division is on a several hours scale, in our movies we occasionally observe the intermediate steps of peroxisome division – elongated peroxisomes, beads on a string -like phenotype with several small peroxisomes positioned one after the other, however all these intermediates are rare.

We added more detail on the de novo assay in the results section.

Page 4 line 29 "we constructed a CRISPR/Cas9 PEX11b KO" in our GFP-PMP70 HEK293 background (I assume)?

[Response] Yes, we added this clarification.

"The ratio of phosphorylated to the overall identified peptides was 4.4% for Ser53". Can the authors explain this?

[Response] A rather low ratio means that only ~4 copies out of 100 were phosphorylated (over 50% for S38, Figure S5B). This could mean not optimal conditions (phosphorylation event happened earlier followed by dephosphorylation or phosphorylation event is just starting, or incomplete activation of PKC), however it becomes irrelevant to further optimize these conditions knowing that phosphorylation on this residue is not relevant for the phenotypes we study (based on the complementation experiments).

The peptide competition assay needs to be explained more clearly. It is hard to follow. This is an in-vitro assay with purified PKC? Using different Pex11 peptides and also a known PKC substrate. What are the substrates? Were any non PKC peptides included - do they all have the consensus sequence - if so then what does this tell us?

[Response] Yes, this is an in vitro assay with known PKC substrate peptide as a control and the identified PEX11b peptides (the sequences are in the method section). We used prediction algorithm to identify PKC consensus sequences in PEX11b sequence and identified 6 sequences including S53. We confirmed by identification of PEX11b phosphopeptides in cells (PEX11b pulldown) that one of them S53 is phosphorylated. This shows that S53 is phosphorylated in the conditions we used, however it doesn't show that PKC can directly phosphorylate it. To show that PKC can directly phosphorylate we used S53 and PKC known substrate in the in vitro assay which does not contain any other kinases except for PKC. As a negative control we used no PKC samples, and we also incorporated all other PEX11b PKC substrate peptides, among which several were shown to be phosphorylated in the in vitro assay, and some were not. This data tells us that PKC can directly phosphorylate S53 sequence and

that it is phosphorylated in cells to a low extent. We then show that this phosphorylation event doesn't regulate peroxisome number, so we do not know the function of this phosphorylation by PKC. For this paper it is a dead end, however the experiments are of value to further investigation of the role of PEX11b phosphorylation status, therefore we included them in the supplementary figures.

We have now clarified the competition assay; however, we would like to make this section simple as it doesn't add to the main conclusions.

"PEX11b S53D overexpression, and that of the WT and S53A, had no effect on peroxisome number (after correction for PEX11b expression level)"

What does this mean after correction for Pex11b expression level? It is already known that Pex11 overexpression may not impact the number but it will likely result in more elongated peroxisomes. It is a little unclear what the expected outcome of these experiments would be? [Response] We corrected to the level of PEX11b overexpression as there was a significant reduction in S53A variant when expressed (Figure S5E). We did this experiment to prove whether phosphorylation of PEX11b by PKC on S53 impacts the number of peroxisomes. We didn't know the outcome; the expectation was either to rule out the possibility or not.

Page 5 Line 18 - There is no concluding sentence and so what do we then take from this? You have found some phosphosites on Pex11-beta. Which could in theory be PKC sites as in vitro PKC can phosphorylate this site - but they do not do anything when you mutate them and you do not show they are PKC sites in cells.

[Response] We added a concluding sentence, stating that PKC regulation of peroxisome biogenesis is not through PEX11b.

Overall - data that Pex11 is a substrate for PKC remains unclear. This leads to the reader questioning what to take from this whole section on Pex11. I strongly suggest addressing this in a revised version.

[Response] In the revised version we put all the experiments of PEX11b section in supplementary. Since there is an addition of DNM1L negative results, we combined this into one section.

Figure 4 - Just a comment: In the VAPA/VAPB double KO cells the peroxisome numbers/density may be reduced but there are also very significant changes in their morphology. Due to the wide-ranging roles of the VAPs in multiple different organelle contact sites in these KO cells, you would expect very significant changes in (all?) organelle morphology and wide-scale disruption to lipid transport and membrane lipid content. See here for an example of VAPA/VAPB double KO in endothelial cells. <https://doi.org/10.1016/j.jsbmb.2023.106349>

Is this therefore the best system to investigate changes in organelle morphology, linked to PKC?

[Response] We agree with the reviewer, that it is not fully clear how the absence of other contact sites in VAPA/B KO affects peroxisome biogenesis. However, we think that VAPA/B KO is an essential model since it is a model that completely lacks peroxisome contacts through

VAPs. Most of the experiments supporting the claims were performed in different models. We now added this limitation to the discussion.

Figure 5 - I was surprised that PKC inhibition seems to have no impact on differentiation in your assays? PKC is supposed to be a key player in differentiation and yet its inhibition seems to have no impact on the process. I am not familiar with all the work on this but has a similar experiment not been performed previously? I note the authors only use minimal differentiation markers. Should more markers of differentiation be considered?

[Response] We think that due to a competition of PKC inhibitor with PKC activator we were only able to achieve partial PKC inhibition, which allowed neuronal differentiation. We didn't find similar experiments, however it is known that there are PKCD positive and negative neuronal populations, it is possible that during general GABA/dopaminergic differentiation we do not see the differences because the neurons are not reaching a mature state in 18 days, nor are we working with specific populations that particularly highly overexpress PKC (such as PKCdelta positive populations that control alcohol use, fear response, and feeding: doi.org/10.1038/nn.3767, <https://www.nature.com/articles/nature09553>, <https://www.science.org/doi/10.1126/sciadv.abq9045>, PMID: [20691763](https://pubmed.ncbi.nlm.nih.gov/20691763/)).

In the revised version we looked at what isoforms of PKC are expressed during neuronal differentiation and we also included neurofilament as additional marker of differentiation (Figure S9C).

The NBR1 silencing experiment in Fig S4, exploring pexophagy. I was confused by what this actually shows? This seems to go against the data in the Kim paper, albeit a different cell type, in which they showed that 2 days silencing NBR1 caused a big change in pexophagy observed <https://doi.org/10.1242/jcs.114819>. How can the authors explain this?

[Response] The idea of this experiment was that if PKC acts through pexophagy then pexophagy regulation alone should show significant increases or decreases in peroxisome number, which we didn't see in control conditions. In this revision we used silencing of NBR1 and NIX (another pexophagy receptor indicated to by Reviewer 3, PMID: 36810161) in both control and PKC activation conditions. We didn't find significant differences in PKC response when pexophagy was inhibited in these conditions (similar upregulation of peroxisomes in PKC activation) from which we concluded that PKC does not act through pexophagy regulation (Figure S4G). We think that the most likely explanation for not observing an increase in the number of peroxisomes during silencing of the pexophagy receptors is peroxisome clustering that can obscure the quantification (cluster of peroxisomes may be counted as one peroxisome by the FiJi algorithm). In HEK293T cells we see a certain amount of enlarged, potentially clustered peroxisomes in silenced conditions –

From Figure 4G – arrowhead indicates enlarged/clustered peroxisome.

Reviewer #2 (Significance (Required)):

- Describe the nature and significance of the advance (e.g. conceptual, technical, clinical) for the field.

The major nature of the advance is that it solidifies/confirms some of the concepts on how peroxisomes can multiply, by using their connections to the ER to promote peroxisome elongation and biogenesis. As the authors have done in Figure 5K we can now potentially add PKC upstream of GSK3-beta in this pathway of regulating the VAP-ACBD5 tethers. From this study a role of the PO-ER tethers can be more firmly linked to peroxisome elongation and biogenesis, although direct evidence is still lacking. We can now potentially link a role for peroxisomes in other PKC related processes, as the authors have done for the process of differentiation. So, some nice observations and connections with some advances in knowledge. As GSK3-beta is a well-known PKC target and GSK3-beta is already known to regulate the PO-ER tethers this of course reduces the overall significance of the work. However, it is much more significant that the PKC regulation data has come from an unbiased screening process rather than a biased, candidate-testing approach. The screen also provides several other regulatory mechanisms to pursue which can be followed up in future work.

There is solid data linking PKC to regulation of GSK3-beta and the ACBD5-VAPB axis. The rest of the work and the discussion around Pex11 is also solid negative data but this does not really add to our understanding. However, as very little is known about regulation of peroxisomes then this study represents a great starting point which can be built on in future work looking at PKC.

- Place the work in the context of the existing literature (provide references, where appropriate). This I have mostly done above. It fits with existing literature and is an incremental increase in knowledge with good potential for future work in this area of PKC regulation.

- State what audience might be interested in and influenced by the reported findings. This will of course appeal to peroxisome researchers but also audiences interested in regulation

Full Revision

of organelles and PKC signalling. This is a very nice study, with a well-utilised initial screen and if the authors address some of the issues I have highlighted to make the outcomes more clear then this will be clearly be worthy of publication in a cell biology-focused journal.

- Define your field of expertise with a few keywords to help the authors contextualize your point of view. Indicate if there are any parts of the paper that you do not have sufficient expertise to evaluate

Organelles, membrane contact sites, lipid metabolism

[Response] We thank this reviewer for thorough reading of our manuscript.

Reviewer #3 (Evidence, reproducibility and clarity (Required)):

This manuscript uses a kinase inhibitor screen in human HEK293T cells to identify inhibitors that reduce or increase peroxisome abundance (peroxisome numbers/ μm^2). The authors found that 16 kinase inhibitors reduced, and 5 upregulated, the number of peroxisomes in HEK293T cells. They then undertook a secondary screen of the hits in untagged human fibroblasts, validating 10 compounds that reduced and 2 that upregulated the number of peroxisomes. Positive regulators (wherein inhibition of the kinase reduces peroxisome number) included: PKC, TGF β R, MEK1/2, ERK2, PDHK, IGF1R, and ALK4/7. Negative regulators (wherein inhibition of the kinase increases peroxisome numbers) were CK2 and IKK β .

Further work involved validation of the role of PKC in the regulation of peroxisome abundance. Because there are multiple classical (cPKC: α , β . and γ), novel (nPKC δ), and atypical (aPKC: ζ) PKC isoforms, the authors tried to see if any inhibitor tested was specific to one PKC isoform and found that this was not the case. Instead, they turned to overexpression of individual PKC isoforms and found that peroxisome proliferation was specific to PKC δ overexpression. Using the phorbol ester, PMA, which activates all PKC isoforms, peroxisome abundance was enhanced in untagged human fibroblasts. The authors interpret these data as an indication of PKC as a positive regulator of peroxisome abundance. Consistent with this idea, inhibition of PKC isoforms in primary human fibroblasts reduced fatty acid oxidation relative to control.

The authors assessed the de novo formation of peroxisomes (by inducing PEX19 in PEX19-KO cells) in the presence and absence of PKC inhibitors and found that PKC was not involved de novo biogenesis. Instead, it affected the PEX11 β -mediated division of peroxisomes. PEX11 β KO cells had reduced peroxisome abundance as was previously shown, but they did not respond to PKC inhibition by further reduction of peroxisome numbers, suggesting PEX11 β requirement for the action of the inhibitor. They ruled out that PKC acts through PPAR α activation. They also ruled out increased pexophagy as the cause of reduced peroxisome abundance in the presence of PKC inhibitors, by showing that depletion of one of the

pexophagy receptors, NBR1, had no effect on peroxisome abundance.

Since yeast Pex11 is known to be activated for peroxisome division via phosphorylation, the PKC-phosphorylation site (Ser53 and other mapped sites of phosphorylation) in PKC was mutated but this was not a factor. However, PKC was found to regulate peroxisome-ER contact sites through inhibition of GSK3 β , which had been shown previously by the Schrader lab to negatively regulate ER-peroxisome contact via phosphorylation of peroxisomal ABCD5 and its interaction with VAPB. Finally, to assess the physiological relevance of this mode of regulation of peroxisome abundance, differentiating human neuronal cells (human neuroblastoma SH-SY5Y) were examined. PKC activity and peroxisome abundance were significantly increased in differentiating cells. Similarly, an increase in PKC activity and peroxisome number was observed during the differentiation of iPSC to neuronal progenitor cells. Inhibition of peroxisome proliferation with PKC inhibitor did not prevent neuronal differentiation of the SH-SY5Y cells even though peroxisomes were reduced.

Referees cross-commenting

this session contains comments of Rev 2 and Rev 3

Rev 3

Point #3, reviewer 3 may have been missed by the other reviewers. The modulation of ER-peroxisome contacts may have nothing to do with the increase in peroxisome abundance or be indirect. Until Drp1 (point2, reviewer 3), the more likely target of GSK3beta is tested, the data as presented maybe right or wrong. I agree with reviewer 2 that it is not worth chasing after Pex11beta with better experiments (e.g reviewer 1, point 2). The paper's emphasis on Pex11b-dependence could just as easily be Drp1-dependence (which acts in the same pathway as Pex11beta)

Rev 2

Yes, I am not so familiar with this role of GSK3-beta (this protein has so many roles!) and I had missed that Drp1 was also a PKC and GSK3 target.

However:

"Phosphorylation of Drp1 at S693 by GSK3 negatively affects Drp1 function".

If this is accurate and I understand correctly then inhibition of GSK3-beta may then be expected to give less Drp1 S693 phosphorylation and therefore effectively ACTIVATE Drp1? Or at least reduce its inhibition? If the decreased peroxisome abundance seen here with the inhibitors is Drp1 dependent then would you expect less peroxisomes/peroxisome density due to activation of Drp1? In my mind this does not fit. It has been well published that loss of Drp1 activity results in elongated peroxisomes and mitochondria but I am not aware of work looking at the outcome of an activation of Drp1 on peroxisome number/elongation.

Rev 3

Drp1 phosphorylation at different sites can be activating or inhibiting, so it is hard to predict the outcome regarding peroxisome abundance. As stated in point 2, Drp1 phosphorylation is responsive to PKC-delta activation and inhibition, while also being involved in peroxisome fission, making it a reasonable target - more so than Pex11beta.

Reviewer #3 (Significance (Required)):

The strength of this study is that it shows that PKC activation, and GSK3 β inhibition, are involved, both in vitro and physiologically, in the regulation of peroxisome abundance in human cells and this is driven primarily by affecting PEX11 β -mediated peroxisome division and not by affecting de novo peroxisome biogenesis and possibly pexophagy. The somewhat weaker conclusion is that ER-peroxisome contacts regulate peroxisome division. The manuscript should be of broad interest to cell biologists and scientists in the peroxisome field. The manuscript does

have several issues that would need to be addressed before being considered to publication, but this should be encouraged.

My expertise is in peroxisome homeostasis and mechanisms of biogenesis and turnover.

[Response] We thank this reviewer for their constructive comments and thorough reading of our manuscript. Please find point-by-point response below.

Major

1. NBR1 is not the only pexophagy receptor (PMID: 36810161) so one should be cautious about the assumption that if NBR1 is knocked down, then pexophagy is completely inhibited (Fig. S4).

[Response] Thank you for indicating that. We included an experiment where we silenced NIX and NBR1 together in control and PKC activation conditions. We observed upregulation in peroxisome number in both control and silencing during PMA treatment, supporting the conclusion that PKC does not act through pexophagy regulation (Figure S4G).

2. There is no direct evidence in mammalian cells that PEX11 β phosphorylation drives peroxisome division, as it does in some but not all yeast. So, it is not surprising that PEX11 β phosphorylation did not affect peroxisome proliferation by PKC. However, a more direct test would be for PKC activation of Drp1, the dynamin-like protein involved in peroxisome and mitochondrial division. PKC δ induces phosphorylation of Drp1. Additionally, PKC δ inhibitor (Go 6983) or PKC δ siRNA reversed phosphorylation of Drp1 (PMID: 35490835). So, the effect of PKC activation or inhibition on Drp1 phosphorylation (especially at S579 which is believed to activate Drp1) should be examined. Notably, phosphorylation of Drp1 at S693 by GSK3 negatively affects Drp1 function (PMID: 23185298). If this hypothesis about Drp1 as the target of PKC is true, there need not be any direct involvement of contact sites, which may be coincidental rather than causal.

[Response] We explored the effect of PKC activation on DNM1L/Drp1 and the effects of mutations suggested by the reviewer. First, we established an assay by creating DNM1L KO cells (Figure S7E). DNM1L KO cells have elongated and sometimes beaded peroxisomes which can be restored to normal multiple small peroxisomes by complementation of wildtype DNM1L-mCherry (Figure S7F). We assumed that during short-term overexpression of DNM1L peroxisome restoration is mainly due to division of elongated peroxisomes, in this case inactive DNM1L would lead to a decrease in restoration when overexpressed in KO background. We examined the mutations that were recommended in human DNM1L - S616 corresponding to the mouse S579 recommended by the Reviewer and 693A (in transcript-219 or 590 and 667 in transcript-206 of DNM1L that we used for the paper). Phosphomimic mutations of human DNM1L as well as non-phosphorylated forms restored peroxisomes similarly to the wildtype (Figure S8A-B). Mutation 590/616 (corresponding to the 579 residue that activates PKC in mouse DNM1L) restored peroxisomes to the same extent and restoration was not dependent on PKC inhibition (Figure S8C)

Additionally, we measured mitochondrial fission during PMA treatment that increases the number of peroxisomes and didn't observe mitochondrial division (DNM1L activation would lead to an increase in mitochondrial fission) (Figure S7D).

From these results we conclude that PKC acts upstream of peroxisome division factors. To confirm its effect on elongation of peroxisomes we used MFF KO. MFF KO has elongated peroxisome structures. Treating MFF KO with PKC inhibitors leads to a reduction in the length of peroxisomes, indicating that PKC regulates earlier steps preceding division (Figure 3I-J). Since steps earlier than MFF are preceding DNM1L we think that the DNM1L phosphorylation by PKC (which is not disputed, and we also see the interaction of PKC with DNM1L in a different study) is not the molecular driver of peroxisome biogenesis during PMA activation.

3. One concern in the data interpretation regarding the direct importance of ER-peroxisome contacts is that the VAP KO is used. VAPA/B make contacts between the ER and many organelles, making it challenging to assign the entire phenotype solely to the loss of ER-peroxisome contacts. It would be much more informative to knock out ACBD5, which should affect mostly peroxisome contacts. Since the effect of PKC activation or inhibition on VAP contacts with other organelles was not measured, it would be an overinterpretation to assign the peroxisome abundance phenotypes solely to ER-peroxisome contacts. Stronger evidence (e.g. ACBD5 KO) should be used to implicate primarily peroxisome-ER contacts.

[Response] We think that although VAPA/B KO cell model has its limitation, it still shows that the contacts (between ER and peroxisomes or other organelles tethered to ER through VAPs) have direct effect on peroxisome biogenesis. ACBD5 KO does not lead to a decrease in peroxisome numbers and has 50% of contacts remaining (shown by Bishop et al. [10.1177/2515256419848641](https://doi.org/10.1177/2515256419848641)), therefore it wouldn't recapitulate peroxisome detachment to the same extent as VAP KO. We also didn't find a decrease in peroxisome number in ACBD5 KO in HEK293T cells (Figure S1J). As per reviewer suggestion we added the limitation to the discussion, pointing towards the VAP regulation of other contact sites.

Minor

1. The quantitation upon which this screen is based is poorly described and possibly limited. It is unclear how the cell segmentation was performed and whether Z-stack images were converted to MIPs for the analyses. It would also be useful to know what a given peroxisome/area means - is this the mean or median for a cell or a population treated a certain way? These details are important for reproducibility.

[Response] We now include the information in the method section. The screen was performed using confocal 2D images, as 3D images were increasingly hard to process for the number of peroxisomes due to overlaps between peroxisomes in the maximum intensity projections, especially when the number of peroxisomes increased. Processing confocal images was done using Fiji software, peroxisomes were identified using Find Maxima plugin after assigning a threshold level using Threshold function based on the control sample (consistent among all the processed images). The area represents a cell cytoplasmic space which was outlined manually

guided by the Threshold function, nuclei were not included in the areas. We added the details into the method section – Peroxisome abundance measurements.

2. In general, the experimental details are very sparse, impacting reproducibility. All drug concentration used, as well as antibody dilutions and promoters used for overexpression of PKC isoforms and PEX genes should be specified consistently.

[Response] We now included the data on drug concentrations and biological repeats in the legends, antibody dilutions in the method section, and promoters for all the plasmids in the plasmid construction section.

3. In yeast, GFP-tagging of Pex1 blocks its function in peroxisome division but not is peroxisomal location. Were any experiments done (e.g. comparison of peroxisome abundance in WT versus complemented PEX11 KO cells) to show that myc-PEX11 is fully functional because if not, both WT and mutants may be insensitive to PKC modulation.

[Response] We now included an experiment showing that complementation of PEX11b KO with myc-PEX11b plasmid restores peroxisome number, which indicates that PEX11b tagged with myc is functioning as peroxisome division factor (Figure S7C).

May 16, 2025

RE: JCB Manuscript #202505040T

Triana Amen
University of Southampton

Dear Dr. Amen,

Thank you for submitting your revised manuscript entitled "Protein Kinase C positively regulates peroxisome biogenesis and promotes peroxisome-endoplasmic reticulum interaction." We would be happy to publish your paper as a Report in JCB pending final revisions necessary to meet our formatting guidelines (see details below).

A. MANUSCRIPT ORGANIZATION AND FORMATTING:

1) Text limits: Character count for Reports is < 20,000, not including spaces. Count includes title page, abstract, introduction, results & discussion, and acknowledgments. Count does not include materials and methods, figure legends, references, tables, or supplemental legends.

**** Reports must have a combined 'Results and Discussion' section. ****

2) Figure formatting: Reports may have up to 5 main text figures. Scale bars must be present on all microscopy images, including inset magnifications. Molecular weight or nucleic acid size markers must be included on all gel electrophoresis. Also, please avoid pairing red and green for images and graphs to ensure legibility for color-blind readers. If red and green are paired for images, please ensure that the particular red and green hues used in micrographs are distinctive with any of the colorblind types. If not, please modify colors accordingly or provide separate images of the individual channels.

3) Statistical analysis: Error bars on graphic representations of numerical data must be clearly described in the figure legend. The number of independent data points (n) represented in a graph must be indicated in the legend. Please indicate whether 'n' refers to technical or biological replicates (i.e. number of analyzed cells, samples or animals, number of independent experiments). If independent experiments with multiple biological replicates have been performed, we recommend using distribution-reproducibility SuperPlots (please see Lord et al., JCB 2020) to better display the distribution of the entire dataset, and report statistics (such as means, error bars, and P values) that address the reproducibility of the findings.

Statistical methods should be explained in full in the materials and methods. For figures presenting pooled data the statistical measure should be defined in the figure legends. Please also be sure to indicate the statistical tests used in each of your experiments (both in the figure legend itself and in a separate methods section) as well as the parameters of the test (for example, if you ran a t-test, please indicate if it was one- or two-sided, etc.). Also, if you used parametric tests, please indicate if the data distribution was tested for normality (and if so, how). If not, you must state something to the effect that "Data distribution was assumed to be normal but this was not formally tested."

4) Abstract and title: The summary should be no longer than 160 words and should communicate the significance of the paper for a general audience. The title should be less than 100 characters including spaces. Make the title concise but accessible to a general readership. We suggest shortening the title to either: "PKC regulates peroxisome biogenesis and promotes peroxisome-endoplasmic reticulum interaction" or "Protein Kinase C promotes peroxisome biogenesis and peroxisome-endoplasmic reticulum interaction".

5) Materials and methods: Should be comprehensive and not simply reference a previous publication for details on how an experiment was performed. Please provide full descriptions (at least in brief) in the text for readers who may not have access to referenced manuscripts. The text should not refer to methods "...as previously described." Please indicate the type of membrane used for immunoblotting as well as describe acquisition and quantification methods.

6) For all cell lines, vectors, strains, constructs/cDNAs, etc. - all genetic material: please include database / vendor ID (e.g. Addgene, ATCC, etc.) or if unavailable, please briefly describe their basic genetic features, even if described in other published work or gifted to you by other investigators (and provide references where appropriate). Please be sure to provide the sequences for all of your oligos: primers, si/shRNA, RNAi, gRNAs, etc. in the materials and methods. You must also indicate in

the methods the source, species, and catalog numbers/vendor identifiers (where appropriate) for all of your antibodies, including secondary. If antibodies are not commercial, please add a reference citation if possible.

7) Microscope image acquisition: The following information must be provided about the acquisition and processing of images:

- a. Make and model of microscope
- b. Type, magnification, and numerical aperture of the objective lenses
- c. Temperature
- d. Imaging medium
- e. Fluorochromes
- f. Camera make and model
- g. Acquisition software
- h. Any software used for image processing subsequent to data acquisition. Please include details and types of operations involved (e.g., type of deconvolution, 3D reconstitutions, surface or volume rendering, gamma adjustments, etc.).

8) References: There is no limit to the number of references cited in a manuscript. References should be cited parenthetically in the text by author and year of publication. Abbreviate the names of journals according to PubMed.

9) Supplemental materials: Reports generally may have 3 supplemental figures and 10 videos. You currently exceed this limit but, in this case, we will be able to give you the extra space but if possible please try to consolidate the current supplemental figures. Tables, like figures, should be provided as individual, editable files. A summary of all supplemental material should appear at the end of the Materials and methods section. Please include one brief sentence per item. Unropped gels/blots should not be included as supplemental material but submitted separately as Source Data (see pt#15 below).

10) eTOC summary: A ~40-50 word summary that describes the context and significance of the findings for a general readership should be included on the title page. The statement should be written in the present tense and refer to the work in the third person. It should begin with "First author name(s) et al..." to match our preferred style.

11) Conflict of interest statement: JCB requires inclusion of a statement in the acknowledgements regarding competing financial interests. If no competing financial interests exist, please include the following statement: "The authors declare no competing financial interests." If competing interests are declared, please follow your statement of these competing interests with the following statement: "The authors declare no further competing financial interests."

12) A separate author contribution section is required following the Acknowledgments in all research manuscripts. All authors should be mentioned and designated by their first and middle initials and full surnames. We encourage use of the CRediT nomenclature (<https://casrai.org/credit/>).

13) ORCID IDs: ORCID IDs are unique identifiers allowing researchers to create a record of their various scholarly contributions in a single place. Please note that ORCID IDs are required for all authors. At resubmission of your final files, please be sure to provide your ORCID ID and those of all co-authors.

15) JCB requires authors to submit Source Data used to generate figures containing gels and Western blots with all revised manuscripts. This Source Data consists of fully uncropped and unprocessed images for each gel/blot displayed in the main and supplemental figures. For assays performed using capillary electrophoresis and/or immunoassay-based detection, authors should instead provide the electropherogram graph(s) for each experiment, plotting fluorescence/chemiluminescence intensity vs. molecular weight/size. Since your paper includes cropped gel and/or blot images, please be sure to provide one Source Data file for each figure gels, blots, and/or capillary electrophoresis assays along with your revised manuscript files. File names for Source Data figures should be alphanumeric without any spaces or special characters (i.e., SourceDataF#, where F# refers to the associated main figure number or SourceDataFS# for those associated with Supplementary figures). For traditional gels and blots, the lanes of the gels/blots should be labeled as they are in the associated figure, the place where cropping was applied should be marked (with a box), and molecular weight/size standards should be labeled wherever possible. For capillary electrophoresis assays, each trace in the graph should be color-coded and labeled to indicate which protein, gene, or sample is being measured (please try to avoid red/green combinations to accommodate our color-blind readers).

Source Data files will be directly linked to specific figures in the published article. Source Data Figures should be provided as individual PDF files (one file per figure). Authors should endeavor to retain a minimum resolution of 300 dpi or pixels per inch. Please review our instructions for export from Photoshop, Illustrator, and PowerPoint here: <https://rupress.org/jcb/pages/submission-guidelines#revised>

16) Journal of Cell Biology now requires a data availability statement for all research article submissions. These statements will be published in the article directly above the Acknowledgments. The statement should address all data underlying the research presented in the manuscript. Please visit the JCB instructions for authors for guidelines and examples of statements at (<https://rupress.org/jcb/pages/editorial-policies#data-availability-statement>).

B. FINAL FILES:

****It is JCB policy that if requested, original data images must be made available to the editors. Failure to provide original images upon request will result in unavoidable delays in publication. Please ensure that you have access to all original data images prior to final submission.****

****The license to publish form must be signed before your manuscript can be sent to production. A link to the electronic license to publish form will be sent to the corresponding author only. Please take a moment to check your funder requirements before choosing the appropriate license.****

Thank you for your attention to these final processing requirements. Please revise and format the manuscript and upload materials within 14 days. If you need an extension for whatever reason, please let us know and we can work with you to determine a suitable revision period.

Thank you for this interesting contribution, we look forward to publishing your paper in Journal of Cell Biology.

Sincerely,

John Aitchison, PhD
Monitoring Editor
Journal of Cell Biology

Dan Simon, PhD
Scientific Editor
Journal of Cell Biology

Full Revision

Manuscript number: RC- 2025-02877R

Corresponding author(s): Daniel, Kaganovich; Triana, Amen

1. General Statements [optional]

We thank all the reviewers for their constructive comments.

Reviewer #1 (Evidence, reproducibility and clarity (Required)):

Using a compound kinase inhibitor screen, Borisyuk et al. identified Protein Kinase C inhibitors which reduce the number of peroxisomes in human HEK cells, fibroblasts and neurons. To validate the peroxisome-specific regulatory effects of PKC they overexpressed several isoforms of PKC and found that only one overexpressed isoform PKCdelta could trigger peroxisome proliferation. Moreover, incubation of cells with known PKC activating compounds, e.g. phorbol esters (PMA), resulted in increasing number of peroxisomes per cell. Preliminary data let the authors suggest that PKC regulation does neither activate the transcriptional regulator PPARalpha nor inhibits selective peroxisomal autophagic degradation dependent of the NBR receptor. A role for PKC to regulate the division of mature peroxisomes is indicated by the observation that in the absence of the division factor PEX11b the abundance of peroxisomes was not affected by PKC inhibition. The data further suggests that PEX11b is not the phosphorylation target of the PKC signalling pathway. A good candidate for phosphorylation is glycogen synthase kinase beta (GSK3 β) which negatively regulates the contact of peroxisomes and ER via ACBD5/VAPB tethering (as recently shown by the Schrader group). Finally, the authors show the existence of the novel transduction cascade in differentiating neurons in culture.

The validation of PKC and identification of other constituents of the regulation of peroxisome proliferation is largely carried out using inhibitors or activators controlling a large variety of cellular responses. Several of the main conclusions based on the application of inhibitors and activators should be supported by additional experiments which directly target the PKCdelta isoform.

[Response] We thank this reviewer for their constructive comments. Please see the point-by-point responses below.

Major concern:

The suggested additional control experiments should be considered to support the following conclusions.

1. Protein Kinase C delta positively regulates peroxisome abundance

After the inhibitor screening, the inhibited kinase of the PKC family was identified. To this end, they overexpressed several kinases of this family but only the PKCdelta isoform shows an

increase in the number of peroxisomes (Fig.2). The next logical question would be whether the identified PKCdelta isoform interferes with peroxisome proliferation only indirectly or in combination with other kinases. To analyze this, the authors should downregulate the identified kinase isoform using siRNA and, as with the inhibitor, see a reduction in peroxisome number.

[Response] To address whether PKCdelta isoform is responsible for the phenotypes that we observed we constructed a CRISPR/Cas9 knockout of PKCdelta in HEK293T cells (Figure S11). We observed no significant decrease in the number of peroxisomes in the KO compared to WT cells, ruling out PKCdelta as the only regulator of peroxisome number (Figure S11). It is important to note that human cells express many isoforms of PKC with reported overlapping roles, therefore it would necessitate a detailed inquiry into what isoforms or their combinations maybe contributing to the effects we observed. We intend to explore it in the future work, and we included an experiment on the expression of different PKC isoforms in neuronal differentiation (see below).

2. Protein Kinase C induces PEX11b-dependent peroxisome formation

The proliferation of peroxisomes is PEX11b-dependent. The authors observed no further reduction in the number of peroxisomes in the absence of PEX11b and incubation with a PKC inhibitor (Fig.3) From this they conclude that PEX11b-dependent peroxisome proliferation is induced by PKC. This may be correct, but in addition they should show that overexpression of PKCdelta in the PEX11b knockout cell line does not lead to an increase in peroxisomes.

[Response] We overexpressed PKCD in PEX11bKO cells and observed no increase in peroxisome number, indicating that PKCD regulation requires PEX11b-dependent peroxisome division (Figure S4A).

3. PKC regulates peroxisome-ER contact sites through GSK3 β inhibition

The authors confirm previous studies that a lack of ER-peroxisome contact leads to a reduced number of peroxisomes. Their data (Fig 4C-F) suggest that this VAP-dependent interaction is regulated by PKC. In their model, PKC inactivated with inhibitors (Go6983) triggers an active form of GSK β , which in turn detaches peroxisomes from the ER. In fact, the inactivating phosphorylation of GSK3 β is suppressed by Go6983 and promoted by PMA (Fig. 4G). Again, siRNA knockdown or and overexpression of PKCdelta would be more appropriate than using the inhibitors and activators, both of which can potentially cause numerous indirect effects.

[Response] As suggested, we measured the interaction between ACBD5 and VAPB using co-immunoprecipitation in the presence of PKCD by overexpressing PKCD-mCherry. We found increased interaction between ACBD5 and VAPB when overexpressing PKCD, confirming our finding that PKC regulates peroxisome-ER contact sites (Figure 4G).

4. Protein kinase C regulates peroxisome abundance during neuronal differentiation

In the last section, they show that the general PKC inhibitor Go6983 reduces the number of peroxisomes in cell culture, albeit only slightly (Fig.5 E,F), and thus possibly also influences neuronal differentiation in humans. In this experimental setup, it is very difficult to show the specific effect of the PKCdelta isoform by siRNA-induced inactivation, but I would like to know whether overexpression of PKCdelta in these cultures reverses the effects. If the model shown is correct, another interesting question would be whether the expression of this isoform is upregulated in neuronal cells upon differentiation (see also minor concerns, point 3). For the

Full Revision

correlation of the model with neuronal differentiation, it would also be interesting to see whether specific GSK3 inhibitors, as shown for HEK cells (Fig. S1G), have a neuronal effect.

[Response] We attempted to overexpress PKCD-mCherry throughout neuronal differentiation. However, we didn't find neuronal cells overexpressing PKCD on day 18. This could be due to the toxicity of high levels of PKCD over long periods of time. In the future we will attempt to create a PKCD KO in neuronal cells with the validated constructs to confirm these findings. To further confirm the importance of PKC isoforms in neuronal differentiation we did an unbiased qPCR analysis of PKC expression during neuronal differentiation (Figure S7C). We found that most of the novel PKCs were upregulated, including PKCdelta, PKCeta and PKCtheta . We will explore whether other novel isoforms besides PKCdelta regulate peroxisome number in the future work.

Minor points:

1. Why were overexpression of PEX19 and PEX3 used as positive and negative controls, respectively, in the inhibitor screen? I was not aware that overexpression of PEX3 in human cell lines leads to a reduction in peroxisome numbers. As far as I know, the cited references refer to yeast experiments, i.e. Höhfeld et al. In addition, the controls used in Figure Fig 1 D,E are missing.

[Response] PEX3 was shown to lead to peroxisome degradation when overexpressed in mammalian cells by Yamashita et al. (doi: [10.4161/auto.29329](https://doi.org/10.4161/auto.29329)) due to peroxisome degradation, therefore we used it as a negative control. We didn't find a reference for overexpression of PEX19 in human cells with quantification of peroxisome number. In our hands it consistently led to increase in peroxisome number, we sited yeast work in which biogenesis factor was first isolated and shown to affect peroxisome number.

We didn't use overexpression in Figures 1D, E as transfection of primary human fibroblasts is very difficult and does not lead to a number of cells required for statistical analysis. In these figures we compared peroxisome number to an untreated control condition and scored significant increase or decrease in peroxisome number.

2. Page 4, line 35 the authors wrote: To also rule out a role for pexophagy, we silenced the NBR1 receptor, which did not affect peroxisome abundance over a two day period (Figure S4F, S4G), confirming that PKC does not upregulate peroxisomes by inhibiting their degradation. (Fig S4 F,G). The essential importance of this receptor for the selective degradation of peroxisomes has been shown previously (Ref 89). To understand the authors' conclusion, one should know under which conditions this experiment was performed, i.e. with or without Go6983 inhibitor ?

[Response] This experiment was performed in control conditions. The idea was that if PKC acts through pexophagy then pexophagy regulation alone should show significant increases or decreases in peroxisome number, which we didn't see in control conditions. In this revision we used silencing of NBR1 and NIX (another pexophagy receptor indicated to by Reviewer 3, PMID: 36810161) in both control and PKC activation conditions. We didn't find significant differences when pexophagy was inhibited in these conditions (similar upregulation of peroxisomes in PKC activation) from which we concluded that PKC does not act through pexophagy regulation (Figure S4G).

Full Revision

3. The authors postulate that PKC activity is particularly high in the rat brain. They refer to a reference 101, (Kikkawa et al), which dates back to 1982 and examines the enzyme localization in the rat brain. The reference on the expression of different PKC isoforms, especially PKCdelta, should be updated.

[Response] We updated the reference on PKC in the brain, by adding references that refer to PKCdelta neuronal populations in the brain that control feeding, alcohol use, and fear (doi.org/10.1038/nn.3767, <https://www.nature.com/articles/nature09553>, <https://www.science.org/doi/10.1126/sciadv.abg9045>, PMID: [20691763](https://pubmed.ncbi.nlm.nih.gov/20691763/)). Since there is lack of comparative expression studies of PKC expression during neuronal development, we decided to analyze PKC expression in SH-SY5Y cells that we are using for neuronal differentiation experiments. Real-time qPCR confirmed upregulation of several PKC isoforms when comparing non-differentiated and 18 day differentiated cultures (Figure S7C), including PKC delta, eta, and theta. These results indicate that other novel isoforms may also be contributing to the increased PKC activity and peroxisome proliferation, a subject for further studies.

4. To better understand the results of the inhibitor screen and to use these as an open data source, a supplemental table should be generated containing name of the compounds, their specificity as well as peroxisome count.

[Response] We include an updated table with the name of the compound, compound specificity to kinases, and peroxisome counts (Supplementary Table 1: Kinase Inhibitor Screen).

Reviewer #1 (Significance (Required)):

The approach to identify the kinases involved in peroxisome biogenesis is novel and straight forward. The presentation of the data as well as the accompanying text are very clear and suitable to arouse the interest of a large readership. The original data set of the screening (see also Minor point 4) can be used as a data source in future work.

The data are of great interest not only for specialists in the field. The identification of PKC significantly contributes to our understanding of the regulation of peroxisome biogenesis.

[Response] We thank this reviewer for their positive assessment of our work.

Reviewer #2 (Evidence, reproducibility and clarity (Required)):

Summary:

This paper from Dr. Amen and colleagues uses a kinase inhibitor screen to identify potential regulators of peroxisome abundance in mammalian cells. They find several promising hits in the screen some of which link to the kinase PKC, a multifunctional kinase involved in a huge number of cellular processes. After exploring the possible mechanisms by which PKC might be regulating peroxisome abundance they eventually link PKC activity to recent work showing that

Full Revision

GSK3-beta can regulate peroxisome connections to the ER. They also found a direct GSK3-beta inhibitor in the screen which further supports this concept. As GSK3-beta is a known PKC target this nicely links these two processes and solidifies/confirms some of the previous work in this area, allowing PKC to now be placed in a hypothetical model upstream of GSK3-beta in the pathway. Other work in the paper seems to rule out a link between PKC and pexophagy, activation of peroxisomal genes and modulation of the peroxisome biogenesis factor Pex11. To show a role for PKC they then link this to cell differentiation, when both peroxisomes and PKC activity are known to be increased and provide data suggesting that the peroxisomal increase during differentiation is dependent on PKC activity.

[Response] We thank this reviewer for their thoughtful and constructive comments and suggestions. We addressed all the comments below.

Major comments:

- Are the key conclusions convincing?

The conclusion that modulation of PKC alters peroxisome abundance is convincing. The conclusion that PKC acts upstream of GSK3-beta to regulate the ACBD5-VAPB axis and thus regulate peroxisome-ER connections is convincing. That this is then directly linked to peroxisome abundance changes is still somewhat less convincing but could be built upon. The conclusion that the increase in peroxisomal abundance during differentiation is PKC dependent is convincing.

The other data surrounding these key points is less convincing and some of the experiments around Pex11 beta are a little confusingly presented, their relevance is less clear and at times this element detract from the other key conclusions.

[Response] In light of the comments regarding the lack of evidence that the PKC regulation of peroxisome-ER contact sites promotes peroxisome division we restructured the manuscript. – We think that more evidence and a direct investigation is required to link contact sites and peroxisome number. Although it is likely that contact sites are required to provide the membrane for the subsequent growth and division of peroxisome, this paper does not intend to bridge this knowledge gap. Showing that modulation of PKC alters peroxisome number and regulates peroxisome-ER contact site is what all reviewers thought as convincing evidence. Therefore, we changed the title to reflect that. We also put PEX11b data in supplementary (former Figure 3I and J) as it is potentially interesting but does not add to the conclusions, while leaving the negative PEX11b results in the manuscript.

- Should the authors qualify some of their claims as preliminary or speculative, or remove them altogether?

The narrative that the authors have around Pex11 is confusing, as none of the data they present seems to show any clear link to Pex11.

"We established a mechanistic connection between PKC activation and peroxisome biogenesis through PEX11b peroxisome formation".

Full Revision

This does not actually seem to fit your data. Your data fits with regulation of peroxisome-ER tethering regulation via GSK3-beta being potentially linked to peroxisome biogenesis and that PKC acts upstream of GSK3-beta in this pathway. In the model the Pex11 part of the pathway is downstream from this event and you don't show any link to Pex11 in any of your data - in fact the opposite - it seems to suggest Pex11 is not involved. The fact that in the Pex11 KO you do not see a change when you inhibit PKC would also fit with this. Pex11 is essential for this process and it is downstream of the PKC-GSK3-Beta-ACBD5-VAP part of the pathway. So perhaps no matter what you do with PKC, Pex11 will still be critical for peroxisome biogenesis. I can see that a huge amount of work has been invested in the Pex11 link, so to remove it entirely seems unnecessary, but I think this aspect makes the current manuscript confusing and so it needs to be clarified.

[Response] We agree with the Reviewer. We now moved the majority of PEX11b experiments to supplementary figures and renamed the result section which now focuses on PEX11b and DNM1L (see below).

- Would additional experiments be essential to support the claims of the paper?

If the main claims of the paper are clearly defined as I have set out above then these would be mostly supported by the current data.

If you want to really show that Pex11 beta is downstream of the PKC-GSK3-beta-ACBD5-VAPB step in the mechanism and make this point, you could consider the following:

The main phenotype of Pex11beta KO cells is elongated peroxisomes with a slightly reduced number (see example here <https://doi.org/10.3389/fcell.2020.577637>) which you presumably see in your own Pex11beta KO cell line. You also show a slightly reduced abundance in Figure 3G when comparing WT with Pex11 KO cells but do not discuss elongation. These elongations could potentially be due to membrane lipids still being provided by the ER, allowing initial peroxisome elongation prior to division but then a defect in the Pex11-driven part of the process (membrane deformation, interaction with MFF/Fis1, DLP1 activation etc) resulting in some elongation but inefficient division. Hence, a smaller number of elongated peroxisomes are observed. If PKC is driving interaction between ACBD5-VAPB, generating PO-ER contacts, allowing membrane growth of peroxisomes, upstream of Pex11beta. Then if you inhibit PKC you might expect to see less elongations in the Pex11 beta KO cells. The same number, but less elongated structures. So you could assess this.

[Response] We analyzed peroxisome size (area in square microns) in WT and PEX11b KO cells in control and PKC inhibition conditions. PKC inhibition resulted in a significant increase in the size of peroxisomes in both WT and PEX11b KO. We cannot explain the results of the increase, however in VAPA/B KO cells that lost contact sites we also see a significant enlargement of peroxisomes. More experiments and super-resolution techniques are needed to understand this phenomenon and to resolve potential clustering issues. Therefore, we did not include this experiment in the paper to avoid more confusion, but we insert it here for the reviewer:

Figure for the reviewer. Graph shows the size of peroxisomes in HEK293T WT and PEX11b KO cells in control and PKC inhibition conditions (Go6983, 10uM, 24h), mean± SEM, N=5-10 biological repeats, each biological repeat is an average of peroxisome size in 100 cells , **** - p<0.0001, *** - p<0.001, ns – nonsignificant.

To clarify the issue of whether PKC acts upstream peroxisome division, we performed a similar experiment using MFF KO. MFF KO has significantly elongated peroxisomes. We quantified the length of the organelle in the control and PKC inhibition conditions and found a significant reduction in the elongation when PKC is inhibited (Figure 3I, J). We also performed several experiments on the DNM1L KO complemented with phosphomutants of DNM1L in different conditions (see Figures S7, S8 and comments for Reviewer 3). However, we didn't find the difference when comparing control and PKC modulation, or phosphomutants, indicating that PKC acts upstream of peroxisome division.

Additionally, we added an experiment showing that overexpression of PKCD increases the contact site formation assessed by co-IP of VAPB-ACBD5 (Figure 4G). Together the data points toward contact site regulation by PKC, rather than regulation of peroxisome division.

- Are the suggested experiments realistic in terms of time and resources?

The authors already have all the tools and may even have the images to be able to do this analysis.

- Are the data and the methods presented in such a way that they can be reproduced?

Yes, in general data are well presented with clear schematics.

- Are the experiments adequately replicated and statistical analysis adequate?

Yes, in general. However, one point to consider is that throughout the paper, N numbers are

Full Revision

used, based on number of cells counted yet the number of actual experimental repeats is unclear. Should N be number of areas or number of experimental repeats? How many biological repeats has this come from? If none then any experimental variation (which can be considerable) is not being considered and could leave to false positives. Consider with reference to this paper: <https://doi.org/10.1083/jcb.202401074>

[Response] We performed 3 biological repeats for each experiment, and the number of areas or cells was pooled from all the biological repeats. We added biological repeats information into the statistical analysis section, and supplementary table 3 – Statistical analysis. We also indicated in legends all the biological repeats to avoid confusion.

Minor comments:

- Specific experimental issues that are easily addressable.

To further convince the reader that the impacts you observe are not PMP70 specific, you could consider to another peroxisome marker to backup PMP70 work. This could be Pex11 - as you have also have a Pex11-FP reporter line used in Fig 3E.

[Response] In revision we added other protein and non-protein markers including PEX14 and PeroxiSPY (Figures S4A, S4G, S7, S8) in addition to PEX11b used in Fig.3E.

- Are prior studies referenced appropriately?

In general yes but at times the references cited to evidence specific points do not clearly justify or explain these points or the evidence for your point is not cited.

E.g. Page 6 line 43

Why do you reference (117) the paper here from the Nunnari lab on ER-mitochondria contact sites. What does this imply?

[Response] This reference was meant as an example of how the known organelle contact can promote organelle division, evidencing the possibility that these two mechanisms can be linked experimentally. Since the sentence was meant as a future research direction, we removed this reference.

E.g. "metabolic control of peroxisome abundance (MCPA)"

Where has this come from - it is not clear from the references gives here (23, 60-63). Please give the reader a little more help and evidence here as from the citations you have given it is hard to know what to make of this point. Consider rephrasing, re-citing or giving a bit more context on how the data from these papers leads to this MCPA - it is not clear how some of them are related. 23- The Chang et al 1999 paper shows altered peroxisome number/morphology in beta oxidation mutant but this is not the same as a regulatory system. 60 - is based on rodent PPARs, 61 is a very old review on PPARs, 62 is about the discovery of the Pex11 gene, 63 - a review on membrane contact sites, the relevance to this point is not clear?

Full Revision

[Response] Thank you for indicating that. We updated the text, removing the sentence and rearranging the paragraph (see below).

E.g. "Both peroxisome biogenesis mechanisms are initiated by several non-mutually exclusive pathways: transcriptional peroxisome proliferator receptor alpha (PPAR α) increases the level of division factors, peroxisome tethering to the endoplasmic reticulum (ER) through a recently defined membrane contact site that is formed between acyl-coenzyme A-binding domain protein 5 (ACBD5) and the ER protein vesicle-associated membrane protein-associated protein (VAPB)58, 59 46 47 potentially sources the membrane, and an increase in peroxisome function in the presence of 48 peroxisomal substrates promotes peroxisome formation via unknown mechanisms."

This sentence is very long and difficult to follow and citation use is confusing. We suggest you rephrase.

[Response] We rephrased and split it into several sentences to improve readability. The references were updated to indicate specific findings of PPAR α role in upregulating PEX11a, ACOX1, ABCD2 and 3 ((doi.org/10.1371/journal.pone.0006796, doi: 10.1155/2010/612089). We also added a citation for peroxisomal substrates increasing peroxisome abundance ([10.1016/j.devcel.2024.06.020](https://doi.org/10.1016/j.devcel.2024.06.020)) and how PEX11b and to a lesser extent PEX11a regulate peroxisome formation (doi.org/10.1074/jbc.273.45.29607).

E.g. "transcriptional peroxisome proliferator receptor alpha (PPAR α) increases the level of division factors"

As this is not so widely known for nons-specialists can you give some more details here, cite the evidence for this in human cells? Which division factors?

[Response] We added appropriate references for the upregulation of PEX11a by PPAR α in human and rodent cells (doi.org/10.1371/journal.pone.0006796, doi: 10.1155/2010/612089)

E.g. "and an increase in peroxisome function in the presence of peroxisomal substrates promotes peroxisome formation via unknown mechanisms".

Again, as above, this is not completely clear so please cite here the evidence for this in human cells, and ideally provide some examples of this.

[Response] We rephrased this section and added citation how very long-chain fatty acids can promote peroxisome abundance through PPAR response ([10.1016/j.devcel.2024.06.020](https://doi.org/10.1016/j.devcel.2024.06.020)).

E.g. "The regulation of peroxisome abundance by pexophagy may be more relevant for clearance of peroxisomes, when excess peroxisomes are degraded, usually after a bout of upregulation 46, 81, 82, 89, 116"

I think this is an interesting idea but is there really any evidence for this in these studies? Or is it more likely pexophagy is usually employed during wider autophagy response, for overall cell survival in response to stress?

[Response] Thank you for indicating that, indeed it would be very relevant for starvation responses. We modified the sentence to reflect that. All the textual modifications are highlighted in the manuscript.

Full Revision

- Do you have suggestions that would help the authors improve the presentation of their data and conclusions?

General points

Intro - needs a bit more specific detail and examples as well as some clarity in some of the sections to explain the rationale for the experiments and what is already known from the literature. In the Introduction - it would be good for the reader to have a very clear and specific example in human cells of the "transient "on demand" need for peroxisomal metabolism and specifically biogenesis. This is very obvious in various yeast, rodents and other organisms but potentially less clear in humans.

[Response] We now added a specific example of transient reversible increase in peroxisome number upon ethanol treatment in liver cells and tissues (<https://doi.org/10.1091/mbc.E24-06-0252>).

I also suggest a thorough proofread - quite a few mistakes and sentences which could be more coherent. E.g. Slight typo on page 2 line 42 - 53-5, Page 3 - line 6 - typo. Figure S1 legend. Line 12 and 13 - should be G and F?

[Response] Thank you for indicating that. We corrected the mistakes and proofread the text.

Please detail statistical analyses and experimental number used in the figure legends.

[Response] We added statistical analyses in the figure legends. Full information is in the Supplementary Table – Statistical Analysis.

Specific minor points:

Please explain briefly why Pex19 and Pex3 were used as controls in the screen and how this works, this is unclear for the non-specialist. Please specify what was the cut-off for the screen for compounds which altered PO density, just statistically significantly different?

[Response] PEX3 was indicated to lead to peroxisome degradation when overexpressed in mammalian cells by Yamashita et al. (doi: [10.4161/auto.29329](https://doi.org/10.4161/auto.29329)) due to peroxisome degradation, therefore we used it as a negative control. We didn't find a reference for overexpression of PEX19 in human cells, in our hands it consistently led to increase in peroxisome number, we cited yeast work in which biogenesis factor was first isolated and shown to affect peroxisome number. We added an explanation in the Results section.

We used statistical significance to determine the "hits" from the screen. We included a table with all the values and kinase information in supplementary (table 3).

Figure 3 - Why have PMP70 levels not been restored by Day 4 of the complementation assay (Fig3C?). Please comment

[Response] We used transient transfection in our experiments. The blot shows lysates that contain a mix of transfected and untransfected cells. The low restoration reflects the transfection efficiency combined with outgrowing of cells for days after transfection. We think that in transfected cells PMP70 levels were restored fully as visible in the images (Figure 3D).

Full Revision

Page 4 line 22 - this is a potentially important result. What is the anticipated/known time for peroxisome division - how does this relate to your 2 hours for PKC activation (Fig 3E). Can you first see spherical and then elongated peroxisomes when you do this?

Explain with a little bit more detail the de novo assay for the non-specialist.

[Response] We saw full restoration of peroxisomes in complemented cells over the course of 4 days during de novo formation (when peroxisomes form, they likely start dividing). Peroxisome number doubles upon cell division, HEK293T cells divide approximately every 12 hours and shortly after division cells will have a number of peroxisomes comparable to the mother cell. The anticipated time for peroxisome division is on a several hours scale, in our movies we occasionally observe the intermediate steps of peroxisome division – elongated peroxisomes, beads on a string -like phenotype with several small peroxisomes positioned one after the other, however all these intermediates are rare.

We added more detail on the de novo assay in the results section.

Page 4 line 29 "we constructed a CRISPR/Cas9 PEX11b KO" in our GFP-PMP70 HEK293 background (I assume)?

[Response] Yes, we added this clarification.

"The ratio of phosphorylated to the overall identified peptides was 4.4% for Ser53". Can the authors explain this?

[Response] A rather low ratio means that only ~4 copies out of 100 were phosphorylated (over 50% for S38, Figure S5B). This could mean not optimal conditions (phosphorylation event happened earlier followed by dephosphorylation or phosphorylation event is just starting, or incomplete activation of PKC), however it becomes irrelevant to further optimize these conditions knowing that phosphorylation on this residue is not relevant for the phenotypes we study (based on the complementation experiments).

The peptide competition assay needs to be explained more clearly. It is hard to follow. This is an in-vitro assay with purified PKC? Using different Pex11 peptides and also a known PKC substrate. What are the substrates? Were any non PKC peptides included - do they all have the consensus sequence - if so then what does this tell us?

[Response] Yes, this is an in vitro assay with known PKC substrate peptide as a control and the identified PEX11b peptides (the sequences are in the method section). We used prediction algorithm to identify PKC consensus sequences in PEX11b sequence and identified 6 sequences including S53. We confirmed by identification of PEX11b phosphopeptides in cells (PEX11b pulldown) that one of them S53 is phosphorylated. This shows that S53 is phosphorylated in the conditions we used, however it doesn't show that PKC can directly phosphorylate it. To show that PKC can directly phosphorylate we used S53 and PKC known substrate in the in vitro assay which does not contain any other kinases except for PKC. As a negative control we used no PKC samples, and we also incorporated all other PEX11b PKC substrate peptides, among which several were shown to be phosphorylated in the in vitro assay, and some were not. This data tells us that PKC can directly phosphorylate S53 sequence and

Full Revision

that it is phosphorylated in cells to a low extent. We then show that this phosphorylation event doesn't regulate peroxisome number, so we do not know the function of this phosphorylation by PKC. For this paper it is a dead end, however the experiments are of value to further investigation of the role of PEX11b phosphorylation status, therefore we included them in the supplementary figures.

We have now clarified the competition assay; however, we would like to make this section simple as it doesn't add to the main conclusions.

"PEX11b S53D overexpression, and that of the WT and S53A, had no effect on peroxisome number (after correction for PEX11b expression level)"

What does this mean after correction for Pex11b expression level? It is already known that Pex11 overexpression may not impact the number but it will likely result in more elongated peroxisomes. It is a little unclear what the expected outcome of these experiments would be?

[Response] We corrected to the level of PEX11b overexpression as there was a significant reduction in S53A variant when expressed (Figure S5E). We did this experiment to prove whether phosphorylation of PEX11b by PKC on S53 impacts the number of peroxisomes. We didn't know the outcome; the expectation was either to rule out the possibility or not.

Page 5 Line 18 - There is no concluding sentence and so what do we then take from this? You have found some phosphosites on Pex11-beta. Which could in theory be PKC sites as in vitro PKC can phosphorylate this site - but they do not do anything when you mutate them and you do not show they are PKC sites in cells.

[Response] We added a concluding sentence, stating that PKC regulation of peroxisome biogenesis is not through PEX11b.

Overall - data that Pex11 is a substrate for PKC remains unclear. This leads to the reader questioning what to take from this whole section on Pex11. I strongly suggest addressing this in a revised version.

[Response] In the revised version we put all the experiments of PEX11b section in supplementary. Since there is an addition of DNM1L negative results, we combined this into one section.

Figure 4 - Just a comment: In the VAPA/VAPB double KO cells the peroxisome numbers/density may be reduced but there are also very significant changes in their morphology. Due to the wide-ranging roles of the VAPs in multiple different organelle contact sites in these KO cells, you would expect very significant changes in (all?) organelle morphology and wide-scale disruption to lipid transport and membrane lipid content. See here for an example of VAPA/VAPB double KO in endothelial cells. <https://doi.org/10.1016/j.jsbmb.2023.106349>

Is this therefore the best system to investigate changes in organelle morphology, linked to PKC?

[Response] We agree with the reviewer, that it is not fully clear how the absence of other contact sites in VAPA/B KO affects peroxisome biogenesis. However, we think that VAPA/B KO is an essential model since it is a model that completely lacks peroxisome contacts through

Full Revision

VAPs. Most of the experiments supporting the claims were performed in different models. We now added this limitation to the discussion.

Figure 5 - I was surprised that PKC inhibition seems to have no impact on differentiation in your assays? PKC is supposed to be a key player in differentiation and yet its inhibition seems to have no impact on the process. I am not familiar with all the work on this but has a similar experiment not been performed previously? I note the authors only use minimal differentiation markers. Should more markers of differentiation be considered?

[Response] We think that due to a competition of PKC inhibitor with PKC activator we were only able to achieve partial PKC inhibition, which allowed neuronal differentiation. We didn't find similar experiments, however it is known that there are PKC positive and negative neuronal populations, it is possible that during general GABA/dopaminergic differentiation we do not see the differences because the neurons are not reaching a mature state in 18 days, nor are we working with specific populations that particularly highly overexpress PKC (such as PKCdelta positive populations that control alcohol use, fear response, and feeding: doi.org/10.1038/nn.3767, <https://www.nature.com/articles/nature09553>, <https://www.science.org/doi/10.1126/sciadv.abg9045>, PMID: [20691763](https://pubmed.ncbi.nlm.nih.gov/20691763/)).

In the revised version we looked at what isoforms of PKC are expressed during neuronal differentiation and we also included neurofilament as additional marker of differentiation (Figure S9C).

The NBR1 silencing experiment in Fig S4, exploring pexophagy. I was confused by what this actually shows? This seems to go against the data in the Kim paper, albeit a different cell type, in which they showed that 2 days silencing NBR1 caused a big change in pexophagy observed <https://doi.org/10.1242/jcs.114819>. How can the authors explain this?

[Response] The idea of this experiment was that if PKC acts through pexophagy then pexophagy regulation alone should show significant increases or decreases in peroxisome number, which we didn't see in control conditions. In this revision we used silencing of NBR1 and NIX (another pexophagy receptor indicated to by Reviewer 3, PMID: 36810161) in both control and PKC activation conditions. We didn't find significant differences in PKC response when pexophagy was inhibited in these conditions (similar upregulation of peroxisomes in PKC activation) from which we concluded that PKC does not act through pexophagy regulation (Figure S4G). We think that the most likely explanation for not observing an increase in the number of peroxisomes during silencing of the pexophagy receptors is peroxisome clustering that can obscure the quantification (cluster of peroxisomes may be counted as one peroxisome by the FiJi algorithm). In HEK293T cells we see a certain amount of enlarged, potentially clustered peroxisomes in silenced conditions –

Full Revision

From Figure 4G – arrowhead indicates enlarged/clustered peroxisome.

Reviewer #2 (Significance (Required)):

- Describe the nature and significance of the advance (e.g. conceptual, technical, clinical) for the field.

The major nature of the advance is that it solidifies/confirms some of the concepts on how peroxisomes can multiply, by using their connections to the ER to promote peroxisome elongation and biogenesis. As the authors have done in Figure 5K we can now potentially add PKC upstream of GSK3-beta in this pathway of regulating the VAP-ACBD5 tethers. From this study a role of the PO-ER tethers can be more firmly linked to peroxisome elongation and biogenesis, although direct evidence is still lacking. We can now potentially link a role for peroxisomes in other PKC related processes, as the authors have done for the process of differentiation. So, some nice observations and connections with some advances in knowledge. As GSK3-beta is a well-known PKC target and GSK3-beta is already known to regulate the PO-ER tethers this of course reduces the overall significance of the work. However, it is much more significant that the PKC regulation data has come from an unbiased screening process rather than a biased, candidate-testing approach. The screen also provides several other regulatory mechanisms to pursue which can be followed up in future work.

There is solid data linking PKC to regulation of GSK3-beta and the ACBD5-VAPB axis. The rest of the work and the discussion around Pex11 is also solid negative data but this does not really add to our understanding. However, as very little is known about regulation of peroxisomes then this study represents a great starting point which can be built on in future work looking at PKC.

- Place the work in the context of the existing literature (provide references, where appropriate). This I have mostly done above. It fits with existing literature and is an incremental increase in knowledge with good potential for future work in this area of PKC regulation.

- State what audience might be interested in and influenced by the reported findings.

This will of course appeal to peroxisome researchers but also audiences interested in regulation

Full Revision

of organelles and PKC signalling. This is a very nice study, with a well-utilised initial screen and if the authors address some of the issues I have highlighted to make the outcomes more clear then this will be clearly be worthy of publication in a cell biology-focused journal.

- Define your field of expertise with a few keywords to help the authors contextualize your point of view. Indicate if there are any parts of the paper that you do not have sufficient expertise to evaluate

Organelles, membrane contact sites, lipid metabolism

[Response] We thank this reviewer for thorough reading of our manuscript.

Reviewer #3 (Evidence, reproducibility and clarity (Required)):

This manuscript uses a kinase inhibitor screen in human HEK293T cells to identify inhibitors that reduce or increase peroxisome abundance (peroxisome numbers/ μm^2). The authors found that 16 kinase inhibitors reduced, and 5 upregulated, the number of peroxisomes in HEK293T cells. They then undertook a secondary screen of the hits in untagged human fibroblasts, validating 10 compounds that reduced and 2 that upregulated the number of peroxisomes. Positive regulators (wherein inhibition of the kinase reduces peroxisome number) included: PKC, TGF β R, MEK1/2, ERK2, PDHK, IGF1R, and ALK4/7. Negative regulators (wherein inhibition of the kinase increases peroxisome numbers) were CK2 and IKK β .

Further work involved validation of the role of PKC in the regulation of peroxisome abundance. Because there are multiple classical (cPKC: α , β . and γ), novel (nPKC δ), and atypical (aPKC: ζ) PKC isoforms, the authors tried to see if any inhibitor tested was specific to one PKC isoform and found that this was not the case. Instead, they turned to overexpression of individual PKC isoforms and found that peroxisome proliferation was specific to PKC δ overexpression. Using the phorbol ester, PMA, which activates all PKC isoforms, peroxisome abundance was enhanced in untagged human fibroblasts. The authors interpret these data as an indication of PKC as a positive regulator of peroxisome abundance. Consistent with this idea, inhibition of PKC isoforms in primary human fibroblasts reduced fatty acid oxidation relative to control.

The authors assessed the de novo formation of peroxisomes (by inducing PEX19 in PEX19-KO cells) in the presence and absence of PKC inhibitors and found that PKC was not involved de novo biogenesis. Instead, it affected the PEX11 β -mediated division of peroxisomes. PEX11 β KO cells had reduced peroxisome abundance as was previously shown, but they did not respond to PKC inhibition by further reduction of peroxisome numbers, suggesting PEX11 β requirement for the action of the inhibitor. They ruled out that PKC acts through PPAR α activation. They also ruled out increased pexophagy as the cause of reduced peroxisome abundance in the presence of PKC inhibitors, by showing that depletion of one of the

Full Revision

pexophagy receptors, NBR1, had no effect on peroxisome abundance.

Since yeast Pex11 is known to be activated for peroxisome division via phosphorylation, the PKC-phosphorylation site (Ser53 and other mapped sites of phosphorylation) in PKC was mutated but this was not a factor. However, PKC was found to regulate peroxisome-ER contact sites through inhibition of GSK3 β , which had been shown previously by the Schrader lab to negatively regulate ER-peroxisome contact via phosphorylation of peroxisomal ABCD5 and its interaction with VAPB. Finally, to assess the physiological relevance of this mode of regulation of peroxisome abundance, differentiating human neuronal cells (human neuroblastoma SH-SY5Y) were examined. PKC activity and peroxisome abundance were significantly increased in differentiating cells. Similarly, an increase in PKC activity and peroxisome number was observed during the differentiation of iPSC to neuronal progenitor cells. Inhibition of peroxisome proliferation with PKC inhibitor did not prevent neuronal differentiation of the SH-SY5Y cells even though peroxisomes were reduced.

Referees cross-commenting

this session contains comments of Rev 2 and Rev 3

Rev 3

Point #3, reviewer 3 may have been missed by the other reviewers. The modulation of ER-peroxisome contacts may have nothing to do with the increase in peroxisome abundance or be indirect. Until Drp1 (point2, reviewer 3), the more likely target of GSK3beta is tested, the data as presented maybe right or wrong. I agree with reviewer 2 that it is not worth chasing after Pex11beta with better experiments (e.g reviewer 1, point 2). The paper's emphasis on Pex11b-dependence could just as easily be Drp1-dependence (which acts in the same pathway as Pex11beta)

Rev 2

Yes, I am not so familiar with this role of GSK3-beta (this protein has so many roles!) and I had missed that Drp1 was also a PKC and GSK3 target.

However:

"Phosphorylation of Drp1 at S693 by GSK3 negatively affects Drp1 function".

If this is accurate and I understand correctly then inhibition of GSK3-beta may then be expected to give less Drp1 S693 phosphorylation and therefore effectively ACTIVATE Drp1? Or at least reduce its inhibition? If the decreased peroxisome abundance seen here with the inhibitors is Drp1 dependent then would you expect less peroxisomes/peroxisome density due to activation of Drp1? In my mind this does not fit. It has been well published that loss of Drp1 activity results in elongated peroxisomes and mitochondria but I am not aware of work looking at the outcome of an activation of Drp1 on peroxisome number/elongation.

Rev 3

Drp1 phosphorylation at different sites can be activating or inhibiting, so it is hard to predict the outcome regarding peroxisome abundance. As stated in point 2, Drp1 phosphorylation is responsive to PKC-delta activation and inhibition, while also being involved in peroxisome fission, making it a reasonable target - more so than Pex11beta.

Reviewer #3 (Significance (Required)):

The strength of this study is that it shows that PKC activation, and GSK3 β inhibition, are involved, both in vitro and physiologically, in the regulation of peroxisome abundance in human cells and this is driven primarily by affecting PEX11 β -mediated peroxisome division and not by affecting de novo peroxisome biogenesis and possibly pexophagy. The somewhat weaker conclusion is that ER-peroxisome contacts regulate peroxisome division. The manuscript should be of broad interest to cell biologists and scientists in the peroxisome field. The manuscript does

Full Revision

have several issues that would need to be addressed before being considered to publication, but this should be encouraged.

My expertise is in peroxisome homeostasis and mechanisms of biogenesis and turnover.

[Response] We thank this reviewer for their constructive comments and thorough reading of our manuscript. Please find point-by-point response below.

Major

1. NBR1 is not the only pexophagy receptor (PMID: 36810161) so one should be cautious about the assumption that if NBR1 is knocked down, then pexophagy is completely inhibited (Fig. S4).

[Response] Thank you for indicating that. We included an experiment where we silenced NIX and NBR1 together in control and PKC activation conditions. We observed upregulation in peroxisome number in both control and silencing during PMA treatment, supporting the conclusion that PKC does not act through pexophagy regulation (Figure S4G).

2. There is no direct evidence in mammalian cells that PEX11 β phosphorylation drives peroxisome division, as it does in some but not all yeast. So, it is not surprising that PEX11 β phosphorylation did not affect peroxisome proliferation by PKC. However, a more direct test would be for PKC activation of Drp1, the dynamin-like protein involved in peroxisome and mitochondrial division. PKC δ induces phosphorylation of Drp1. Additionally, PKC δ inhibitor (Go 6983) or PKC δ siRNA reversed phosphorylation of Drp1 (PMID: 35490835). So, the effect of PKC activation or inhibition on Drp1 phosphorylation (especially at S579 which is believed to activate Drp1) should be examined. Notably, phosphorylation of Drp1 at S693 by GSK3 negatively affects Drp1 function (PMID: 23185298). If this hypothesis about Drp1 as the target of PKC is true, there need not be any direct involvement of contact sites, which may be coincidental rather than causal.

[Response] We explored the effect of PKC activation on DNM1L/Drp1 and the effects of mutations suggested by the reviewer. First, we established an assay by creating DNM1L KO cells (Figure S7E). DNM1L KO cells have elongated and sometimes beaded peroxisomes which can be restored to normal multiple small peroxisomes by complementation of wildtype DNM1L-mCherry (Figure S7F). We assumed that during short-term overexpression of DNM1L peroxisome restoration is mainly due to division of elongated peroxisomes, in this case inactive DNM1L would lead to a decrease in restoration when overexpressed in KO background. We examined the mutations that were recommended in human DNM1L - S616 corresponding to the mouse S579 recommended by the Reviewer and 693A (in transcript-219 or 590 and 667 in transcript-206 of DNM1L that we used for the paper). Phosphomimic mutations of human DNM1L as well as non-phosphorylated forms restored peroxisomes similarly to the wildtype (Figure S8A-B). Mutation 590/616 (corresponding to the 579 residue that activates PKC in mouse DNM1L) restored peroxisomes to the same extent and restoration was not dependent on PKC inhibition (Figure S8C)

Full Revision

Additionally, we measured mitochondrial fission during PMA treatment that increases the number of peroxisomes and didn't observe mitochondrial division (DNM1L activation would lead to an increase in mitochondrial fission) (Figure S7D).

From these results we conclude that PKC acts upstream of peroxisome division factors. To confirm its effect on elongation of peroxisomes we used MFF KO. MFF KO has elongated peroxisome structures. Treating MFF KO with PKC inhibitors leads to a reduction in the length of peroxisomes, indicating that PKC regulates earlier steps preceding division (Figure 3I-J). Since steps earlier than MFF are preceding DNM1L we think that the DNM1L phosphorylation by PKC (which is not disputed, and we also see the interaction of PKC with DNM1L in a different study) is not the molecular driver of peroxisome biogenesis during PMA activation.

3. One concern in the data interpretation regarding the direct importance of ER-peroxisome contacts is that the VAP KO is used. VAPA/B make contacts between the ER and many organelles, making it challenging to assign the entire phenotype solely to the loss of ER-peroxisome contacts. It would be much more informative to knock out ACBD5, which should affect mostly peroxisome contacts. Since the effect of PKC activation or inhibition on VAP contacts with other organelles was not measured, it would be an overinterpretation to assign the peroxisome abundance phenotypes solely to ER-peroxisome contacts. Stronger evidence (e.g. ACBD5 KO) should be used to implicate primarily peroxisome-ER contacts.

[Response] We think that although VAPA/B KO cell model has its limitation, it still shows that the contacts (between ER and peroxisomes or other organelles tethered to ER through VAPs) have direct effect on peroxisome biogenesis. ACBD5 KO does not lead to a decrease in peroxisome numbers and has 50% of contacts remaining (shown by Bishop et al. [10.1177/2515256419848641](https://doi.org/10.1177/2515256419848641)), therefore it wouldn't recapitulate peroxisome detachment to the same extent as VAP KO. We also didn't find a decrease in peroxisome number in ACBD5 KO in HEK293T cells (Figure S1J). As per reviewer suggestion we added the limitation to the discussion, pointing towards the VAP regulation of other contact sites.

Minor

1. The quantitation upon which this screen is based is poorly described and possibly limited. It is unclear how the cell segmentation was performed and whether Z-stack images were converted to MIPs for the analyses. It would also be useful to know what a give peroxisome/area means - is this the mean or median for a cell or a population treated a certain way? These details are important for reproducibility.

[Response] We now include the information in the method section. The screen was performed using confocal 2D images, as 3D images were increasingly hard to process for the number of peroxisomes due to overlaps between peroxisomes in the maximum intensity projections, especially when the number of peroxisomes increased. Processing confocal images was done using FiJi software, peroxisomes were identified using Find Maxima plugin after assigning a threshold level using Threshold function based on the control sample (consistent among all the processed images). The area represents a cell cytoplasmic space which was outlined manually

Full Revision

guided by the Threshold function, nuclei were not included in the areas. We added the details into the method section – Peroxisome abundance measurements.

2. In general, the experimental details are very sparse, impacting reproducibility. All drug concentration used, as well as antibody dilutions and promoters used for overexpression of PKC isoforms and PEX genes should be specified consistently.

[Response] We now included the data on drug concentrations and biological repeats in the legends, antibody dilutions in the method section, and promoters for all the plasmids in the plasmid construction section.

3. In yeast, GFP-tagging of Pex1 blocks its function in peroxisome division but not is peroxisomal location. Were any experiments done (e.g. comparison of peroxisome abundance in WT versus complemented PEX11 KO cells) to show that myc-PEX11 is fully functional because if not, both WT and mutants may be insensitive to PKC modulation.

[Response] We now included an experiment showing that complementation of PEX11b KO with myc-PEX11b plasmid restores peroxisome number, which indicates that PEX11b tagged with myc is functioning as peroxisome division factor (Figure S7C).